# Enhancing Performance of Explainable AI Models with Constrained Concept Refinement

**Geyu Liang**[1]  **Senne Michielssen**[2]  **Salar Fattahi**[1]

## Abstract

The trade-off between accuracy and interpretability has long been a challenge in machine learning (ML). This tension is particularly significant for emerging *interpretable-by-design* methods, which aim to redesign ML algorithms for trustworthy interpretability but often sacrifice accuracy in the process. In this paper, we address this gap by investigating the impact of deviations in concept representations—an essential component of interpretable models—on prediction performance and propose a novel framework to mitigate these effects. The framework builds on the principle of optimizing concept embeddings under constraints that preserve interpretability. Using a generative model as a test-bed, we rigorously prove that our algorithm achieves zero loss while progressively enhancing the interpretability of the resulting model. Additionally, we evaluate the practical performance of our proposed framework in generating explainable predictions for image classification tasks across various benchmarks. Compared to existing explainable methods, our approach not only improves prediction accuracy while preserving model interpretability across various large-scale benchmarks but also achieves this with significantly lower computational cost.

## 1. Introduction

ML algorithms are often caught in a dilemma between interpretability and performance. Models such as linear regression (Hastie, 2009) and decision trees (Quinlan, 1986) provide straightforward interpretability through parameter weights and rule-based predictions. However, they fre-quently underperform on complex tasks. On the other hand, high-performing models, such as deep neural networks (Le-Cun et al., 2015) and large language models (Vaswani, 2017), are notoriously opaque, given their large parameter spaces and intricate architectures.

Numerous methods have been proposed to extract interpretability from complex models (Baehrens et al., 2010; Simonyan et al., 2013; Zeiler & Fergus, 2014; Shrikumar et al., 2017; Selvaraju et al., 2017; Smilkov et al., 2017; Kolek et al., 2020; Subramanya et al., 2019). However, these approaches typically adopt post-hoc strategies, utilizing sensitivity analysis to identify the key parameters that influence predictions. Consequently, these methods lack guarantees of providing explanations for random prediction or ensuring their trustworthiness (Adebayo et al., 2018; Rudin, 2019; Kindermans et al., 2019; Ghorbani et al., 2019; Slack et al., 2020).

An alternative solution is to develop models that are *interpretable by design*. These models intrinsically integrate transparency into their architecture, providing explanations directly tied to their predictions. Concept Bottleneck Models (CBMs, (Koh et al., 2020)) exemplify this approach by mapping input data to an intermediate representation of human-defined concepts, which is then used for prediction. While concept-based models are promising, they require datasets annotated with concept scores, limiting their applicability. Extensions of CBMs aim to improve the approach by utilizing pretrained encoders like CLIP (Yuksekgonul et al., 2022; Oikarinen et al., 2023). A recent line of research (Chattopadhyay et al., 2022; 2023; 2024) has introduced a decision-tree-based architecture with enhanced explainability. The predictions of the model are generated through a sequence of queries, each closely associated with human-specified concepts.

A high level design paradigm of expainable AI is illustrated in Figure 1. Within these models, concept embeddings play a pivotal role: they not only influence prediction generation but also serve as the source of interpretability. However, recent studies have raised several questions against its reliability. These challenges are centered around its ambiguity (Margeloiu et al., 2021; Watson, 2022; Kim et al., 2023; Marconato et al., 2023), fragility (Furby et al., 2024), and

[1]Department of Industrial and Operations Engineering, University of Michigan, Ann Arbor, MI, US. [2]Department of Computer Science, Princeton University, Princeton, NJ, US. Correspondence to: Salar Fattahi <fattahi@umich.edu>.

*Proceedings of the 42nd International Conference on Machine Learning*, Vancouver, Canada. PMLR 267, 2025. Copyright 2025 by the author(s).

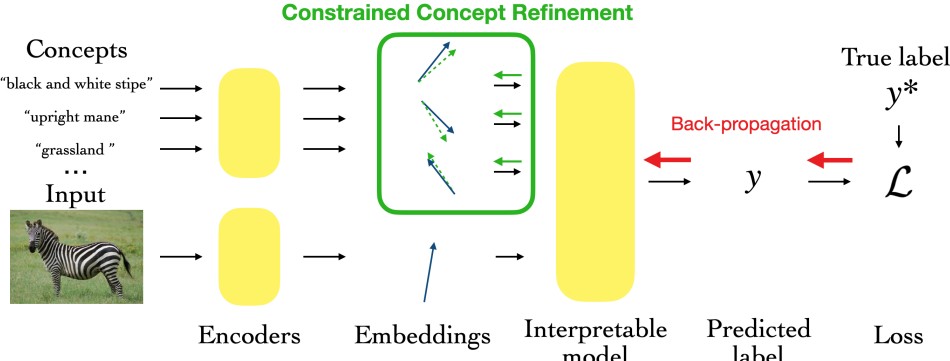

*Figure 1.* The red arrows represent the backpropagation training process for classic explainable AI models. This paper extends the training process to refine concept embeddings with constraints on their deviation from initial embeddings, represented by green arrows and box.

low accuracy (Zarlenga et al., 2022; Chowdhury et al., 2024). For embeddings generated with pretrained models, their encoders are reported to face issues such as domain adaptation (Shao et al., 2023; Gondal et al., 2024; Feng et al., 2024; Wang & Kang, 2024), biases in pretraining data (Hamidieh et al., 2024), and sensitivity to encoding errors (Ranjan et al., 2024).

Efforts to address these challenges include introducing unsupervised modules to enhance concept expressiveness (Sawada & Nakamura, 2022), training residual learning modules in parallel with concept encoders (Yuksekgonul et al., 2022), learning dual embeddings per concept (Zarlenga et al., 2022), and adding specialized data encoders to address domain shifts (Chowdhury et al., 2024). However, these approaches all involve introducing additional black-box learning modules, which undermines the purpose of explainable AI. Moreover, theoretical guarantees for such approaches, which is of particular interest for safety-critical domains, such as medical imaging (Barragán-Montero et al., 2021) and autonomous vehicles (Bensalem et al., 2023), remain largely unexplored.

To address these challenges, this paper proposes a novel framework termed *Constrained Concept Refinement (CCR)*. Unlike existing methods that introduce additional learning modules, our approach directly optimizes concept embeddings within a constrained parameter space. By restricting embeddings to a small neighborhood around their initialization, our framework offers a tunable trade-off between performance and interpretability, and in certain settings, it can simultaneously improve both. Specifically, we present the following contributions:

- **Theoretical necessity.** We rigorously demonstrate the necessity of refining concept embeddings by establishing a non-vanishing worst-case lower bound on model performance when such refinements are not applied

to the concepts (Theorem 2.6). This highlights and motivates the idea of concept embedding refinement, which is at the crux of our proposed method.

- **Accuracy and interpretability guarantees.** We show that CCR overcomes the aforementioned challenge by eliminating performance degradation through the appropriate refinement of concepts (Theorem 3.3). Furthermore, to quantify the performance of CCR, we consider a generative model as a test-bed, where both "interpretability" and "accuracy" admit crisp mathematical definitions. Under this model, we demonstrate that CCR achieves zero training loss while progressively enhancing interpretability (Theorem 3.4).

- **Application in interpretable image classification.** We demonstrate the practical efficacy of CCR on multiple image classification benchmarks including CIFAR 10/100 (Krizhevsky et al., 2009), ImageNet (Deng et al., 2009), CUB200 (Wah et al., 2011) and Places365 (Zhou et al., 2017). In all benchmarks except CUB-200, CCR outperforms two recently developed explainable methods for visual classification tasks—CLIP-IP-OMP (Chattopadhyay et al., 2024) and label-free CBM (lf-CBM) (Oikarinen et al., 2023)—in terms of prediction accuracy while preserving interpretability, achieving this with a tenfold reduction in runtime.

On a high level, our main contribution is the introduction of a framework in which concept embeddings in explainable AI models are refined within a restricted parameter space to attain a better balance between predictive performance and interpretability. The remainder of the paper substantiates this claim along two key dimensions:

1. **Theoretical validation.** We demonstrate that our method is both **necessary** (Section 2.1) and **effective** (Section 3) under the theoretical framework introduced

by (Chattopadhyay et al., 2024), whose background is reviewed in Section 2. This framework serves as a suitable testbed for our contributions for two main reasons: (1) Among interpretable-by-design models (Zhou et al., 2018; Koh et al., 2020; Yuksekgonul et al., 2022; Oikarinen et al., 2023; Chattopadhyay et al., 2023), only (Chattopadhyay et al., 2024) presents a generative model wherein both performance and interpretability are rigorously defined. (2) The algorithm motivated by this generative framework achieves state-of-the-art results within the class of interpretable AI methods.

2. **Empirical evaluation.** We conduct experiments on multiple benchmark datasets for image classification tasks to assess the practical effectiveness of our approach (Section 4).

**Notations.** For a matrix $\boldsymbol{A}$, we denote its spectral norm by $\|\mathbf{A}\|_2$, Frobenius norm by $\|\mathbf{A}\|_F$, and maximum column-wise $\ell_2$-norm by $\|\mathbf{A}\|_{1,2}$. Define $[n] = \{1, 2, \ldots, n\}$. A matrix $\boldsymbol{A} \in \mathbb{R}^{n \times m}$ is called column-orthogonal if $\boldsymbol{A}^\top \boldsymbol{A} = \boldsymbol{I}_{m \times m}$. Events with probability at least $1 - n^{-\omega(1)}$ are said to occur with high probability.

## 2. Revisiting IP-OMP

We consider a classical *interpretable-by-design* setting in the literature (Chattopadhyay et al., 2022; 2024), namely the task of predicting a random variable by sequentially selecting a list of related random variables, termed *queries*, and checking their values. The problem is defined as follows:

**Definition 2.1** (Single Variable Prediction by Query Selection). Given an *input feature* $\mathbf{x} \in \mathbb{R}^d$, a set of *queries* $\{q_i\}_{i=1}^n \subset \mathbb{R}$, and their associated *query features* (also referred to as *concept embeddings*) $\{\mathbf{d}_i\}_{i=1}^n \subset \mathbb{R}^d$, the objective is to predict the *target random variable* $y \in \mathbb{R}$ by selecting at most $k$ queries from $\{q_i\}_{i=1}^n$, leveraging the query features $\{\mathbf{d}_i\}_{i=1}^n$ to guide the selection process. Specifically, the query set $\{q_i\}_{i=1}^n$ consists of random variables that are correlated with $y$, whose observed values can contribute to predicting $y$.

Definition 2.1 encompasses a broad spectrum of ML problems with applications in animal identification (Lampert et al., 2009), medical diagnosis (Graziani et al., 2018; Clough et al., 2019), visual question-answering (Yi et al., 2018), image retrieval (Bucher et al., 2019), and so on.

One illustrative example is the game played among friends, where the target random variable refers to the answer to the question "what am I thinking now?". In this context, the queries $\{q_i\}_{i=1}^n$ are binary random variables representing "yes/no" answers to specific questions like "Is it an animal?". Here, the query feature is a vector embedding of each ques-

tion, encapsulating the information relevant to that question. Additionally, the participants are limited to asking at most $k$ questions.

In (Geman & Jedynak, 1996), the authors introduced a classic greedy-style algorithm termed *Information Pursuit* (IP) to address the task outlined in Definition 2.1. Specifically, IP iteratively selects the most informative query based on the observed values of previously selected queries. We provide a formal definition of IP for subsequent discussion:

**Definition 2.2** (Information Pursuit). At every iteration $t = 1, \ldots, k$, IP selects $\pi(t)$-th query according to:

$$\pi(t) = \arg\max_{1 \le i \le n} \Big\{ I(q_i, y \mid q_{\pi(1)} = r_{\pi(1)},$$

$$q_{\pi(2)} = r_{\pi(2)}, \ldots, q_{\pi(t-1)} = r_{\pi(t-1)}) \Big\}.$$

Here $I(\cdot, \cdot)$ denotes mutual information, $\pi(\cdot) : [k] \to [n]$ is an injective map representing the selection of queries, and $r_i$ denotes the observed value for $q_i$. The final output of IP is defined as the maximum likelihood estimator given all observed queries:

$$y_{\text{IP}} = \arg\max_{\tilde{y}} \Big\{ \mathbb{P}(y = \tilde{y} \mid q_{\pi(1)} = r_{\pi(1)},$$

$$q_{\pi(2)} = r_{\pi(2)}, \ldots, q_{\pi(k)} = r_{\pi(k)}) \Big\}. \quad (1)$$

In (Chattopadhyay et al., 2023), IP is demonstrated to achieve highly interpretable predictions with competitive accuracies. However, a significant limitation of IP is its substantial computational expense, as mutual information depends not only on the set $\{\mathbf{d}_i\}_{i=1}^n$ but also on previous observations, leading to an exponentially large input space. (Chattopadhyay et al., 2024) addresses this complexity by imposing additional assumptions on the underlying generative model. The first quantifies the connection between $y$ and $q_i$, while the second defines their connections with $\mathbf{d}_i$.

**Assumption 2.3** (Generative model for IP-OMP). There exists an unknown random vector $\mathbf{z} \in \mathbb{R}^d$ drawn from a standard Normal distribution that satisfies $y = \langle \mathbf{x}, \mathbf{z} \rangle$. Moreover, the queries satisfy $q_i = \langle \mathbf{v}_i, \mathbf{z} \rangle$, where $\mathbf{v}_i$ is a *latent feature vector* associated with query $i$.

Under the above assumption, the target random variable and the queries are correlated through the unknown random variable $\mathbf{z}$. In this setting, a natural way to define the query features is by setting $\mathbf{d}_i = \mathbf{v}_i$. This entails an exact prior knowledge of the latent feature vectors.

**Assumption 2.4** (Prior knowledge of latent feature vectors). The latent feature vectors $\{\mathbf{v}_i\}_{i=1}^n$ are precisely observed.

Under Assumption 2.3 and 2.4, (Chattopadhyay et al., 2024) establish a connection between IP and Orthogonal Matching Pursuit (OMP) (Pati et al., 1993), a seminal algorithm in the field of sparse coding.

**Theorem 2.5** (IP-OMP (Chattopadhyay et al., 2024)). *Under Assumption 2.3 and 2.4 and upon setting* $\mathbf{d}_i = \mathbf{v}_i$ *for* $i = 1, \ldots, n$, *the selection rule* $\pi(\cdot)$ *defined in Definition 2.2 admits a closed-form expression given by:*

$$\pi(t) = \arg\max_{1 \leq i \leq n} \frac{|\langle \Pi^{\perp}_{\boldsymbol{D}^{(t-1)}} \mathbf{d}_i, \Pi^{\perp}_{\boldsymbol{D}^{(t-1)}} \mathbf{x} \rangle|}{\|\Pi^{\perp}_{\boldsymbol{D}^{(t-1)}} \mathbf{d}_i\|_2 \|\Pi^{\perp}_{\boldsymbol{D}^{(t-1)}} \mathbf{x}\|_2},$$

*where* $\boldsymbol{D}^{(t-1)} = \begin{bmatrix} \mathbf{d}_{\pi(1)} & \mathbf{d}_{\pi(2)} & \cdots & \mathbf{d}_{\pi(t-1)} \end{bmatrix}$. *This selection criterion, referred to as IP-OMP, differs from OMP solely by the inclusion of the normalization term* $\|\Pi^{\perp}_{\boldsymbol{D}_{t-1}} \mathbf{d}_i\|_2 \|\Pi^{\perp}_{\boldsymbol{D}_{t-1}} \mathbf{x}\|_2$.

The main theoretical contribution of Theorem 2.5 lies in its ability to transform the computation of mutual information into that of a linear projection, thereby significantly reducing computational complexity. Although the equivalence is derived under simplifying assumptions, IP-OMP demonstrates remarkable generalization capabilities across various benchmark experiments, outperforming models based on the classical IP algorithm (Chattopadhyay et al., 2024).

### 2.1. A Major Hurdle: Inaccurate Query Features

As discussed in the previous section, Theorem 2.5, and as a result, the success of IP-OMP is contingent upon correctly choosing the query features $\{\mathbf{d}_i\}_{i=1}^{n}$.

For learning algorithms, the effective utilization of $\{\mathbf{d}_i\}_{i=1}^{n}$ is crucial for the accurate prediction of $y$. On the other hand, humans derive comprehension from $\{\mathbf{d}_i\}_{i=1}^{n}$ as these elements are intended to embed concepts that are interpretable by humans. In practical applications, $\{\mathbf{d}_i\}_{i=1}^{n}$ are either embedded and learned from predefined datasets where concepts are explicitly labeled (Koh et al., 2020) or generated by pretrained multimodal models, with CLIP being the most popular example (Oikarinen et al., 2023).

However, these learned embeddings are often misaligned or inaccurate due to the inherent ambiguity and noise in the training of the models used to generate. For example, unlike carefully curated datasets, CLIP relies on large-scale, noisy image-text pairs scraped from the internet. These pairs often include mislabeled data, vague descriptions, or culturally biased associations, resulting in inconsistencies in representation, as recently reported by (Dutta et al., 2023).

This suggests that Assumption 2.4 may be overly optimistic. In this section, we will illustrate how the violation of Assumption 2.4—specifically, deviations of the available query features from the latent feature vectors—results in a proportional degradation in the performance of IP-OMP. For the remainder of this paper, we denote the observed query feature set compactly by the matrix $\boldsymbol{D} = \begin{bmatrix} \mathbf{d}_1 & \mathbf{d}_2 & \cdots & \mathbf{d}_n \end{bmatrix} \in \mathbb{R}^{d \times n}$. This matrix $\boldsymbol{D}$ is also referred to as a *feature query matrix*. We also refer to $\boldsymbol{D}^* = \begin{bmatrix} \mathbf{v}_1 & \mathbf{v}_2 & \cdots & \mathbf{v}_n \end{bmatrix} \in \mathbb{R}^{d \times n}$ as the ground truth

latent feature matrix. Assumption 2.4 can now be interpreted as assuming $\boldsymbol{D} = \boldsymbol{D}^*$, stating that we are using the ground truth latent feature vectors as the feature query matrix. This matrix can then be utilized to determine which queries should be observed.

Let $f_{\boldsymbol{D}}(\mathbf{x}, \{r_{\pi(i)}\}_{i=1}^{k})$ be the maximum likelihood estimator of $y$, defined as Equation (1), when the selection rule $\pi(\cdot)$ is obtained by executing IP-OMP using $\boldsymbol{D}$ as the query feature set in Theorem 2.5. For brevity, when the context is clear, we refer to this maximum likelihood estimator simply as $f_{\boldsymbol{D}}$. Moreover, define $\mathcal{L}(y_{\text{pred}})$ as the squared population loss of any estimator $y_{\text{pred}}$:

$$\mathcal{L}(y_{\text{pred}}) = \mathbb{E}_{\mathbf{z} \sim \mathcal{N}(\mathbf{0}, \boldsymbol{I}_{d \times d})} \left[ (y - y_{\text{pred}})^2 \right].$$

where $\mathbf{z} \sim \mathcal{N}(\mathbf{0}, \boldsymbol{I}_{d \times d})$ is the unknown random vector used in the generative model described in Assumption 2.3.

To showcase the effect of deviation in the query features, we consider a scenario where the input $\mathbf{x}$ can be written as a linear combination of $k$ latent feature vectors (corresponding to $k$ columns of $\boldsymbol{D}^*$). This assumption is prevalent in the sparse coding literature (Arora et al., 2015; Agarwal et al., 2016; Liang et al., 2022) and ensures that using $\boldsymbol{D}^*$ as the query feature set guarantees good prediction performance. However, even in this ideal scenario, we demonstrate that even a small column-wise deviation from $\boldsymbol{D}^*$ can lead to performance degradation of IP-OMP.

**Theorem 2.6.** *Suppose that Assumption 2.3 holds. Furthermore, suppose that* $\boldsymbol{D}^*$ *is column-orthogonal and* $\mathbf{x} = \boldsymbol{D}^* \boldsymbol{\beta}$ *for some $k$-sparse vector $\boldsymbol{\beta}$ with* $\|\boldsymbol{\beta}\|_0 = k$. *Additionally, assume that the non-zero entries in $\boldsymbol{\beta}$ have absolute values bounded below by $\gamma$ and above by $\Gamma$. Then, for any* $\epsilon \in \left(0, \frac{1}{\sqrt{1 + 16\Gamma^2/\gamma^2}}\right)$, *there exists another orthonormal matrix* $\tilde{\boldsymbol{D}} \in \mathbb{R}^{d \times n}$ *such that* $\|\tilde{\boldsymbol{D}} - \boldsymbol{D}^*\|_{1,2} \leq \epsilon$ *and*

$$\mathcal{L}(f_{\tilde{\boldsymbol{D}}}) - \mathcal{L}(f_{\boldsymbol{D}^*}) \geq \frac{81(k-1)\epsilon^2\gamma^2}{200}. \tag{2}$$

We present the complete proof of Theorem 2.6 in Appendix C.1, which proceeds as follows: (1) we derive the closed form solution for $\mathcal{L}(f_{\boldsymbol{D}})$ using the column-orthogonality of $\boldsymbol{D}$ and the distribution of $\mathbf{z}$; (2) we then construct the example that satisfies Equation (2) by rotating columns of $\boldsymbol{D}^*$ alongside the subspace spanned by itself.

When $\Gamma/\gamma$ is bounded by a constant, we have $\|\mathbf{x}\|_2^2 = \|\boldsymbol{\beta}\|_2^2 = \Theta(k\gamma^2)$ which simplifies Equation (2) to $\mathcal{L}(f_{\tilde{\boldsymbol{D}}}) - \mathcal{L}(f_{\boldsymbol{D}^*}) = \Omega(\epsilon^2\|\mathbf{x}\|_2^2)$. This indicates that the perturbation $\epsilon$ is fully captured by the resulting gap in the squared loss, scaled by a factor of $\|\mathbf{x}\|_2^2$.

# 3. Our Proposed Framework

Building upon our discussion in the preceding section, it is pertinent to consider the following question:

*Since an inaccurate $D$ can adversely affect the performance of IP-OMP, can $D$ be optimized to mitigate this effect while preserving—or even enhancing—interpretability?*

There are two critical challenges to address before answering this question. First, modifying $D$ may diminish the encoded information that is interpretable by humans, thereby compromising the *interpretable-by-design* principle of information pursuit. Second, treating the query features as variables introduces significant complexity in deriving the optimal decision rule, as the closed-form expression of IP-OMP presented in Theorem 2.5 is no longer valid.

To overcome these challenges, we propose obtaining a *correction* $\Delta D$ that minimizes $\mathcal{L}(\tilde{f}_{D+\Delta D})$, where $\tilde{f}_D$ serves as a differentiable surrogate of the original estimator produced by IP. To preserve interpretability, we constrain the corrected query feature matrix $D + \Delta D$ to remain within a small neighborhood around $D$, where $D$ represents the potentially inaccurate initial query feature matrix (e.g., the one obtained from CLIP). This leads to the following constrained optimization problem:

$$\min_{\|\Delta D\|_{1,2} \leq \rho} \mathcal{L}(\tilde{f}_{D+\Delta D}) \tag{3}$$

Indeed, the choice of the correction radius $\rho$ is crucial as it controls the trade-off between prediction accuracy and interpretability. Setting $\rho = 0$ recovers the IP-OMP, and therefore, may suffer from the aforementioned performance degradation. Conversely, choosing a large value for $\rho$ can mitigate the performance degradation of IP-OMP, but at the cost of compromising the interpretability.

Our meta-algorithm for Problem (3), called *constrained concept refinement* (CCR), is presented in Algorithm 1.

---

**Algorithm 1** Constrained Concept Refinement

1: **Input**: Initial query feature matrix $D$, correction radius $\rho$, training dataset $\mathcal{D}$.
2: Initialize $\Delta D^{(0)} = 0_{d \times n}$
3: **while** $t = 1, 2, \ldots, T$ **do**
4:    **Forward propagation:** calculate $\mathcal{L}(\tilde{f}_{D+\Delta D^{(t-1)}})$.
5:    **Backward propagation:** update $\Delta D^{(t)}$ using the gradient $\partial \mathcal{L}/\partial \Delta D^{(t-1)}$.
6:    Perform projection to ensure $\|\Delta D^{(t)}\|_{1,2} \leq \rho$.
7: **end while**
8: Return $\tilde{f}_{D+\Delta D^{(T)}}$

---

Below, we provide a description of its various steps.

**Choice of the surrogate estimator.** To motivate the choice of the surrogate estimator $\tilde{f}_D$, let us revisit the generative model in Assumption 2.3, and additionally assume that the ground truth latent feature matrix $D^*$ is column-orthogonal. By assuming $D^*$ to be column-orthogonal, we effectively assume an ideal scenario in which queries are mutually independent. Specifically, it follows that $\langle d_i^*, z \rangle$ is independent of $\langle d_j^*, z \rangle$ when $d_i^* \perp d_j^*$ and $z \sim \mathcal{N}(0, I_{d \times d})$. The primary motivation for focusing on an column-orthogonal $D^*$ is that, under this assumption, the estimator $f_D$ obtained from IP-OMP becomes inherently differentiable with respect to $D$:

**Lemma 3.1.** *Assuming that $D = D^*$ and $D^*$ is column-orthogonal, we have*

$$f_D\left(x, \{r_{\pi(i)}\}_{i=1}^k\right) = \sum_{i \in S} \langle d_i, x \rangle r_i,$$

*where $S = \arg\max_{T \subseteq [n], |T| \leq k} \sum_{i \in T} |\langle d_i, x \rangle|$ corresponds to the indices of the top-$k$ largest values of $\{|\langle d_i, x \rangle|\}_{i=1}^n$.*

The proof of Lemma 3.1 is deferred to Appendix C.4.1. Assuming a small initial error, i.e., $\|D^* - D\| \leq \rho$ for some small $\rho > 0$, the iterates $D^{(t)} = D + \Delta D^{(t)}$ of Algorithm 1 stay approximately column-orthogonal. This consideration motivates the introduction of the following surrogate estimator $\tilde{f}$:

$$\tilde{f}_D\left(x, \{r_{\pi(i)}\}_{i=1}^k\right) = \sum_{i \in S} \langle d_i, x \rangle r_i,$$
$$\text{where} \quad S = \arg\max_{T \subseteq [n], |T| \leq k} \sum_{i \in T} |\langle d_i, x \rangle|. \tag{4}$$

It is important to note that we do not require $\tilde{f}$ to be differentiable everywhere; rather, differentiability is required only almost surely along the trajectory of Algorithm 1, which we will rigorously demonstrate. Alternative differentiable surrogates for $f$ include the task-driven dictionary learning method (Mairal et al., 2011) and the unrolled dictionary learning method (Malézieux et al., 2021; Tolooshams & Ba, 2021), which can be used in place of Equation (4).

**Backward propagation and projection** For the backward propagation and projection steps, we propose to adopt gradient descent updates, followed by a projection step based on $\ell_2$-norm.

$$\Delta D^{(t+1)}$$
$$= \arg\min_{\|\Delta D\|_{1,2} \leq \rho} \left\| \Delta D - \left( \Delta D^{(t)} - \eta \frac{\partial \mathcal{L}(\tilde{f}_{D+\Delta D^{(t)}})}{\partial \Delta D^{(t)}} \right) \right\|_2, \tag{5}$$

We note that the specific instantiation of the proposed meta-algorithm and its steps is inherently task-dependent. For

example, while we have only discussed the squared loss, a more common choice in image classification tasks is the cross-entropy loss. Additionally, our choice of the surrogate estimator relies on the assumption that the ground truth latent feature matrix is column-orthogonal. When this assumption is not met, it becomes necessary to enforce it via *concept dispersion*. Both of these modifications are discussed extensively within the context of image classification in Section 4.

However, a more fundamental question has remained unanswered: how can "interpretability"—a concept inherently meaningful only to humans—be measured and quantified? One approach is to assess interpretability in an ad-hoc manner by running the candidate method and relying on human judgment to determine whether the predictions are interpretable for different individual samples. This has been the de facto approach adopted in nearly all previous works. An alternative way—which we seek in this work—is to study a simple test-bed for investigation, where "interpretability" admits a crisp mathematical formulation.

### 3.1. Accuracy and Interpretability Guarantees

Consider the following probabilistic generative model.

**Assumption 3.2** (Probabilistic generative model). Suppose that the target random variable $y$ and queries are generated according to Assumption 2.3. Moreover, suppose that ground truth latent feature matrix $\boldsymbol{D}^*$ is column-orthogonal and the input $\mathbf{x}$ satisfies $\mathbf{x} = \boldsymbol{D}^*\boldsymbol{\beta}$, where: (1) the support of $\boldsymbol{\beta}$, denoted as $S^*$, is drawn uniformly from the set of $k$-element subsets of $[n]$; and (2) the non-zero elements of $\boldsymbol{\beta}$ are i.i.d. and satisfy $\mathbb{E}[\boldsymbol{\beta}_i] = 0$, $\mathrm{Var}[\boldsymbol{\beta}_i] = \sigma^2$, and $\gamma \leq |\boldsymbol{\beta}_i| \leq \Gamma$.

The generative model in Assumption 3.2 is akin to those explored in the sparse coding literature (Arora et al., 2015; Agarwal et al., 2016; Ravishankar et al., 2020; Liang et al., 2022). Under this model, prediction error can be evaluated using the squared loss, while interpretability can be assessed by the proximity of the query feature matrix $\boldsymbol{D}^{(T)}$, produced by CCR, to the ground truth latent feature matrix $\boldsymbol{D}^*$.

Our next theorem shows that CCR effectively resolves the challenge faced by IP-OMP, as outlined in Theorem 2.6.[1] Let $\boldsymbol{D}^{(t)} = \boldsymbol{D} + \Delta\boldsymbol{D}^{(t)}$ be the corrected query feature matrix generated by CCR at iteration $t$.

**Theorem 3.3.** *Suppose that Assumption 3.2 holds. Moreover, suppose that that the initial query feature matrix $\boldsymbol{D}$ satisfies $\|\boldsymbol{D} - \boldsymbol{D}^*\|_{1,2} = \rho \leq \frac{\gamma}{8\sqrt{k}\Gamma}$ and the step-size $\eta$ satisfies $0 < \eta < \frac{1}{2\|\mathbf{x}\|_2^2}$. Algorithm 1 with surrogate estima-*

---

[1]We note that the result of Theorem 2.6 also holds for the generative model described in Assumption 3.2.

*tor* (4) *and update rule* (5) *satisfies:*

$$\mathcal{L}(\tilde{f}_{\boldsymbol{D}^{(t+1)}}) \leq \left(1 - 2\eta\|\mathbf{x}\|_2^2\right)^2 \mathcal{L}(\tilde{f}_{\boldsymbol{D}^{(t)}}).$$

While Theorem 3.3 addresses the performance limitations of IP-OMP, it does not guarantee improved interpretability. This outcome is unsurprising, since the reliance of the method on only *a single* input $\mathbf{x}$ ensures that only the columns of $\boldsymbol{D}$ with indices in $S^*$ can be improved, essentially leaving the columns outside $S^*$ unmodified. Moreover, if $\boldsymbol{D}^*$ is not full-rank, our analysis shows that $\boldsymbol{D}$ may deviate from $\boldsymbol{D}^*$ in directions column-orthogonal to the column space of $\boldsymbol{D}^*$; these deviations cannot be eliminated by Equation (5). This observation is further supported by numerical experiments (see Figure 4).

To tackle these challenges, we assume that $\boldsymbol{D}^*$ is full-rank. Moreover, it is essential to use a sufficient number of i.i.d. input samples to ensure that each column of $\boldsymbol{D}^*$ contributes to at least one input sample. This aligns more closely with practical scenarios, such as image classification, where multiple samples are typically used during the training phase. We denote the i.i.d. input samples and their corresponding target random variables, generated according to Assumption 3.2, as $\{\mathbf{x}^h\}_{h=1}^m$ and $\{y^h\}_{h=1}^m$, respectively. For any estimator $\mathbf{y}_{\text{pred}} = \{y_{\text{pred}}^h\}_{h=1}^m$, the aggregated squared loss is defined as

$$\mathcal{L}_m(\mathbf{y}_{\text{pred}}) = \frac{1}{m} \sum_{h=1}^{m} \mathbb{E}_{\mathbf{z}^h \sim \mathcal{N}(\mathbf{0}, \boldsymbol{I}_{d\times d})} \left[(y^h - y_{\text{pred}}^h)^2\right]. \tag{6}$$

**Theorem 3.4.** *Suppose that $\{\mathbf{x}^h\}_{h=1}^m$ and $\{y^h\}_{h=1}^m$ are i.i.d. samples generated from Assumption 3.2 with a fixed full-rank column-orthogonal $\boldsymbol{D}^*$. Suppose that $m = \Omega\left(\frac{n^6}{\sigma^2 k^5}\right)$. Moreover, suppose that the initial query feature matrix $\boldsymbol{D}$ satisfies $\|\boldsymbol{D} - \boldsymbol{D}^*\|_{1,2} = \rho \leq \frac{\gamma}{8\sqrt{k}\Gamma}$ and the step-size $\eta$ satisfies $\eta = O\left(\frac{1}{\sigma^2}\right)$. With high probability, Algorithm 1 with surrogate estimator* (4) *and update rule* (5) *applied to the aggregated squared loss* (6) *satisfies:*

- **(Interpretability)** $\|\boldsymbol{D}^{(t+1)} - \boldsymbol{D}^*\|_{1,2} \leq \tau\|\boldsymbol{D}^{(t)} - \boldsymbol{D}^*\|_{1,2}$, *where* $\tau = \sqrt{1 - \frac{k(k-1)\sigma^2}{2n^2}\eta}$.

- **(Accuracy)** $\mathcal{L}_m(\tilde{f}_{\boldsymbol{D}^{(t)}}) \leq \frac{k\sum_{h=1}^m \|\mathbf{x}^h\|_2^2}{m}\|\boldsymbol{D}^{(t)} - \boldsymbol{D}^*\|_{1,2}^2$.

According to Theorem 3.4, by leveraging multiple samples, CCR achieves the best of both worlds: it guarantees convergence to the ground truth latent feature matrix $\boldsymbol{D}^*$, progressively enhancing interpretability, while simultaneously driving $\mathcal{L}_m$ to zero, thereby achieving perfect accuracy.

Contrary to traditional analyses in dictionary learning (Arora et al., 2015; Liang et al., 2022), the proofs of Theorem 3.3

and Theorem 3.4 require a careful alignment of the updates from gradient descent with the direction of the true solution. In the setting of Theorem 3.4, this is even more challenging because the finite sample size $m$ inevitably causes deviations from the population-level behavior. Moreover, we must further control the deviation introduced by the projection step. The detailed proofs of these theorems are provided in Appendix C. We also include a detailed discussion on both theorems in Appendix B and experiments that validate them in Appendix D.1.

# 4. Application: Interpretable Image Classification

In this section, we showcase the performance of our proposed CCR framework from Algorithm 1 on interpretable image classification task. The Python implementation of algorithm can be found here: github.com/lianggeyuleo/CCR.git.

The image classification setting differs slightly from the setting provided in Definition 2.1. Specifically, we are given a dataset $\{\mathbf{a}^i, \mathbf{y}^i\}_{i=1}^m$ consisting of $m$ image ($\mathbf{a}$)-label ($\mathbf{y}$) pairs, where the goal is to predict the correct label for a given image. Here, $\mathbf{y}$ represents a one-hot label vector. Additionally, this setting provides access to a concept set $\{\mathbf{c}_i\}_{i=1}^n$, which consists of $n$ key concepts that can aid in classifying an image. These concepts are typically textual; for example, in the case of animal images, the concept set might include descriptors like "stocky body" or "black stripes".

---

**Algorithm 2** CCR for Interpretable Image Classification

---

1: **Input**: Dataset of image-label pairs $\{\mathbf{a}^i, \mathbf{y}^i\}_{i=1}^m$, concept set $\{\mathbf{c}_i\}_{i=1}^n$.
2: Use CLIP to embed the images $\{\mathbf{a}^i\}_{i=1}^m$ into input features $\{\mathbf{x}^i\}_{i=1}^m$, and the concepts $\{\mathbf{c}_i\}_{i=1}^n$ into query features $\{\mathbf{d}_i\}_{i=1}^n$.
3: Initialize $\boldsymbol{D}^{(0)}$ via concept dispersion on $\{\mathbf{d}_i\}_{i=1}^n$.
4: Initialize $\mathbf{s}_i^{(0)} = \text{HT}_\lambda\left(\boldsymbol{D}^{(0)^\top}\mathbf{x}^i\right)$ for $i = 1, \ldots, m$.
5: Initialize $\boldsymbol{L}^{(0)}$ with random values.
6: **while** $t = 1, 2, \ldots, T$ **do**
7:     Calculate $\mathcal{L}_m = \sum_i \text{CE}\left(\boldsymbol{L}\mathbf{s}_i^{(t-1)}, \mathbf{y}^i\right)$.
8:     Update $\boldsymbol{D}^{(t)} = \boldsymbol{D}^{(t-1)} - \eta_D\, \partial\mathcal{L}/\partial\boldsymbol{D}^{(t-1)}$.
9:     Normalize and project $\boldsymbol{D}^{(t)}$.
10:    Update $\boldsymbol{L}^{(t)} = \boldsymbol{L}^{(t-1)} - \eta_L\, \partial\mathcal{L}/\partial\boldsymbol{L}^{(t-1)}$.
11:    Update $\mathbf{s}_i^{(t)} = \text{HT}_\lambda\left(\boldsymbol{D}^{(t)^\top}\mathbf{x}^i\right)$ for $i = 1, \ldots, m$.
12: **end while**
13: **Return** $\boldsymbol{D}^{(T)}$, $\boldsymbol{L}^{(T)}$, and $\{\mathbf{s}_i^{(T)}\}_{i=1}^m$.

---

We formally introduce our algorithm in Algorithm 2. Prior to comparing our approach with other interpretable AI meth-

ods, we provide a detailed explanation of the key components in Algorithm 2.

**CLIP embedding.** We employ the multi-modal model CLIP (Radford et al., 2021)—a recently introduced large Vision-Language Model—to embed both images and concepts into the same latent space.

**Concept dispersion.** After obtaining the CLIP embeddings, the query features tend to cluster too closely, making the query feature matrix $\boldsymbol{D}^{(0)}$ far from column-orthogonal. To address this, we introduce a *concept dispersion* procedure (see Algorithm 3 in the appendix). This heuristic approach enhances the mutual orthogonality of query features by increasing their relative angles, thereby improving the orthogonality of $\boldsymbol{D}^{(0)}$. Notably, this is achieved while preserving interpretability by maintaining the relative positions of features in the embedded space. Further details on the dispersion step are provided in Appendix D.2.

**Hard-thresholding.** In Equation (4), $\tilde{f}$ is computed in two steps: (1) constructing the sparse code $\mathbf{s}$ by keeping only the entries with the top-$k$ absolute values of $\boldsymbol{D}^\top\mathbf{x}$; and (2) setting $\tilde{f}$ as the inner product of $\mathbf{s}$ with $\mathbf{r} = [r_1, \ldots, r_i]^\top \in \mathbb{R}^n$. Here, we replace the top-$k$ selection with the entry-wise hard-thresholding operator:

$$\text{HT}_\lambda(x) = \begin{cases} x, & \text{if } |x| \geq \lambda, \\ 0, & \text{if } |x| < \lambda. \end{cases}$$

This approach allows for parallelization and reduced computational cost, although the exact sparsity level $k$ can no longer be directly specified; instead, $k$ is determined implicitly by $\lambda$. Nonetheless, adjusting $\lambda$ provides similar control over sparsity.

**Linear Layer.** To compute $\tilde{f}_D$ as defined in Equation (4), access to $\mathbf{r}$ is required, which is unavailable in the image classification setting. To address this, we train a linear classifier that takes the sparse code $\mathbf{s}$ as input and outputs a weight vector over the possible labels. The label with the highest weight is selected as the prediction. The loss function is defined as the cross-entropy between the linear layer's output and the true label $\mathbf{y}$.

**Embedding normalization and projection.** After each iteration, we perform a projection step similar to Equation (5) for each updated concept embedding. Additionally, we normalize each embedding to ensure that it remains on the unit ball (see Algorithm 4 in Appendix D.2).

## 4.1. Performance

We compare the performance of Algorithm 2 against two recently proposed explainable AI methods, CLIP-IP-OMP (Chattopadhyay et al., 2024) and label-free CBM (lf-CBM) (Oikarinen et al., 2023), as well as the CCR baseline without

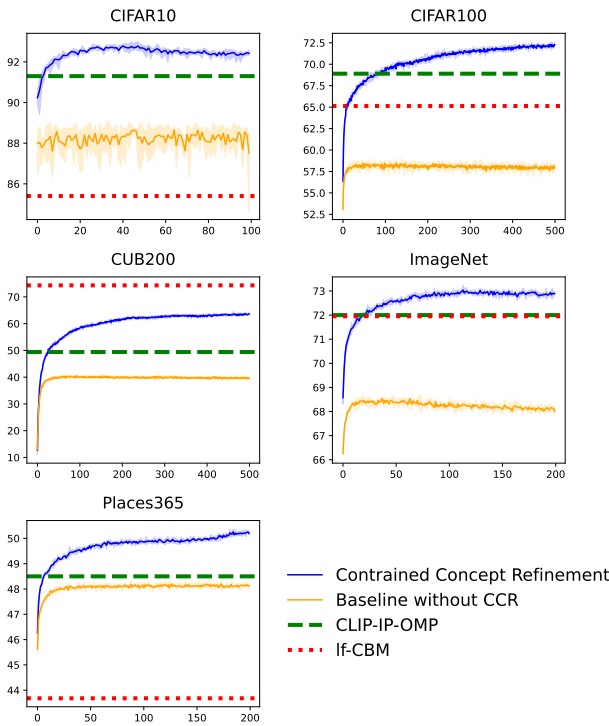

*Figure 2.* Prediction accuracy of CCR and its baseline across iterations, with the final test accuracy of CLIP-IP-OMP and lf-CBM indicated for reference. For CCR and its baseline, we run each experiment for five times and present the average test accuracy at each time step. The shaded area is bounded by the maximum and minimum accuracy obtained over five runs.

*length* or $k$, used in CCR. For ImageNet and Places365, we report the best accuracy achieved by CLIP-IP-OMP across all explanation lengths. Since Lf-CBM does not allow tuning of its explanation length, we directly report its accuracy. The constraint parameter $\rho$ for CCR is fixed at $0.1$ for all experiments.

As shown in Figure 2, CCR consistently outperforms its baseline, CLIP-IP-OMP, and lf-CBM across all benchmarks except CUB-200. This exception is expected, as lf-CBM was built on a ResNet-18 backbone specifically trained on CUB-200, whereas both CLIP-IP-OMP and CCR rely on CLIP as their encoder, which was trained on more diverse and general datasets.

Additionally, we report the average explanation length (AEL, $k$), average sparsity ratio (ASR, $k/n$), and average concept embedding deviation (ACED, $\frac{1}{n}\sum_{i=1}^{n}\|\mathbf{d}_i - \mathbf{d}_i^{(0)}\|_2$) of CCR in the following table:

|  | AEL | ASR | ACED |
|---|---|---|---|
| CIFAR10 | 11.09 | 8.66% | 0.056 |
| CIFAR100 | 19.67 | 2.39% | 0.061 |
| CUB200 | 27.52 | 13.2% | 0.062 |
| ImageNet | 48.97 | 1.08% | 0.097 |
| Places365 | 44.43 | 2.01% | 0.041 |

*Table 1.* Average explanation length (AEL), average sparsity ratio(ASR) and average concept embedding deviation (ACED) for CCR.

the concept refinement step, in the context of explainable image classification.

For the baseline version of CCR, we set $\eta_D = 0$ at Step 8 of Algorithm 2, ensuring that the comparison isolates the effect of concept refinement. The two methods we compare against represent state-of-the-art explainable AI approaches, particularly in terms of scalability. Lf-CBM was the first CBM-type model to be scaled to datasets as large as ImageNet, while CLIP-IP-OMP further improves computational efficiency while maintaining competitive accuracy. For a detailed introduction to both methods, we refer to Appendix A.

The evaluation is conducted across five image classification benchmarks: CIFAR-10, CIFAR-100 (Krizhevsky et al., 2009), ImageNet (Deng et al., 2009), CUB-200 (Wah et al., 2011), and Places365 (Zhou et al., 2017).

To ensure a fair comparison, all methods use the same concept set, which is generated by GPT-3 (Brown et al., 2020). A detailed description of the generation process can be found in (Oikarinen et al., 2023). For the CIFAR-10/100 and CUB-200 datasets, we tune CLIP-IP-OMP to match the average sparsity level of **s**, also referred to as the *explanation*

Thanks to the parallelization enabled by hard thresholding and the computational efficiency of backpropagation, CCR significantly reduces computational costs compared to CLIP-IP-OMP and lf-CBM, particularly when applied to large-scale datasets such as ImageNet ($\approx$ 1.2 million images) and Places365 ($\approx$ 1.8 million images). As reported in (Chattopadhyay et al., 2024), training lf-CBM on ImageNet requires $\approx$ 50 hours on an NVIDIA RTX A5000 GPU. Under the same experimental conditions, CLIP-IP-OMP, with an average explanation length of $\approx$ 50, incurs a computational cost of $\approx$ 40 hours. In our computational environment, using an NVIDIA Tesla V100 GPU, CLIP-IP-OMP remains comparably expensive, requiring $\approx$ 33 hours for $k = 50$. In contrast, CCR (with $k \approx 49$, as shown in Table 1) processes ImageNet in only $\approx$ 2 hours (corresponding to 200 iterations) while achieving even higher accuracy, demonstrating a substantial improvement in computational efficiency. To make CLIP-IP-OMP more computationally feasible, one can reduce $k$; however, for $k = 10$, while the processing time drops to $\approx$ 6 hours, it comes at the cost of a significant accuracy decline to $\approx 63\%$.

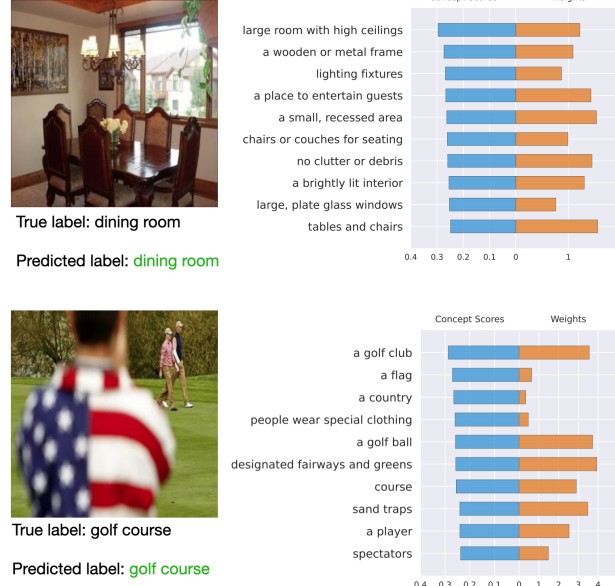

*Figure 3.* The first example illustrates a *simple case* where CCR successfully learns the correct concepts. The second example represents a *misleading case*, where the image contains concepts like "a flag" that, while relevant and visually apparent, could potentially mislead classification. However, CCR effectively extracts both useful and misleading information, assigning appropriate weights to ensure the correct prediction.

### 4.2. Interpretability

To illustrate the interpretability of Algorithm 2, we follow the methodology outlined in (Chattopadhyay et al., 2024) and present the most significant coefficients and weights from the algorithm on different samples from the Places365 dataset. As shown in Figure 3, we highlight the top 10 concepts in **s** with the highest values (left) along with their corresponding weights in the linear layer **L** for the predicted label (right). The selected concepts are semantically relevant to the image, and the linear layer effectively assigns substantial weight to key concepts such as "designated fairways and greens", enabling accurate prediction while appropriately disregarding concepts that, although relevant, may be misleading, such as "a flag" or "a country". For additional case studies on other datasets, we refer the reader to Appendix D.5.

## Impact Statement

This paper introduces Constrained Concept Refinement (CCR), a principled framework that helps bridge the long-standing gap between interpretability and accuracy in machine learning. By constraining the refinement of concept embeddings to lie within a small neighborhood around their initial values, CCR enables interpretable-by-design models to improve prediction performance without compromising their explainability. Our theoretical and empirical results demonstrate the effectiveness of CCR across a variety of tasks and datasets, both in terms of predictive performance and computational efficiency.

The broader impact of this work lies in its potential to advance the practical adoption of interpretable machine learning methods in real-world settings. In particular, the computational efficiency afforded by CCR may facilitate the deployment of explainable artificial intelligence (XAI) techniques in resource-constrained environments.

From an ethical perspective, the capacity to generate explanations that are faithful, stable, and aligned with human-interpretable concepts contributes to addressing critical concerns related to algorithmic accountability and bias. For the proposed method, hyperparameter tuning remains a crucial yet underexplored component for ensuring effective performance in practical applications. This aspect warrants careful consideration to ensure that the resulting outputs are both reliable and justifiable.

## Acknowledgements

This research is supported, in part, by NSF CAREER Award CCF-2337776, NSF Award DMS-2152776, and ONR Award N00014-22-1-2127.

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

# A. Related Literature

In this section, we provide a more comprehensive literature review.

**Explainable AI.** In the main body of the paper, we provided an overview of related methods in explainable AI. Here, we focus on two approaches that are closely related to and comparable with our proposed method. The first is the CLIP-IP-OMP method introduced by (Chattopadhyay et al., 2024), which leverages CLIP embeddings to greedily solve sparse coding problems for each input, subsequently passing the results through a linear layer to generate final predictions. Our method builds upon the same theoretical framework established in (Chattopadhyay et al., 2024). Notably, the baseline model (without concept refinement) considered in Section 3 and Section 4 serves as a differentiable surrogate for CLIP-IP-OMP. The second approach is the label-free CBM (lf-CBM) proposed by (Oikarinen et al., 2023), which employs the inner product between CLIP embeddings to achieve state-of-the-art results in interpretable models. Specifically, lf-CBM was the first CBM model scaled to ImageNet. A detailed comparison of the performance of our method against these two approaches is provided in Section 4.

**Sparse coding and dictionary learning.** Sparse Coding (Olshausen & Field, 1997) refers to the process of representing a given signal (or vector) as a sparse linear combination of fixed signals, collectively referred to as a *dictionary*. Dictionary Learning (Aharon et al., 2006) extends Sparse Coding to a bi-variable setting, allowing the dictionary itself to be optimized. The objective is to identify a dictionary capable of generating effective sparse codes for a specific signal distribution. (Chattopadhyay et al., 2024) establishes a novel connection between sparse coding and explainable AI, treating concept embeddings as the dictionary. Our proposed method directly optimizes concept embeddings, which draws parallels to dictionary learning, although the two approaches address fundamentally different objectives. Among works on dictionary learning, (Mairal et al., 2011) is particularly relevant, as it optimizes dictionaries for downstream tasks. However, our method distinguishes itself through the constrained concept requirement. Interestingly, (Mairal et al., 2011) aligns with our framework as a differentiable learning module, and integrating this approach with our method presents an intriguing avenue for future work. Another promising direction for future research is leveraging the *personalized* dictionary learning framework proposed by (Liang et al., 2024) to extend our approach to Explainable AI in the context of heterogeneous datasets.

**Theoretical guarantees.** To the best of our knowledge, our work is the first to establish a convergence guarantee in the setting where dictionary atoms are updated for downstream tasks. The most closely related work is (Chattopadhyay et al., 2024), which introduced IP-OMP. However, the connection between IP-OMP and column-orthogonal matching pursuit lacks rigor due to the presence of normalization terms (see Theorem 2.5), making it difficult to prove that IP-OMP minimizes the loss $\mathcal{L}$ or $\mathcal{L}_m$ using the optimality conditions of OMP (Tropp, 2004). Another related approach is task-driven dictionary learning (Mairal et al., 2011), which establishes the differentiability of the objective function but does not provide a convergence guarantee.

# B. Further Discussion

In this section, we provide further discussion on Theorem 3.3 and Theorem 3.4.

**Tolerance, Sample Size, and Convergence Speed.** Both theorems establish that the population loss converges to zero, provided that the initial dictionary estimate is within $\frac{1}{\sqrt{k}}$ in the column-wise $\|\cdot\|_{1,2}$-norm from the ground-truth dictionary $\boldsymbol{D}^*$. Assuming that a constant fraction of the queries in $\boldsymbol{D}^*$ is utilized to generate each $\mathbf{x}^h$, i.e., $k = \Omega(n)$, Theorem 3.4 indicates that each column of $\boldsymbol{D}^*$ can be accurately recovered with a sample size of $m = O(n)$ and a convergence rate of $\alpha = 1 - \Omega(1)$. Notably, the linear sample size and linear convergence rate is consistent with the best-known results in dictionary learning (Arora et al., 2015; Liang et al., 2022).

**Sparsity Level.** It is well established in the dictionary learning literature that higher sparsity levels in ground-truth sparse codes (i.e., larger values of $k$ in Assumption 3.2) present greater challenges for recovery; see (Arora et al., 2015). Indeed, (Arora et al., 2015) asserts that sparsity levels beyond $k = \Omega\left(\frac{n}{\log n}\right)$ rarely succeed in practice, even though several approaches have been proposed to handle cases where $k = \Omega(n)$ (Sun et al., 2016; Liang et al., 2022). In contrast, our results reveal that the sparsity level $k$ plays a nuanced role when optimizing $\boldsymbol{D}$ for the downstream task $\mathcal{L}$. On the one hand, a larger $k$ imposes a more stringent initial error bound for both theorems. On the other hand, a larger $k$ in Theorem 3.4 improves the convergence rate $\tau$ and reduces the required sample size $m$. The former phenomenon is consistent with

findings in dictionary learning literature, where exact support recovery becomes more feasible for sparser generative models. A high-level explanation for the latter is that a denser generative model for $\beta$ provides each $\mathbf{x}$ with more information, thereby expediting the training process.

## C. Proof of Theorems

### C.1. Proof of Theorem 2.6

Our proof strategy follows two main steps:

- **Step 1:** We prove that $\mathcal{L}(f_{\boldsymbol{D}^*}) = 0$. To do so, we first derive an explicit form of $f_{\boldsymbol{D}^*}$, and then show $\mathcal{L}(f_{\boldsymbol{D}^*}) = 0$ for this explicit form.

- **Step 2:** We explicitly design a column-orthogonal $\tilde{\boldsymbol{D}}$ such that $\|\tilde{\boldsymbol{D}} - \boldsymbol{D}^*\|_{1,2} \leq \epsilon$ and $\mathcal{L}(f_{\tilde{\boldsymbol{D}}}) \geq \frac{2(k-1)\epsilon^2\gamma^2}{5}$.

To streamline the presentation, we denote the $i$-th column of $\boldsymbol{D}^*$ as $\mathbf{d}_i^*$. Note that $\mathbf{d}_i^*$ is identical to $\mathbf{v}_i$ used in the paper.

• **Step 1: Establishing $\mathcal{L}(f_{\boldsymbol{D}^*}) = 0$.** We first provide an explicit characterization of $f_{\boldsymbol{D}^*}$.

**Lemma C.1.** *We have*

$$f_{\boldsymbol{D}^*}\left(\mathbf{x}, \{r_{\pi(i)}\}_{i=1}^k\right) = \sum_{i \in S} \langle \mathbf{d}_i^*, \mathbf{x} \rangle r_i,$$

*where $S = \{\pi(1), \pi(2), ..., \pi(k)\}$ is the index set selected by IP-OMP applied to $\mathbf{x}$ and $\boldsymbol{D}^*$.*

Our next goal is to provide an explicit characterization of $S$, introduced in the above lemma.

**Lemma C.2.** *For any column-orthogonal $\boldsymbol{D}$, the index set $S = \{\pi(1), \pi(2), ..., \pi(k)\}$ selected by IP-OMP applied to $\mathbf{x}$ and $\boldsymbol{D}$ corresponds to the indices of the top-$k$ largest values of $\{|\langle \mathbf{d}_i, \mathbf{x} \rangle|\}_{i=1}^n$:*

$$S = \underset{T \subseteq [n], |T| \leq k}{\arg\max} \sum_{i \in T} |\langle \mathbf{d}_i, \mathbf{x} \rangle|.$$

*Moreover, if $\boldsymbol{D} = \boldsymbol{D}^*$, we have $S = S^*$, where $S^*$ is the support of $\beta$ defined in Theorem 2.6.*

We defer the proofs of Lemma C.1 and Lemma C.2 to Appendix C.4.2 and Appendix C.4.3, respectively. Combining the above two lemmas, we obtain:

$$f_{\boldsymbol{D}^*}\left(\mathbf{x}, \{r_{\pi(i)}\}_{i=1}^k\right) = \sum_{i \in S} \langle \mathbf{d}_i^*, \mathbf{x} \rangle r_i, \text{ where } S = \underset{T \subseteq [n], |T| \leq k}{\arg\max} \sum_{i \in T} |\langle \mathbf{d}_i^*, \mathbf{x} \rangle|$$

Given this explicit form of $f_{\boldsymbol{D}^*}$ generated by IP-OMP, we are now ready to investigate the population loss $\mathcal{L}$ of that estimator.

For any column-column-orthogonal $\boldsymbol{D}$, one can write

$$
\begin{aligned}
\mathcal{L}\left(f_{\boldsymbol{D}}\right) &= \mathbb{E}_{\mathbf{z} \sim \mathcal{N}(\mathbf{0}, \boldsymbol{I}_{d \times d})}\left[\left(\sum_{i \in S} \langle \mathbf{d}_i, \mathbf{x} \rangle r_i - \langle \mathbf{x}, \mathbf{z} \rangle\right)^2\right] \\
&= \mathbb{E}_{\mathbf{z} \sim \mathcal{N}(\mathbf{0}, \boldsymbol{I}_{d \times d})}\left[\left(\sum_{i \in S} \langle \mathbf{d}_i, \mathbf{x} \rangle \langle \mathbf{d}_i^*, \mathbf{z} \rangle - \langle \mathbf{x}, \mathbf{z} \rangle\right)^2\right] \\
&= \mathbb{E}_{\mathbf{z} \sim \mathcal{N}(\mathbf{0}, \boldsymbol{I}_{d \times d})}\left[\left(\sum_{i \in S} \langle \mathbf{d}_i^* \mathbf{d}_i^\top \mathbf{x}, \mathbf{z} \rangle - \langle \mathbf{x}, \mathbf{z} \rangle\right)^2\right] \\
&= \mathbb{E}_{\mathbf{z} \sim \mathcal{N}(\mathbf{0}, \boldsymbol{I}_{d \times d})}\left[\left(\langle \boldsymbol{D}_S^* \boldsymbol{D}_S^\top \mathbf{x}, \mathbf{z} \rangle - \langle \mathbf{x}, \mathbf{z} \rangle\right)^2\right] \\
&= \mathbb{E}_{\mathbf{z} \sim \mathcal{N}(\mathbf{0}, \boldsymbol{I}_{d \times d})}\left[\left(\langle \boldsymbol{D}_S^* \boldsymbol{D}_S^\top \mathbf{x} - \mathbf{x}, \mathbf{z} \rangle\right)^2\right]
\end{aligned}
\tag{7}
$$

It is easy to verify that random variable $\langle \boldsymbol{D}_S^* \boldsymbol{D}_S^\top \mathbf{x} - \mathbf{x}, \mathbf{z} \rangle$ is distributed as $\mathcal{N}(0, \|\boldsymbol{D}_S^* \boldsymbol{D}_S^\top \mathbf{x} - \mathbf{x}\|_2^2)$. As a result, we can conclude that

$$\mathcal{L}(f_{\boldsymbol{D}}) = \|\boldsymbol{D}_S^* \boldsymbol{D}_S^\top \mathbf{x} - \mathbf{x}\|_2^2 \quad \text{for any column-orthogonal } \boldsymbol{D}. \tag{8}$$

Given Lemma C.2, we have $S = S^*$. We can subsequently conclude that:

$$\boldsymbol{D}_S^* \boldsymbol{D}_S^{*\top} \mathbf{x} = \boldsymbol{D}_{S^*}^* \boldsymbol{D}_{S^*}^{*\top} \mathbf{x} = \boldsymbol{D}_{S^*}^* \boldsymbol{D}_{S^*}^{*\top} \boldsymbol{D}_{S^*}^* \boldsymbol{\beta}_{S^*} = \boldsymbol{D}_{S^*}^* \boldsymbol{\beta}_{S^*} = \mathbf{x},$$

This leads to $\mathcal{L}(f_{\boldsymbol{D}^*}) = \|\boldsymbol{D}_S^* \boldsymbol{D}_S^{*\top} \mathbf{x} - \mathbf{x}\|_2^2 = 0$, thereby completing the proof of our first step.

• **Step 2: Constructing $\tilde{\boldsymbol{D}}$.** Without loss of generality, let us assume $S^* = [k]$, where $k$ to be even. The case where $k$ is odd is easily proved by following the argument below and replacing $k$ with $k-1$. Consider the following explicit form for $\tilde{\boldsymbol{D}}$:

$$\begin{cases} \tilde{\mathbf{d}}_i = \cos\theta \mathbf{d}_i^* + \sin\theta \mathbf{d}_{i+k/2}^*, & 1 \le i \le k/2, \\ \tilde{\mathbf{d}}_{i+k/2} = -\sin\theta \mathbf{d}_i^* + \cos\theta \mathbf{d}_{i+k/2}^* & 1 \le i \le k/2, \\ \tilde{\mathbf{d}}_i = \mathbf{d}_i^* & i > k, \end{cases}$$

where $\theta$ is an angle to be determined later.

Intuitively, $\tilde{\boldsymbol{D}}$ is constructed by iteratively selecting a pair of columns in $\boldsymbol{D}_S^*$ and rotating them by an angle $\theta > 0$ within the two-dimensional subspace spanned by each pair. Based on its definition, we can establish the following properties for $\tilde{\boldsymbol{D}}$:

**Lemma C.3.** $\tilde{\boldsymbol{D}}$ *has the following properties:*

- $\tilde{\boldsymbol{D}}$ *is column-orthogonal.*

- $\|\tilde{\boldsymbol{D}} - \boldsymbol{D}^*\|_{1,2} = 2\sin\frac{\theta}{2}$.

The proof of Lemma C.3 is presented in Appendix C.4.4

Our next step is to calculate $\mathcal{L}(f_{\tilde{\boldsymbol{D}}})$. Note that $\tilde{\boldsymbol{D}}$ is column-orthogonal according to Lemma C.3. Therefore, according to Equation (8), we have $\mathcal{L}(f_{\tilde{\boldsymbol{D}}}) = \|\boldsymbol{D}_S^* \tilde{\boldsymbol{D}}_S^\top \mathbf{x} - \mathbf{x}\|_2^2$. According to Lemma C.2, in order to obtain the index set $S$ selected by IP-OMP applied to $\mathbf{x}$ and $\tilde{\boldsymbol{D}}$, we need to find the indices of the top-$k$ largest values of $|\langle \tilde{\mathbf{d}}_i, \mathbf{x} \rangle|$. For $1 \le i \le k/2$, we have:

$$|\langle \tilde{\mathbf{d}}_i, \mathbf{x} \rangle| = \left| \left\langle \cos\theta \mathbf{d}_i^* + \sin\theta \mathbf{d}_{i+k/2}^*, \sum_{j=1}^k \beta_j \mathbf{d}_j^* \right\rangle \right|$$
$$= |\cos\theta \beta_i + \sin\theta \beta_{i+k/2}|$$

Upon choosing $\theta$ such that $\tan\theta < \gamma/\Gamma$, we have

$$|\cos\theta \beta_i| \ge \left| \frac{\sin\theta \gamma}{\tan\theta} \right| > |\sin\theta \Gamma| \ge |\sin\theta \beta_{i+k/2}|,$$

which guarantees that $|\langle \tilde{\mathbf{d}}_i, \mathbf{x} \rangle| > 0$. Similarly, for $1 + k/2 \le i \le k$, we have:

$$|\langle \tilde{\mathbf{d}}_i, \mathbf{x} \rangle| = \left| \left\langle \cos\theta \mathbf{d}_i^* - \sin\theta \mathbf{d}_{i-k/2}^*, \sum_{j=1}^k \beta_j \mathbf{d}_j^* \right\rangle \right|$$
$$= |\cos\theta \beta_i - \sin\theta \beta_{i-k/2}|$$
$$> 0.$$

For $i > k$, we have $|\langle \tilde{\mathbf{d}}_i, \mathbf{x} \rangle| = |\langle \mathbf{d}_i^*, \mathbf{x} \rangle| = 0$. As a result, when $\tan \theta < \gamma / \Gamma$, Lemma C.2 guarantees that the index set selected by IP-OMP is $S = [k] = S^*$. Given this fact, we can calculate $\mathcal{L}(f_{\tilde{\mathbf{D}}})$ as:

$$
\begin{aligned}
\mathcal{L}(f_{\tilde{\mathbf{D}}}) &= \|\mathbf{D}_S^* \tilde{\mathbf{D}}_S^\top \mathbf{x} - \mathbf{x}\|_2^2 \\
&= \|\mathbf{D}_{S^*}^* \tilde{\mathbf{D}}_{S^*}^\top \mathbf{x} - \mathbf{x}\|_2^2 \\
&= \|\mathbf{D}_{S^*}^* \tilde{\mathbf{D}}_{S^*}^\top \sum_{i=1}^k \beta_i \mathbf{d}_i^* - \sum_{i=1}^k \beta_i \mathbf{d}_i^*\|_2^2 \\
&\overset{(a)}{=} \left\| \sum_{i=1}^{k/2} \left( \cos\theta \beta_i + \sin\theta \beta_{i+k/2} \right) \mathbf{d}_i^* + \sum_{i=1+k/2}^k \left( \cos\theta \beta_i - \sin\theta \beta_{i-k/2} \right) \mathbf{d}_i^* - \sum_{i=1}^k \beta_i \mathbf{d}_i^* \right\|_2^2 \\
&= \left\| \sum_{i=1}^{k/2} \left( (\cos\theta - 1)\beta_i + \sin\theta \beta_{i+k/2} \right) \mathbf{d}_i^* + \sum_{i=1+k/2}^k \left( (\cos\theta - 1)\beta_i - \sin\theta \beta_{i-k/2} \right) \mathbf{d}_i^* \right\|_2^2 \\
&\overset{(b)}{=} \sum_{i=1}^{k/2} \left( (\cos\theta - 1)\beta_i + \sin\theta \beta_{i+k/2} \right)^2 + \sum_{i=1+k/2}^k \left( (\cos\theta - 1)\beta_i - \sin\theta \beta_{i-k/2} \right)^2,
\end{aligned}
\tag{9}
$$

where (a) is based on the construction of $\tilde{\mathbf{D}}$, and (b) is due to the assumption that $\mathbf{D}^*$ is column-orthogonal. When $\sin\frac{\theta}{2} \leq \frac{1}{\sqrt{1+16\Gamma^2/\gamma^2}}$, we have for $1 \leq i \leq k/2$:

$$
\begin{aligned}
\sin\theta &= 2\sin\frac{\theta}{2}\cos\frac{\theta}{2} \\
&= 2\sin\frac{\theta}{2}\sqrt{1 - \sin^2\frac{\theta}{2}} \\
&\geq 2\sin\frac{\theta}{2} \cdot \frac{4\Gamma}{\gamma}\frac{1}{\sqrt{1+16\Gamma^2/\gamma^2}} \\
&\geq \frac{8\Gamma}{\gamma}\sin^2\frac{\theta}{2} \\
&= \frac{4\Gamma}{\gamma}(1 - \cos\theta),
\end{aligned}
$$

which leads to

$$
\begin{aligned}
&\left( (\cos\theta - 1)\beta_i + \sin\theta \beta_{i+k/2} \right)^2 \\
&= (\cos\theta - 1)^2 \beta_i^2 + 2(\cos\theta - 1)\sin\theta \beta_i \beta_{i+k/2} + \sin^2\theta \beta_{i+k/2}^2 \\
&\geq 2(\cos\theta - 1)\sin\theta \beta_i \beta_{i+k/2} + \sin^2\theta \beta_{i+k/2}^2 \\
&\geq -\left| \frac{\gamma}{2\Gamma}\sin^2\theta \beta_i \beta_{i+k/2} \right| + \sin^2\theta \beta_{i+k/2}^2 \\
&\geq \frac{1}{2}\sin^2\theta \beta_{i+k/2}^2.
\end{aligned}
$$

With identical arguments, we have for $1 + k/2 \leq i \leq k$:

$$
\left( (\cos\theta - 1)\beta_i + \sin\theta \beta_{i-k/2} \right)^2 \geq \frac{1}{2}\sin^2\theta \beta_{i-k/2}^2.
$$

As a result, Equation (9) reduces to:

$$
\begin{aligned}
\mathcal{L}(f_{\tilde{\mathbf{D}}}) &\geq \sum_{i=1}^{k/2} \frac{1}{2}\sin^2\theta \beta_{i+k/2}^2 + \sum_{i=1+k/2}^k \frac{1}{2}\sin^2\theta \beta_{i-k/2}^2 \\
&\geq \frac{k}{2}\sin^2\theta \gamma^2
\end{aligned}
$$

Finally, we need to perform a change of variable by setting $\epsilon = 2\sin\frac{\theta}{2}$. Elementary calculation gives $\sin\theta = \frac{\epsilon\sqrt{4-\epsilon^2}}{2}$ and $\tan\theta = \frac{\epsilon\sqrt{4-\epsilon^2}}{2-\epsilon^2}$. When $\epsilon < \frac{1}{2}$, we have $\sin\theta \geq 9\epsilon/10$ and $\tan\theta \leq 6\epsilon/5$. Recall that our proof requires $\tan\theta < \gamma/\Gamma$ and $\sin\frac{\theta}{2} \leq \frac{1}{\sqrt{1+16\Gamma^2/\gamma^2}}$. As a result, when $\epsilon = 2\sin\frac{\theta}{2} \leq \min\left(\frac{1}{2}, \frac{5\gamma}{6\Gamma}, \frac{1}{\sqrt{1+16\Gamma^2/\gamma^2}}\right) = \frac{1}{\sqrt{1+16\Gamma^2/\gamma^2}}$, we have

$$\mathcal{L}(f_{\tilde{\boldsymbol{D}}}) - \mathcal{L}(f_{\boldsymbol{D}^*}) = \mathcal{L}(f_{\tilde{\boldsymbol{D}}}) \geq \frac{81k\epsilon^2\gamma^2}{200}.$$

As we mentioned, for an odd value of $k$, an analogous argument can be made to arrive at a similar bound of

$$\mathcal{L}(f_{\tilde{\boldsymbol{D}}}) - \mathcal{L}(f_{\boldsymbol{D}^*}) = \mathcal{L}(f_{\tilde{\boldsymbol{D}}}) \geq \frac{81(k-1)\epsilon^2\gamma^2}{200},$$

This completes the proof of Theorem 2.6. $\qquad\square$

## C.2. Proof of Theorem 3.3

To prove the convergence of $\mathcal{L}\left(\tilde{f}_{\boldsymbol{D}^{(t)}}\right)$, we use an inductive approach where at each iteration, we will subsequently prove: (1) $\tilde{f}$ recovers the exact support $S^*$; (2) gradient descent will make progress towards one of the optimal minimizers which will be explicitly defined later; (3) $\mathcal{L}\left(\tilde{f}_{\boldsymbol{D}^{(t)}}\right)$ will decrease linearly.

Recall that $\boldsymbol{D}^{(t)} = \boldsymbol{D} + \Delta\boldsymbol{D}^{(t)}$. The equations below rewrite Equation (5) for each column $i \in [n]$ of $\boldsymbol{D}^{(t)}$.

$$\mathbf{d}_i^{(t+0.5)} = \mathbf{d}_i^{(t)} - \eta\frac{\partial\mathcal{L}(\tilde{f}_{\boldsymbol{D}^{(t)}})}{\partial\mathbf{d}_i^{(t)}}, \tag{10}$$

$$\mathbf{d}_i^{(t+1)} = \underset{\mathbf{d}:\|\mathbf{d}-\mathbf{d}_i^{(0)}\|_2\leq\rho}{\arg\min} \|\mathbf{d} - \mathbf{d}_i^{(t+0.5)}\|_2. \tag{11}$$

It is easy to verify that the above update rules are equivalent to Equation (5). We will use induction to prove this theorem. In particular, define

$$A(t): \quad \mathcal{L}(\tilde{f}_{\boldsymbol{D}^{(t)}}) \leq \tau^2\mathcal{L}(\tilde{f}_{\boldsymbol{D}^{(t-1)}}), \text{where } \tau = 1 - 2\eta\|\mathbf{x}\|_2^2.$$

To prove this we define an auxiliary feature vector which will play an important role in our arguments:

$$\widehat{\mathbf{d}}_i = \mathbf{d}_i^{(0)} + \frac{\beta_i - \mathbf{x}^\top\mathbf{d}_i^{(0)}}{\|\mathbf{x}\|_2^2}\mathbf{x}, \qquad \forall i \in [n]. \tag{12}$$

Based on this auxiliary feature vector, we define the following event:

$$B(t): \quad \widehat{\mathbf{d}}_i - \mathbf{d}_i^{(t)} = \frac{\beta_i - \mathbf{x}^\top\mathbf{d}_i^{(t)}}{\|\mathbf{x}\|_2^2}\mathbf{x}, \qquad \forall i \in [n].$$

Now, our proof strategy is to show that, for $t = 0, 1, \ldots$:

- $B(t)$ implies $B(t+1)$,

- $B(t)$ implies $A(t+1)$.

This two statements combined will establish the correctness of Theorem 3.3 Indeed, the base case $A(0)$ is trivially satisfied due to Equation (12).

● **Establishing** $B(t) \implies B(t+1)$**.** To begin with, we use the following lemma to show that $\tilde{f}$ is able to find the correct support when $\boldsymbol{D}^{(t)}$ is close enough to $\boldsymbol{D}$:

**Lemma C.4.** *For any $\boldsymbol{D}$ such that $\|\boldsymbol{D} - \boldsymbol{D}^*\|_{1,2} \leq 2\rho < \frac{\gamma}{4\sqrt{k}\Gamma}$,we have*

$$S = \underset{T\subseteq[n],|T|\leq k}{\arg\max} \sum_{i\in T}|\langle\mathbf{d}_i, \mathbf{x}\rangle| = S^*.$$

We defer the proof of Lemma C.4 to Appendix C.4.5. To invoke Lemma C.4, we have

$$\|\boldsymbol{D}^{(t)} - \boldsymbol{D}^*\|_{1,2} \le \|\boldsymbol{D}^{(t)} - \boldsymbol{D}^{(0)}\|_{1,2} + \|\boldsymbol{D}^{(0)} - \boldsymbol{D}^*\|_{1,2} \le \rho + \rho = 2\rho.$$

The bound on $\|\boldsymbol{D}^{(t)} - \boldsymbol{D}^{(0)}\|_{1,2}$ follows from the projection step in Equation (11) while the bound on $\|\boldsymbol{D}^{(0)} - \boldsymbol{D}^*\|_{1,2}$ follows from the initial error bound. We can then conclude that $S = S^*$ at every iteration $t$, which means that Equation (10) and Equation (11) will only change $i \in S^*$, while for other $i \notin S^*$, we have $\mathbf{d}_i^{(t+1)} = \mathbf{d}_i^{(t)}$. This immediately implies that $B(t)$ is trivially satisfied for $\forall i \notin S^*$ and $\forall t$. Now, the goal is to show that $B(t) \implies B(t+1)$ for $\forall i \in S^*$.

Based on Equation (8), we have

$$\mathcal{L}(\tilde{f}_{\boldsymbol{D}^{(t)}}) = \|\boldsymbol{D}_{S^*}^* \boldsymbol{D}_{S^*}^{(t)\top} \mathbf{x} - \mathbf{x}\|_2^2,$$

Subsequently, the gradient of $\mathcal{L}$ at $\boldsymbol{D}_{S^*}^{(t)}$ can be written as:

$$\begin{aligned}
\frac{\partial \mathcal{L}(\tilde{f}_{\boldsymbol{D}^{(t)}})}{\partial \boldsymbol{D}_{S^*}^{(t)}} &= 2\mathbf{x}\left(\boldsymbol{D}_{S^*}^* \boldsymbol{D}_{S^*}^{(t)\top} \mathbf{x} - \mathbf{x}\right)^\top \boldsymbol{D}_{S^*}^* \\
&= 2\mathbf{x}\left(\boldsymbol{D}_{S^*}^* \boldsymbol{D}_{S^*}^{(t)\top} \mathbf{x} - \boldsymbol{D}_{S^*}^* \boldsymbol{\beta}_{S^*}\right)^\top \boldsymbol{D}_{S^*}^* \\
&= 2\mathbf{x}\left(\boldsymbol{D}_{S^*}^{(t)\top} \mathbf{x} - \boldsymbol{\beta}_{S^*}\right)^\top \boldsymbol{D}_{S^*}^{*\top} \boldsymbol{D}_{S^*}^* \\
&= 2\mathbf{x}\left(\boldsymbol{D}_{S^*}^{(t)\top} \mathbf{x} - \boldsymbol{\beta}_{S^*}\right)^\top.
\end{aligned}$$

For $i \in S^*$, we have

$$\frac{\partial \mathcal{L}(\tilde{f}_{\boldsymbol{D}^{(t)}})}{\partial \mathbf{d}_i^{(t)}} = 2(\mathbf{d}_i^{(t)\top} \mathbf{x} - \beta_i)\mathbf{x}.$$

This implies that:

$$\begin{aligned}
\widehat{\mathbf{d}}_i - \mathbf{d}_i^{(t+0.5)} &= \widehat{\mathbf{d}}_i - \mathbf{d}_i^{(t)} + 2\eta(\mathbf{d}_i^{(t)\top}\mathbf{x} - \beta_i)\mathbf{x} \\
&\stackrel{A(t)}{=} \frac{\beta_i - \mathbf{x}^\top \mathbf{d}_i^{(t)}}{\|\mathbf{x}\|_2^2}\mathbf{x} + 2\eta(\mathbf{d}_i^{(t)\top}\mathbf{x} - \beta_i)\mathbf{x} \\
&= \frac{\beta_i - \mathbf{x}^\top\left(\mathbf{d}_i^{(t+0.5)} + 2\eta(\mathbf{d}_i^{(t)\top}\mathbf{x} - \beta_i)\mathbf{x}\right)}{\|\mathbf{x}\|_2^2}\mathbf{x} + 2\eta(\mathbf{d}_i^{(t)\top}\mathbf{x} - \beta_i)\mathbf{x} \\
&= \frac{\beta_i - \mathbf{x}^\top \mathbf{d}_i^{(t+0.5)}}{\|\mathbf{x}\|_2^2}\mathbf{x} - 2\eta(\mathbf{d}_i^{(t)\top}\mathbf{x} - \beta_i)\mathbf{x} + 2\eta(\mathbf{d}_i^{(t)\top}\mathbf{x} - \beta_i)\mathbf{x} \\
&= \frac{\beta_i - \mathbf{x}^\top \mathbf{d}_i^{(t+0.5)}}{\|\mathbf{x}\|_2^2}\mathbf{x}.
\end{aligned}$$

In other words, $B(t)$ implies $B(t+0.5)$. Suppose $\|\mathbf{d}_i^{(t+0.5)} - \mathbf{d}_i^{(0)}\|_2 \le \rho$, then we have $\mathbf{d}_i^{(t+1)} = \mathbf{d}_i^{(t+0.5)}$, which readily implies $B(t+1)$. On the other hand, if $\|\mathbf{d}_i^{(t+0.5)} - \mathbf{d}_i^{(0)}\|_2 > \rho$, we have:

$$\begin{aligned}
\mathbf{d}_i^{(t+0.5)} - \mathbf{d}_i^{(0)} &= \left(\mathbf{d}_i^{(t+0.5)} - \widehat{\mathbf{d}}_i\right) - \left(\mathbf{d}_i^{(0)} - \widehat{\mathbf{d}}_i\right) \\
&\stackrel{A(t+0.5),A(0)}{=} -\frac{\beta_i - \mathbf{x}^\top \mathbf{d}_i^{(t+0.5)}}{\|\mathbf{x}\|_2^2}\mathbf{x} + \frac{\beta_i - \mathbf{x}^\top \mathbf{d}_i^{(0)}}{\|\mathbf{x}\|_2^2}\mathbf{x} \\
&= \frac{\mathbf{x}^\top \mathbf{d}_i^{(t+0.5)} - \mathbf{x}^\top \mathbf{d}_i^{(0)}}{\|\mathbf{x}\|_2^2}\mathbf{x},
\end{aligned} \tag{13}$$

Therefore, when $\|\mathbf{d}_i^{(t+0.5)} - \mathbf{d}_i^{(0)}\|_2 > \rho$, we have:

$$
\begin{aligned}
\mathbf{d}_i^{(t+1)} &= \underset{\mathbf{d}:\|\mathbf{d}-\mathbf{d}_i^{(0)}\|_2 \leq \rho}{\arg\min} \|\mathbf{d} - \mathbf{d}_i^{(t+0.5)}\|_2 \\
&= \mathbf{d}_i^{(0)} + \rho \frac{\mathbf{d}_i^{(t+0.5)} - \mathbf{d}_i^{(0)}}{\|\mathbf{d}_i^{(t+0.5)} - \mathbf{d}_i^{(0)}\|_2} \\
&= \mathbf{d}_i^{(0)} + \rho \operatorname{sign}\left(\mathbf{x}^\top \mathbf{d}_i^{(t+0.5)} - \mathbf{x}^\top \mathbf{d}_i^{(0)}\right) \frac{\mathbf{x}}{\|\mathbf{x}\|_2}.
\end{aligned}
\tag{14}
$$

This in turn implies

$$
\begin{aligned}
&\widehat{\mathbf{d}}_i - \mathbf{d}_i^{(t+1)} \\
&= \widehat{\mathbf{d}}_i - \mathbf{d}_i^{(0)} - \rho \operatorname{sign}\left(\mathbf{x}^\top \mathbf{d}_i^{(t+0.5)} - \mathbf{x}^\top \mathbf{d}_i^{(0)}\right) \frac{\mathbf{x}}{\|\mathbf{x}\|_2} \\
&\overset{A(0)}{=} \frac{\beta_i - \mathbf{x}^\top \mathbf{d}_i^{(0)}}{\|\mathbf{x}\|_2^2} \mathbf{x} - \rho \operatorname{sign}\left(\mathbf{x}^\top \mathbf{d}_i^{(t+0.5)} - \mathbf{x}^\top \mathbf{d}_i^{(0)}\right) \frac{\mathbf{x}}{\|\mathbf{x}\|_2} \\
&= \frac{\beta_i - \mathbf{x}^\top \left(\mathbf{d}_i^{(t+1)} - \rho \operatorname{sign}\left(\mathbf{x}^\top \mathbf{d}_i^{(t+0.5)} - \mathbf{x}^\top \mathbf{d}_i^{(0)}\right) \frac{\mathbf{x}}{\|\mathbf{x}\|_2}\right)}{\|\mathbf{x}\|_2^2} \mathbf{x} - \rho \operatorname{sign}\left(\mathbf{x}^\top \mathbf{d}_i^{(t+0.5)} - \mathbf{x}^\top \mathbf{d}_i^{(0)}\right) \frac{\mathbf{x}}{\|\mathbf{x}\|_2} \\
&= \frac{\beta_i - \mathbf{x}^\top \mathbf{d}_i^{(t+1)}}{\|\mathbf{x}\|_2^2} \mathbf{x},
\end{aligned}
$$

establishing $B(t+1)$, as desired.

• **Establishing** $B(t) \implies A(t+1)$. First, we establish $\|\widehat{\mathbf{d}}_i - \mathbf{d}_i^{(t+1)}\|_2 \leq \tau \|\widehat{\mathbf{d}}_i - \mathbf{d}_i^{(t)}\|_2$. We do so in two steps. First, we prove that $\|\widehat{\mathbf{d}}_i - \mathbf{d}_i^{(t+0.5)}\|_2 \leq \tau \|\widehat{\mathbf{d}}_i - \mathbf{d}_i^{(t)}\|_2$. Then, we show that $\|\widehat{\mathbf{d}}_i - \mathbf{d}_i^{(t+1)}\|_2 \leq \|\widehat{\mathbf{d}}_i - \mathbf{d}_i^{(t+0.5)}\|_2$.

To establish $\|\widehat{\mathbf{d}}_i - \mathbf{d}_i^{(t+1)}\|_2 \leq \tau \|\widehat{\mathbf{d}}_i - \mathbf{d}_i^{(t)}\|_2$, one can write:

$$
\begin{aligned}
\|\widehat{\mathbf{d}}_i - \mathbf{d}_i^{(t+0.5)}\|_2 &= \|\widehat{\mathbf{d}}_i - \mathbf{d}_i^{(t)} + 2\eta(\mathbf{d}_i^{(t)\top}\mathbf{x} - \beta_i)\mathbf{x}\|_2 \\
&\overset{A(t)}{=} \left\|\widehat{\mathbf{d}}_i - \mathbf{d}_i^{(t)} - 2\eta\|\mathbf{x}\|_2^2\left(\widehat{\mathbf{d}}_i - \mathbf{d}_i^{(t)}\right)\right\|_2 \\
&\leq \left|1 - 2\eta\|\mathbf{x}\|_2^2\right| \|\widehat{\mathbf{d}}_i - \mathbf{d}_i^{(t)}\|_2 \\
&= \tau \|\widehat{\mathbf{d}}_i - \mathbf{d}_i^{(t)}\|_2.
\end{aligned}
$$

Next, we show that $\|\widehat{\mathbf{d}}_i - \mathbf{d}_i^{(t+1)}\|_2 \leq \|\widehat{\mathbf{d}}_i - \mathbf{d}_i^{(t+0.5)}\|_2$. To do so, we need the following elementary lemma:

**Lemma C.5.** *Given a fixed vector* $\mathbf{v}$ *and* $\mathbf{a} = a\mathbf{v}$, $\mathbf{b} = b\mathbf{v}$, $\mathbf{c} = c\mathbf{v}$, *where* $\mathbf{a}$, $\mathbf{b}$ *and* $\mathbf{c}$ *are vectors and* $a$, $b$, $c$ *are scalars, if* $|c| \geq |b| \geq a$ *and* $bc > 0$, *we have* $\|\mathbf{c} - \mathbf{a}\|_2 \geq \|\mathbf{b} - \mathbf{a}\|_2$.

We will prove Lemma C.5 in Appendix C.4.6. Now we invoke Lemma C.5 with:

$$
\begin{aligned}
\mathbf{v} &= \frac{\mathbf{x}}{\|\mathbf{x}\|_2}, \\
a &= \frac{\beta_i - \mathbf{x}^\top \mathbf{d}_i^{(0)}}{\|\mathbf{x}\|_2}, \\
b &= \rho \operatorname{sign}\left(\mathbf{x}^\top \mathbf{d}_i^{(t+0.5)} - \mathbf{x}^\top \mathbf{d}_i^{(0)}\right), \\
c &= \frac{\mathbf{x}^\top \mathbf{d}_i^{(t+0.5)} - \mathbf{x}^\top \mathbf{d}_i^{(0)}}{\|\mathbf{x}\|_2}.
\end{aligned}
$$

Here $bc > 0$ is given by their definition. Moreover $|c| \geq |b|$ is established as:

$$
\begin{aligned}
|c| &= \left| \frac{\mathbf{x}^\top \mathbf{d}_i^{(t+0.5)} - \mathbf{x}^\top \mathbf{d}_i^{(0)}}{\|\mathbf{x}\|_2} \right| \\
&= \left\| \frac{\mathbf{x}^\top \mathbf{d}_i^{(t+0.5)} - \mathbf{x}^\top \mathbf{d}_i^{(0)}}{\|\mathbf{x}\|_2^2} \mathbf{x} \right\|_2 \\
&\overset{(13)}{=} \|\mathbf{d}_i^{(t+0.5)} - \mathbf{d}_i^{(0)}\|_2 \\
&\geq \rho \\
&= |b|.
\end{aligned}
$$

Finally, $|b| > a$ is given by:

$$
\begin{aligned}
a &= \frac{\boldsymbol{\beta}_i - \mathbf{x}^\top \mathbf{d}_i^{(0)}}{\|\mathbf{x}\|_2} \\
&= \frac{\mathbf{x}^\top \mathbf{d}_i^* - \mathbf{x}^\top \mathbf{d}_i^{(0)}}{\|\mathbf{x}\|_2} \\
&\leq \frac{\|\mathbf{d}_i^* - \mathbf{d}_i^{(0)}\|_2 \|\mathbf{x}\|_2}{\|\mathbf{x}\|_2} \\
&\leq \|\mathbf{D}^* - \mathbf{D}^{(0)}\|_{1,2} \\
&= \rho \\
&= |b|.
\end{aligned}
\tag{15}
$$

Now, upon substituting these parameters in Lemma C.5, we have $\|\mathbf{c} - \mathbf{a}\|_2 \geq \|\mathbf{b} - \mathbf{a}\|_2$, which can be rewritten as:

$$
\left\| \frac{\mathbf{x}^\top \mathbf{d}_i^{(t+0.5)} - \mathbf{x}^\top \mathbf{d}_i^{(0)}}{\|\mathbf{x}\|_2^2} \mathbf{x} - \frac{\boldsymbol{\beta}_i - \mathbf{x}^\top \mathbf{d}_i^{(0)}}{\|\mathbf{x}\|_2^2} \mathbf{x} \right\|_2 \geq \left\| \rho \, \mathrm{sign}\left( \mathbf{x}^\top \mathbf{d}_i^{(t+0.5)} - \mathbf{x}^\top \mathbf{d}_i^{(0)} \right) \frac{\mathbf{x}}{\|\mathbf{x}\|_2} - \frac{\boldsymbol{\beta}_i - \mathbf{x}^\top \mathbf{d}_i^{(0)}}{\|\mathbf{x}\|_2^2} \mathbf{x} \right\|_2
$$

$$
\left\| \left( \mathbf{d}_i^{(t+0.5)} - \mathbf{d}_i^{(0)} \right) - \left( \widehat{\mathbf{d}}_i - \mathbf{d}_i^{(0)} \right) \right\|_2 \overset{(a)}{\geq} \left\| \left( \mathbf{d}_i^{(t+1)} - \mathbf{d}_i^{(0)} \right) - \left( \widehat{\mathbf{d}}_i - \mathbf{d}_i^{(0)} \right) \right\|_2
$$

$$
\|\mathbf{d}_i^{(t+0.5)} - \widehat{\mathbf{d}}_i\|_2 \geq \|\mathbf{d}_i^{(t+1)} - \widehat{\mathbf{d}}_i\|_2
$$

Here inequality (a) is obtained by Equation (13), Equation (12), and Equation (14). We can now conclude that:

$$
\|\mathbf{d}_i^{(t+1)} - \widehat{\mathbf{d}}_i\|_2 \leq \|\mathbf{d}_i^{(t+0.5)} - \widehat{\mathbf{d}}_i\|_2 \leq \tau \|\mathbf{d}_i^{(t)} - \widehat{\mathbf{d}}_i\|_2.
\tag{16}
$$

Finally, we are ready to establish $A(t+1)$. We rewrite $\mathcal{L}(\tilde{f}_{\boldsymbol{D}^{(t)}})$ as:

$$
\begin{aligned}
\mathcal{L}(\tilde{f}_{\boldsymbol{D}^{(t)}}) &= \|\boldsymbol{D}_{S^*}^* \boldsymbol{D}_{S^*}^{(t)\top} \mathbf{x} - \mathbf{x}\|_2^2 \\
&= \|\boldsymbol{D}_{S^*}^* \boldsymbol{D}_{S^*}^{(t)\top} \mathbf{x} - \boldsymbol{D}_{S^*}^* \boldsymbol{\beta}_{S^*}\|_2^2 \\
&= \|\boldsymbol{D}_{S^*}^{(t)\top} \mathbf{x} - \boldsymbol{\beta}_{S^*}\|_2^2 \\
&= \sum_{i \in S^*} \left( \mathbf{x}^\top \mathbf{d}_i^{(t)} - \boldsymbol{\beta}_i \right)^2 \\
&= \sum_{i \in S^*} \left\| \frac{\boldsymbol{\beta}_i - \mathbf{x}^\top \mathbf{d}_i^{(t)}}{\|\mathbf{x}\|_2^2} \mathbf{x} \right\|_2^2 \|\mathbf{x}\|_2^2 \\
&\overset{B(t)}{=} \sum_{i \in S^*} \left\| \mathbf{d}_i^{(t)} - \widehat{\mathbf{d}}_i \right\|_2^2 \|\mathbf{x}\|_2^2
\end{aligned}
$$

On the other hand,

$$
\mathcal{L}(\tilde{f}_{\boldsymbol{D}^{(t+1)}}) = \sum_{i \in S^*} \left\| \mathbf{d}_i^{(t+1)} - \widehat{\mathbf{d}}_i \right\|_2^2 \|\mathbf{x}\|_2^2
$$

can be established by identical arguments. Finally, we have

$$\mathcal{L}(\tilde{f}_{\boldsymbol{D}^{(t+1)}}) = \sum_{i \in S^*} \left\| \mathbf{d}_i^{(t+1)} - \widehat{\mathbf{d}}_i \right\|_2^2 \|\mathbf{x}\|_2^2 \overset{(16)}{\leq} \tau^2 \sum_{i \in S^*} \left\| \mathbf{d}_i^{(t)} - \widehat{\mathbf{d}}_i \right\|_2^2 \|\mathbf{x}\|_2^2 = \mathcal{L}(\tilde{f}_{\boldsymbol{D}^{(t)}}),$$

which is exactly $B(t+1)$. As a result, we have proved Theorem 3.3.

### C.3. Proof of Theorem 3.4

To prove the statements of this theorem, we first establish the convergence of $\|\mathbf{d}_i^{(t)} - \mathbf{d}_i^*\|_2$ for every $i \in [n]$. Then, we provide the desired upper bound on $\mathcal{L}_m\left(\tilde{f}_{\boldsymbol{D}^{(t)}}\right)$ in terms of $\|\boldsymbol{D}^{(t)} - \boldsymbol{D}^*\|_{1,2}^2$.

We will follow the same index convention defined in Equation (10) and Equation (11), with $\mathcal{L}$ replaced by $\mathcal{L}_m$. For each $h \in [m]$, we use $\boldsymbol{\beta}^h$ and $S^{h*}$ to denote the corresponding variables in Assumption 3.2. We define the set $Q_i$ as the index set of $h$ such that $i$ is in the support of $\boldsymbol{\beta}^h$:

$$Q_i := \left\{ h \in [m] \mid i \in S^{h*} \right\}.$$

Given Lemma C.4, we have:

$$\frac{\partial \mathcal{L}_m(\tilde{f}_{\boldsymbol{D}^{(t)}})}{\partial \mathbf{d}_i^{(t)}} = \frac{1}{m} \sum_{h \in Q_i} 2(\mathbf{d}_i^{(t)\top} \mathbf{x}^h - \beta_i^h) \mathbf{x}^h, \qquad \forall i \in [n].$$

To ensure that $\frac{\partial \mathcal{L}_m(\tilde{f}_{\boldsymbol{D}^{(t)}})}{\partial \mathbf{d}_i^{(t)}}$ is indeed aligned with $\mathbf{d}_i^{(t)} - \mathbf{d}_i^*$, we first need the following lemma:

**Lemma C.6.** *Suppose that $m = \Omega\left(\frac{n^6}{\sigma^2 k^5}\right)$. With probability at least $1 - 2\exp\{\log n - n\} - 2\exp\left\{\log n - \frac{km}{8n}\right\}$, for all $i \in [n]$,*

$$\frac{1}{m} \sigma_d \left( \sum_{h \in Q_i} \mathbf{x}^h \mathbf{x}^{h\top} \right) \geq \frac{k(k-1)\sigma^2}{4n^2},$$

$$\frac{1}{m} \sigma_1 \left( \sum_{h \in Q_i} \mathbf{x}^h \mathbf{x}^{h\top} \right) \leq \frac{4k\sigma^2}{n}.$$

We defer the proof of Lemma C.6 to Appendix C.4.7. Now, we can bound $\|\mathbf{d}_i^{(t+0.5)} - \mathbf{d}_i^*\|_2^2$ as:

$$\|\mathbf{d}_i^{(t+0.5)} - \mathbf{d}_i^*\|_2^2 = \left\| \mathbf{d}_i^{(t)} - \frac{\eta}{m} \sum_{h \in Q_i} 2(\mathbf{d}_i^{(t)\top} \mathbf{x}^h - \beta_i^h) \mathbf{x}^h - \mathbf{d}_i^* \right\|_2^2$$

$$= \|\mathbf{d}_i^{(t)} - \mathbf{d}_i^*\|_2^2 - 4\eta \left\langle \mathbf{d}_i^{(t)} - \mathbf{d}_i^*, \frac{1}{m} \sum_{h \in Q_i} (\mathbf{d}_i^{(t)\top} \mathbf{x}^h - \beta_i^h) \mathbf{x}^h \right\rangle + 4\eta^2 \left\| \frac{1}{m} \sum_{h \in Q_i} (\mathbf{d}_i^{(t)\top} \mathbf{x}^h - \beta_i^h) \mathbf{x}^h \right\|_2^2. \quad (17)$$

For the second term on the right hand side, we have:

$$
\left\langle \mathbf{d}_i^{(t)} - \mathbf{d}_i^*, \frac{1}{m} \sum_{h \in Q_i} (\mathbf{d}_i^{(t)\top} \mathbf{x}^h - \beta_i^h) \mathbf{x}^h \right\rangle
$$

$$
= \left\langle \mathbf{d}_i^{(t)} - \mathbf{d}_i^*, \frac{1}{m} \sum_{h \in Q_i} (\mathbf{d}_i^{(t)\top} \mathbf{x}^h - \mathbf{d}_i^{*\top} \mathbf{x}^h) \mathbf{x}^h \right\rangle
$$

$$
= \left\langle \mathbf{d}_i^{(t)} - \mathbf{d}_i^*, \frac{1}{m} \sum_{h \in Q_i} \mathbf{x}^h \mathbf{x}^{h\top} (\mathbf{d}_i^{(t)} - \mathbf{d}_i^*) \right\rangle
$$

$$
\geq \frac{1}{m} \sigma_d \left( \sum_{h \in Q_i} \mathbf{x}^h \mathbf{x}^{h\top} \right) \left\| \mathbf{d}_i^{(t)} - \mathbf{d}_i^* \right\|_2^2
$$

$$
\overset{Lemma\ C.6}{\geq} \frac{k(k-1)\sigma^2}{4n^2} \left\| \mathbf{d}_i^{(t)} - \mathbf{d}_i^* \right\|_2^2.
$$

For the third term on the right hand side of Equation (17), we have:

$$
\left\| \frac{1}{m} \sum_{h \in Q_i} (\mathbf{d}_i^{(t)\top} \mathbf{x}^h - \beta_i^h) \mathbf{x}^h \right\|_2^2
$$

$$
= \left\| \frac{1}{m} \sum_{h \in Q_i} \mathbf{x}^h \mathbf{x}^{h\top} (\mathbf{d}_i^{(t)} - \mathbf{d}_i^*) \right\|_2^2
$$

$$
\leq \frac{1}{m^2} \sigma_1^2 \left( \sum_{h \in Q_i} \mathbf{x}^h \mathbf{x}^{h\top} \right) \left\| \mathbf{d}_i^{(t)} - \mathbf{d}_i^* \right\|_2^2
$$

$$
\overset{Lemma\ C.6}{\leq} \frac{16 k^2 \sigma^4}{n^2} \left\| \mathbf{d}_i^{(t)} - \mathbf{d}_i^* \right\|_2^2.
$$

When $\eta < \frac{k-1}{128 k \sigma^2}$, the above two inequalities can be combined with Equation (17) to arrive at:

$$
\begin{aligned}
\| \mathbf{d}_i^{(t+0.5)} - \mathbf{d}_i^* \|_2^2 &\leq \left( 1 - \frac{k(k-1)\sigma^2}{n^2} \eta + \frac{64 k^2 \sigma^4}{n^2} \eta^2 \right) \| \mathbf{d}_i^{(t)} - \mathbf{d}_i^* \|_2^2 \\
&\leq \left( 1 - \frac{k(k-1)\sigma^2}{2n^2} \eta \right) \| \mathbf{d}_i^{(t)} - \mathbf{d}_i^* \|_2^2 \\
&\leq \tau^2 \| \mathbf{d}_i^{(t)} - \mathbf{d}_i^* \|_2^2.
\end{aligned}
\tag{18}
$$

Next, we aim to show that $\| \mathbf{d}_i^{(t+1)} - \mathbf{d}_i^* \|_2^2 \leq \| \mathbf{d}_i^{(t+0.5)} - \mathbf{d}_i^* \|_2^2$. Recall that:

$$
\mathbf{d}_i^{(t+1)} = \arg \min_{\mathbf{d} \in D} \| \mathbf{d} - \mathbf{d}_i^{(t+0.5)} \|_2^2 \quad \text{where} \quad D = \{ \mathbf{d} : \| \mathbf{d} - \mathbf{d}_i^{(0)} \|_2 \leq \rho \}.
$$

It is obvious that $D$ is convex and $\| \mathbf{d} - \mathbf{d}_i^{(t+0.5)} \|_2^2$ is strongly convex with respect to $\mathbf{d}$. As a result, first order stationary condition requires that:

$$
\left\langle \nabla_{\mathbf{d} = \mathbf{d}_i^{(t+1)}} \| \mathbf{d} - \mathbf{d}_i^{(t+0.5)} \|_2^2, \ \tilde{\mathbf{d}} - \mathbf{d}_i^{(t+1)} \right\rangle = 2 \left\langle \mathbf{d}_i^{(t+1)} - \mathbf{d}_i^{(t+0.5)}, \tilde{\mathbf{d}} - \mathbf{d}_i^{(t+1)} \right\rangle \geq 0,
$$

for any $\tilde{\mathbf{d}} \in D$. Given the initial error bound $\| \mathbf{D}^{(0)} - \mathbf{D}^* \|_{1,2} \leq \rho$, we have $\mathbf{d}^* \in D$, which results in:

$$
\left\langle \mathbf{d}_i^{(t+1)} - \mathbf{d}_i^{(t+0.5)}, \mathbf{d}^* - \mathbf{d}_i^{(t+1)} \right\rangle \geq 0.
$$

We subsequently have:

$$\|\mathbf{d}_i^* - \mathbf{d}_i^{(t+0.5)}\|_2^2 = \left\|\left(\mathbf{d}_i^{(t+1)} - \mathbf{d}_i^{(t+0.5)}\right) + \left(\mathbf{d}^* - \mathbf{d}_i^{(t+1)}\right)\right\|_2^2$$

$$= \|\mathbf{d}_i^{(t+1)} - \mathbf{d}_i^{(t+0.5)}\|_2^2 + 2\left\langle\mathbf{d}_i^{(t+1)} - \mathbf{d}_i^{(t+0.5)}, \mathbf{d}^* - \mathbf{d}_i^{(t+1)}\right\rangle + \|\mathbf{d}_i^* - \mathbf{d}_i^{(t+1)}\|_2^2$$

$$\geq \|\mathbf{d}_i^* - \mathbf{d}_i^{(t+1)}\|_2^2, \tag{19}$$

where the last inequality follows from the fact that both $\|\mathbf{d}_i^{(t+1)} - \mathbf{d}_i^{(t+0.5)}\|_2^2$ and $\left\langle\mathbf{d}_i^{(t+1)} - \mathbf{d}_i^{(t+0.5)}, \mathbf{d}^* - \mathbf{d}_i^{(t+1)}\right\rangle$ are non-negative. Combining Equation (18) and Equation (19), we have

$$\|\mathbf{d}_i^{(t+1)} - \mathbf{d}_i^*\|_2^2 \leq \|\mathbf{d}_i^{(t+0.5)} - \mathbf{d}_i^*\|_2^2 \leq \tau^2\|\mathbf{d}_i^{(t)} - \mathbf{d}_i^*\|_2^2.$$

Finally, note that

$$\|\boldsymbol{D}^{(t+1)} - \boldsymbol{D}^*\|_{1,2} = \max_{i\in[n]}\{\|\mathbf{d}_i^{(t+1)} - \mathbf{d}_i^*\|_2\} \leq \tau\max_{i\in[n]}\{\|\mathbf{d}_i^{(t)} - \mathbf{d}_i^*\|_2\} \leq \tau\|\boldsymbol{D}^{(t)} - \boldsymbol{D}^*\|_{1,2},$$

Next, we show that $\mathcal{L}_m\left(\boldsymbol{D}^{(t)}\right)$ is upper bounded by $\|\boldsymbol{D}^{(t)} - \boldsymbol{D}^*\|_{1,2}$:

$$\mathcal{L}_m\left(\tilde{f}_{\boldsymbol{D}^{(t)}}\right) = \frac{1}{m}\sum_{h=1}^m \|\boldsymbol{D}_{S^{h*}}^* \boldsymbol{D}_{S^{h*}}^{(t)\top}\mathbf{x}^h - \mathbf{x}^h\|_2^2$$

$$= \frac{1}{m}\sum_{h=1}^m \|\boldsymbol{D}_{S^{h*}}^{(t)\top}\mathbf{x}^h - \boldsymbol{\beta}_{S^{h*}}^h\|_2^2$$

$$= \frac{1}{m}\sum_{h=1}^m \sum_{i\in S^{h*}} \left(\mathbf{x}^{h\top}\mathbf{d}_i^{(t)} - \mathbf{x}^{h\top}\mathbf{d}_i^*\right)^2$$

$$\leq \frac{1}{m}\sum_{h=1}^m \sum_{i\in S^{h*}} \|\mathbf{x}^h\|_2^2\|\mathbf{d}_i^{(t)} - \mathbf{d}_i^*\|_2^2$$

$$\leq \frac{1}{m}\sum_{h=1}^m k\|\mathbf{x}^h\|_2^2\|\boldsymbol{D}^{(t)} - \boldsymbol{D}^*\|_{1,2}^2$$

$$= \frac{k\sum_{h=1}^m \|\mathbf{x}^h\|_2^2}{m}\|\boldsymbol{D}^{(t)} - \boldsymbol{D}^*\|_{1,2}^2.$$

To complete the proof of Theorem 3.4, we need to examine the success probability which the above linear convergence occurs with. The only probability statement we use is Lemma C.6 and we only invoke it once. So the total success probability is at least:

$$1 - 2\exp\{\log n - n\} - 2\exp\left\{\log n - \frac{km}{8n}\right\},$$

which is $1 - n^{-\omega(1)}$ when $m = \Omega\left(\frac{n^6}{\sigma^2 k^5}\right)$. This completes the proof of Theorem 3.4. $\qquad\square$

## C.4. Proof of Lemmas

### C.4.1. PROOF OF LEMMA 3.1

Lemma 3.1 directly follows from Lemma C.1 and Lemma C.2.

### C.4.2. PROOF OF LEMMA C.1

We can decompose $y$ as follows:

$$y = \langle\mathbf{x}, \mathbf{z}\rangle = \langle\Pi_{\boldsymbol{D}_S^*}\mathbf{x}, \mathbf{z}\rangle + \langle\Pi_{\boldsymbol{D}_S^*}^{\perp}\mathbf{x}, \mathbf{z}\rangle$$

Note that $\mathbf{z} \sim \mathcal{N}(\mathbf{0}, \boldsymbol{I}_{d \times d})$. This implies that $\Pi_{\boldsymbol{D}_S^*}^{\perp} \mathbf{z}$ is independent of $\Pi_{\boldsymbol{D}_S^*} \mathbf{z}$ (Anderson et al., 1958), which in turn entails that $\langle \Pi_{\boldsymbol{D}_S^*}^{\perp} \mathbf{x}, \mathbf{z} \rangle$ is independent of $\langle \Pi_{\boldsymbol{D}_S^*} \mathbf{x}, \mathbf{z} \rangle$ and the events $\{\langle \mathbf{d}_i^*, \mathbf{z} \rangle = r_i \quad \forall i \in S\}$. Therefore, we have:

$$\text{MLE}\left(\langle \mathbf{x}, \mathbf{z} \rangle \mid \langle \mathbf{d}_i^*, \mathbf{z} \rangle = r_i \quad \forall i \in S\right) = \quad \text{MLE}\left(\langle \Pi_{\boldsymbol{D}_S^*} \mathbf{x}, \mathbf{z} \rangle \mid \langle \mathbf{d}_i^*, \mathbf{z} \rangle = r_i \quad \forall i \in S\right)$$
$$+ \text{MLE}\left(\langle \Pi_{\boldsymbol{D}_S^*}^{\perp} \mathbf{x}, \mathbf{z} \rangle\right).$$

It turns out that the first term on the right hand side is deterministic:

$$\langle \Pi_{\boldsymbol{D}_S^*} \mathbf{x}, \mathbf{z} \rangle = \langle \boldsymbol{D}_S^* \boldsymbol{D}_S^{*\top} \mathbf{x}, \mathbf{z} \rangle$$
$$= \langle \sum_{i \in S} \langle \mathbf{d}_i^*, \mathbf{x} \rangle \mathbf{d}_i^*, \mathbf{z} \rangle$$
$$= \sum_{i \in S} \langle \mathbf{d}_i^*, \mathbf{x} \rangle \langle \mathbf{d}_i^*, \mathbf{z} \rangle$$
$$= \sum_{i \in S} \langle \mathbf{d}_i^*, \mathbf{x} \rangle r_i.$$

For the last equality we used $q_i = r_i$. For the second term, we have:

$$\text{MLE}\left(\langle \Pi_{\boldsymbol{D}_S^*}^{\perp} \mathbf{x}, \mathbf{z} \rangle\right) = \text{MLE}\left(\langle \Pi_{\boldsymbol{D}_S^*}^{\perp} \mathbf{x}, \Pi_{\boldsymbol{D}_S^*}^{\perp} \mathbf{z} \rangle\right)$$
$$\overset{(a)}{=} 0.$$

The equality $(a)$ is due to the observation that $\Pi_{\boldsymbol{D}_S^*}^{\perp} \mathbf{z}$ is indeed distributed as $\mathcal{N}(\mathbf{0}, \boldsymbol{I}_{(d-k) \times (d-k)})$ after a simple dimensionality reduction (Anderson et al., 1958). The resulting random variable $\langle \Pi_{\boldsymbol{D}_S^*}^{\perp} \mathbf{x}, \Pi_{\boldsymbol{D}_S^*}^{\perp} \mathbf{z} \rangle$ is distributed as $\mathcal{N}(0, \|\Pi_{\boldsymbol{D}_S^*}^{\perp} \mathbf{x}\|_2^2)$, leading to the equality $(a)$. To sum up, we have

$$\text{MLE}\left(\langle \mathbf{x}, \mathbf{z} \rangle \mid \langle \mathbf{d}_i^*, \mathbf{z} \rangle = r_i \quad \forall i \in S\right) = \sum_{i \in S} \langle \mathbf{d}_i^*, \mathbf{x} \rangle r_i.$$

$\square$

### C.4.3. PROOF OF LEMMA C.2

Let us define

$$S_k = \underset{T \subseteq [n], |T| \leq k}{\arg\max} \sum_{i \in T} |\langle \mathbf{d}_i, \mathbf{x} \rangle|.$$

We will use induction to show that $S_k$ corresponds to the indices selected by IP-OMP after $k$ iterations. At the first iteration, according to (Chattopadhyay et al., 2024), IP-OMP simply selects the index $i = \arg\max_{j \in [n]} |\langle \mathbf{d}_j, \mathbf{x} \rangle| / \|\mathbf{d}_j\|_2 \|\mathbf{x}\|_2 = \arg\max_{j \in [n]} |\langle \mathbf{d}_j, \mathbf{x} \rangle|$, where the second equality is due to $\|\mathbf{d}_j\|_2 = 1$ for all $j \in [n]$. As a result we have $S_1 = \{i\}$ and the base case is established.

Suppose that the statement holds for $S_{k-1} = \arg\max_{T \subseteq [n], |T| \leq k-1} \sum_{i \in T} |\langle \mathbf{d}_i, \mathbf{x} \rangle|$. Let $\boldsymbol{D}_{S_{k-1}}$ be the submatrix of $\boldsymbol{D}$ which consists of columns indexed by $S_{k-1}$. Then IP-OMP will select

$$i = \underset{j \in [n], j \notin S_{k-1}}{\arg\max} \frac{|\langle \Pi_{\boldsymbol{D}_{S_{k-1}}}^{\perp} \mathbf{d}_j, \Pi_{\boldsymbol{D}_{S_{k-1}}}^{\perp} \mathbf{x} \rangle|}{\|\Pi_{\boldsymbol{D}_{S_{k-1}}}^{\perp} \mathbf{d}_j\|_2 \|\Pi_{\boldsymbol{D}_{S_{k-1}}}^{\perp} \mathbf{x}\|_2}.$$

As $\boldsymbol{D}$ is column-orthogonal, we have $\Pi_{\boldsymbol{D}_{S_{k-1}}}^{\perp} \mathbf{d}_j = \mathbf{d}_j$ and $\Pi_{\boldsymbol{D}_{S_{k-1}}} \mathbf{d}_j = \mathbf{0} \ \forall j \notin S_{k-1}$. Therefore, for $\forall j \notin S_{k-1}$, we have

$$\langle \mathbf{d}_j, \mathbf{x} \rangle = \langle \Pi_{\boldsymbol{D}_{S_{k-1}}}^{\perp} \mathbf{d}_j, \Pi_{\boldsymbol{D}_{S_{k-1}}}^{\perp} \mathbf{x} \rangle + \langle \Pi_{\boldsymbol{D}_{S_{k-1}}} \mathbf{d}_j, \Pi_{\boldsymbol{D}_{S_{k-1}}} \mathbf{x} \rangle$$
$$= \langle \Pi_{\boldsymbol{D}_{S_{k-1}}}^{\perp} \mathbf{d}_j, \Pi_{\boldsymbol{D}_{S_{k-1}}}^{\perp} \mathbf{x} \rangle.$$

As a result, we can rewrite the selection rule of IP-OMP as

$$
\begin{aligned}
i &= \underset{j \in [n], j \notin S_{k-1}}{\arg\max} \frac{|\langle \Pi_{\boldsymbol{D}_{S_{k-1}}}^{\perp} \mathbf{d}_j, \Pi_{\boldsymbol{D}_{S_{k-1}}}^{\perp} \mathbf{x} \rangle|}{\|\Pi_{\boldsymbol{D}_{S_{k-1}}}^{\perp} \mathbf{d}_j\|_2 \|\Pi_{\boldsymbol{D}_{S_{k-1}}}^{\perp} \mathbf{x}\|_2} \\
&= \underset{j \in [n], j \notin S_{k-1}}{\arg\max} \frac{|\langle \mathbf{d}_j, \mathbf{x} \rangle|}{\|\mathbf{d}_j\|_2 \|\mathbf{x}\|_2} \\
&= \underset{j \in [n], j \notin S_{k-1}}{\arg\max} |\langle \mathbf{d}_j, \mathbf{x} \rangle|.
\end{aligned}
$$

The last equality is due to $\|\mathbf{d}_j\|_2 = 1$ and the fact that $\|\mathbf{x}\|_2$ is the same for different $j$. This implies that, at iteration $k$, IP-OMP will select the index $j \notin S_{k-1}$ with the largest $|\langle \mathbf{d}_j, \mathbf{x} \rangle|$, which completes the proof of our induction.

Finally, when $\boldsymbol{D} = \boldsymbol{D}^*$, one can verify that $\langle \mathbf{d}_i, \mathbf{x} \rangle = \langle \mathbf{d}_i^*, \boldsymbol{D}^* \boldsymbol{\beta} \rangle = 0$ for $i \notin S^*$. This implies that $S_k = S^*$. $\qquad\square$

### C.4.4. PROOF OF LEMMA C.3

To prove $\tilde{\boldsymbol{D}}$ is column-orthogonal, we first observe that, for $i \leq k$, we have $\|\tilde{\mathbf{d}}_i\|_2^2 = \cos^2 \theta + \sin^2 \theta = 1$. Moreover, for any pair $i < j \leq k/2$, we have

$$
\langle \tilde{\mathbf{d}}_i, \tilde{\mathbf{d}}_j \rangle = \langle \cos\theta \mathbf{d}_i^* + \sin\theta \mathbf{d}_{i+k/2}^*, \cos\theta \mathbf{d}_j^* + \sin\theta \mathbf{d}_{j+k/2}^* \rangle = 0,
$$

which follows since $\langle \mathbf{d}_i^*, \mathbf{d}_j^* \rangle = \langle \mathbf{d}_{i+k/2}^*, \mathbf{d}_j^* \rangle = \langle \mathbf{d}_i^*, \mathbf{d}_{j+k/2}^* \rangle = \langle \mathbf{d}_{i+k/2}^*, \mathbf{d}_{j+k/2}^* \rangle = 0$. A similar argument can be made for any $i < j \leq k$ and $j \neq i + k/2$. For any $i \leq k$ and $j = i + k/2$, we have

$$
\langle \tilde{\mathbf{d}}_i, \tilde{\mathbf{d}}_j \rangle = \langle \cos\theta \mathbf{d}_i^* + \sin\theta \mathbf{d}_{i+k/2}^*, -\sin\theta \mathbf{d}_i^* + \cos\theta \mathbf{d}_{i+k/2}^* \rangle = -\sin\theta \cos\theta + \sin\theta \cos\theta = 0.
$$

Finally, for any $j > i > k$ we trivially have $\langle \tilde{\mathbf{d}}_i, \tilde{\mathbf{d}}_j \rangle = 0$. This completes the proof of column-orthogonality of $\tilde{\boldsymbol{D}}$.

To prove the second statement, note that $\|\tilde{\mathbf{d}}_i - \mathbf{d}_i^*\|_2 = 0$ for every $i > k$. Moreover, for every $i \leq k$, we have

$$
\|\tilde{\mathbf{d}}_i - \mathbf{d}_i^*\|_2 = \|(\cos\theta - 1)\mathbf{d}_i^* + \sin\theta \mathbf{d}_{i+k}^*\|_2 = \sqrt{(\cos\theta - 1)^2 + \sin^2\theta} = \sqrt{2 - 2\cos\theta} = 2\sin\frac{\theta}{2}.
$$

This completes the proof. $\qquad\square$

### C.4.5. PROOF OF LEMMA C.4

The proof strategy we adopt here is similar with the proof of Lemma 3.1 in (Liang et al., 2022). However, we consider a different generative model in this paper, so we present the full proof for the purpose of completeness. We will show that for $i \in S^*$, $|\langle \mathbf{d}_i, \mathbf{x} \rangle| > \gamma/2$ and for $i \notin S^*$, $|\langle \mathbf{d}_i, \mathbf{x} \rangle| < \gamma/2$, which will immediately result in $S = \arg\max_{T \subseteq [n], |T| \leq k} \sum_{i \in T} |\langle \mathbf{d}_i, \mathbf{x} \rangle| = S^*$, which proves Lemma C.4.

We decompose $\langle \mathbf{d}_i, \mathbf{x} \rangle$ as:

$$
\begin{aligned}
\langle \mathbf{d}_i, \mathbf{x} \rangle &= \langle \mathbf{d}_i, \sum_{j \in S^*} \mathbf{d}_j^* \beta_j \rangle \\
&= \underbrace{\langle \mathbf{d}_i, \mathbf{d}_i^* \rangle \beta_i}_{:=\mathcal{A}_i} + \underbrace{\sum_{j \neq i, j \in S^*} \langle \mathbf{d}_i, \mathbf{d}_j^* \rangle \beta_j}_{:=\mathcal{B}_i}
\end{aligned}
\tag{20}
$$

For $\mathcal{A}_i$, we have:

$$
\mathcal{A}_i = \langle \mathbf{d}_i, \mathbf{d}_i^* \rangle \beta_i = \langle \mathbf{d}_i^*, \mathbf{d}_i^* \rangle \beta_i + \langle \mathbf{d}_i - \mathbf{d}_i^*, \mathbf{d}_i^* \rangle \beta_i = \beta_i + \langle \mathbf{d}_i - \mathbf{d}_i^*, \mathbf{d}_i^* \rangle \beta_i.
$$

When $i \in S^*$, we have

$$
\begin{aligned}
|\beta_i + \langle \mathbf{d}_i - \mathbf{d}_i^*, \mathbf{d}_i^* \rangle \beta_i| &\geq (1 - \|\mathbf{d}_i - \mathbf{d}_i^*\|_2)|\beta_i| \\
&\geq (1 - \|\boldsymbol{D} - \boldsymbol{D}^*\|_{1,2})|\beta_i| \\
&\geq (1 - 2\rho)|\beta_i|.
\end{aligned}
$$

Given that $2\rho < \frac{\gamma}{4\sqrt{k}\Gamma} \le \frac{1}{4}$, we can conclude

$$|\mathcal{A}_i| \begin{cases} \ge \frac{3\gamma}{4} & \text{if} \quad i \in S^* \\ = 0 & \text{if} \quad i \notin S^* \end{cases}, \tag{21}$$

given that $\beta_i = 0$ when $i \notin S^*$. For $\mathcal{B}_i$, we have:

$$
\begin{aligned}
|\mathcal{B}_i| &= \left| \sum_{j \ne i, j \in S^*} \langle \mathbf{d}_i, \mathbf{d}_j^* \rangle \beta_j \right| \\
&= \left| \sum_{j \ne i, j \in S^*} \langle \mathbf{d}_i^* + (\mathbf{d}_i - \mathbf{d}_i^*), \mathbf{d}_j^* \rangle \beta_j \right| \\
&= \left| \sum_{j \ne i, j \in S^*} \langle \mathbf{d}_i - \mathbf{d}_i^*, \mathbf{d}_j^* \rangle \beta_j \right| \\
&\le \sum_{j \ne i, j \in S^*} \langle |\mathbf{d}_i - \mathbf{d}_i^*, \mathbf{d}_j^* \rangle | |\beta_j| \\
&\le \Gamma \|\mathbf{d}_i - \mathbf{d}_i^*\|_1 \\
&\le \sqrt{k} \Gamma \|\mathbf{d}_i - \mathbf{d}_i^*\|_2 \\
&\le 2\sqrt{k}\Gamma\rho \\
&\le \gamma/4
\end{aligned}
\tag{22}
$$

Combining Equation (21) and Equation (22), we have that for all $i$:

$$|\langle \mathbf{d}_i, \mathbf{x} \rangle| = |\mathcal{A}_i + \mathcal{B}_i| \begin{cases} > \frac{3\gamma}{4} - \frac{\gamma}{4} = \frac{\gamma}{2} & \text{if} \quad i \in S^* \\ < \frac{\gamma}{4} & \text{if} \quad i \notin S^* \end{cases}, \tag{23}$$

which proves the exact support recovery.

### C.4.6. PROOF OF LEMMA C.5

It is easy to see that $\|\mathbf{c} - \mathbf{a}\|_2 \ge \|\mathbf{b} - \mathbf{a}\|_2$ is equivalent to $|c - a| \ge |b - a|$.

If $c > 0$, by $bc > 0$ we must have $b > 0$. Then $|c| \ge |b| \ge a$ becomes $c \ge b \ge a$, which gives $|c - a| \ge |b - a|$.

If $c < 0$, by $bc > 0$ we must have $b < 0$. Then $|c| \ge |b| \ge a$ becomes $a \ge b \ge c$, which gives $|c - a| \ge |b - a|$ as well.

### C.4.7. PROOF OF LEMMA C.6

The proof of Lemma C.6 consists of two steps:

1. We first use classic concentration inequalities from covariance estimation literature to bound $\sigma_d$ and $\sigma_1$ given $|Q_i|$.

2. Then we will show that when $m$ is large enough, with high probability, we have $|Q_i|$ bounded above and below.

Combining these two steps and taking the union bound will complete the proof of Lemma C.6.

For the fist step, we first notice that:

$$
\begin{aligned}
\sigma_d \left( \sum_{h \in Q_i} \mathbf{x}^h \mathbf{x}^{h\top} \right) &= \sigma_d \left( \sum_{h \in Q_i} \mathbf{D}^* \boldsymbol{\beta}^h \boldsymbol{\beta}^{h\top} \mathbf{D}^{*\top} \right) \\
&= \sigma_d \left( \sum_{h \in Q_i} \boldsymbol{\beta}^h \boldsymbol{\beta}^{h\top} \right)
\end{aligned}
$$

The last equation is due to the fact that $\boldsymbol{D}^*$ is a full-rank orthogonal matrix. Consider the vector $\boldsymbol{\beta}^h$ for $h \in Q_i$. According to our generative model and the definition of $Q_i$, we have $\boldsymbol{\beta}_i^h \neq 0$, and its support outside $i$ is selected uniformly over all $(k-1)$-element subsets of $[n] \setminus \{i\}$. Each nonzero entry of $\boldsymbol{\beta}^h$ has a zero mean, variance of $\sigma^2$, and an absolute value bounded between $\gamma$ and $\Gamma$. Therefore, its covariance matrix defined as $\boldsymbol{\Sigma} := \mathbb{E}\left[\boldsymbol{\beta}^h \boldsymbol{\beta}^{h\top}\right]$ is diagonal, with its $(i,i)$-th entry corresponding to $\sigma^2$, and its $(j,j)$-th entry (with $j \neq i$) corresponding to $\frac{(k-1)\sigma^2}{n}$. We claim that when $|Q_i|$ is sufficiently large, $\sum_{h \in Q_i} \mathbf{x}^h \mathbf{x}^{h\top}$ will concentrate around $|Q_i|\boldsymbol{\Sigma}$. Specifically, we will use the following well-established result:

**Theorem C.7** ((Vershynin, 2018)). *Let $\mathbf{w}$ be a zero-mean sub-Gaussian random vector in $\mathbb{R}^d$ with covariance matrix $\boldsymbol{\Sigma}$, such that*

$$\|\langle \mathbf{w}, \mathbf{q} \rangle\|_{\psi_2} \leq K \left( \mathbb{E}[\langle \mathbf{w}, \mathbf{q} \rangle^2] \right)^{1/2} \quad \text{for any } \mathbf{q} \in \mathbb{R}^d,$$

*for some $K \geq 1$. Here $\| \cdot \|_{\psi_2}$ denotes the sub-Gaussian norm of a random variable. Let $\boldsymbol{W} \in \mathbb{R}^{d \times \widehat{m}}$ be a matrix whose columns have identical and independent distribution as $\mathbf{w}$. Then, for any $u \geq 0$ and with probability at least $1 - 2\exp(-u)$, we have*

$$\left\| \frac{1}{\widehat{m}} \boldsymbol{W} \boldsymbol{W}^\top - \boldsymbol{\Sigma} \right\|_2 \leq C K^2 \left( \sqrt{\frac{d+u}{\widehat{m}}} + \frac{d+u}{\widehat{m}} \right) \|\boldsymbol{\Sigma}\|_2$$

*for some universal constant $C$.*

To invoke Theorem C.7, we note that $\boldsymbol{\beta}^h$ is indeed zero-mean and sub-Gaussian. Therefore, upon setting $\mathbf{w} = \boldsymbol{\beta}^h$, we notice that for any unit-norm $\mathbf{q}$:

$$\|\langle \boldsymbol{\beta}^h, \mathbf{q} \rangle\|_{\psi_2}^2 \leq \sum_{j=1}^d \mathbf{q}_j^2 \|\boldsymbol{\beta}_j^h\|_{\psi_2}^2 \leq C_0 \sigma^2,$$

for some universal constant $C_0$. We also have

$$\mathbb{E}\left[\langle \boldsymbol{\beta}^h, \mathbf{q} \rangle^2\right] = \mathbf{q}^\top \boldsymbol{\Sigma} \mathbf{q} \geq \frac{(k-1)\sigma^2}{n}.$$

By setting $K^2 = \frac{C_0 n}{k-1}$, we can invoke Theorem C.7 with $u = n$ and conclude that, with probability at least $1 - 2\exp\{-n\}$:

$$\sigma_d \left( \sum_{h \in Q_i} \boldsymbol{\beta}^h \boldsymbol{\beta}^{h\top} \right) \geq |Q_i| \sigma_d(\boldsymbol{\Sigma}) - \frac{Cn\sigma}{k-1}\sqrt{2n|Q_i|} = \frac{|Q_i|(k-1)\sigma^2}{n} - \frac{Cn\sigma}{k-1}\sqrt{2n|Q_i|}, \tag{24}$$

$$\sigma_1 \left( \sum_{h \in Q_i} \boldsymbol{\beta}^h \boldsymbol{\beta}^{h\top} \right) \leq |Q_i| \sigma_1(\boldsymbol{\Sigma}) + \frac{Cn\sigma}{k-1}\sqrt{2n|Q_i|} = |Q_i|\sigma^2 + \frac{Cn\sigma}{k-1}\sqrt{2n|Q_i|}, \tag{25}$$

for some universal constant $C$, which concludes the first step. Here we recall that $d = n$ in the full-rank and orthogonal setting.

For the second step, consider one specific $i$ from $[n]$. The random variable $|Q_i|$ follows a binomial distribution with $m$ trials and $\frac{k}{n}$ success rate. We next recall the Chernoff bound:

**Theorem C.8** ((Motwani & Raghavan, 1996)). *Let $X_1, X_2, \ldots, X_N$ be independent Bernoulli random variables with $\mathbb{P}(X_i = 1) = p_i$, and let*

$$X = \sum_{i=1}^N X_i \quad \text{and} \quad \mu = \mathbb{E}[X] = \sum_{i=1}^N p_i.$$

*Then for any $0 < \delta < 1$, we have*

$$\mathbb{P}\big(X \leq (1-\delta)\mu\big) \leq \exp\left\{-\frac{\delta^2 \mu}{2}\right\},$$

*and*

$$\mathbb{P}\big(X \geq (1+\delta)\mu\big) \leq \exp\left\{-\frac{\delta^2 \mu}{2}\right\}.$$

We can then invoke Theorem C.8 with $\delta = \frac{1}{2}$ to obtain

$$\frac{km}{2n} \leq |Q_i| \leq \frac{2km}{n}, \tag{26}$$

with probability at least $1 - 2\exp\left\{-\frac{km}{8n}\right\}$. Finally, by combining Equation (24) and Equation (26), we have

$$\frac{1}{m}\sigma_d\left(\sum_{h \in Q_i} \boldsymbol{\beta}^h \boldsymbol{\beta}^{h\top}\right) \geq \frac{|Q_i|(k-1)\sigma^2}{nm} - \frac{Cn\sigma}{(k-1)m}\sqrt{2n|Q_i|}$$

$$\geq \frac{k(k-1)\sigma^2}{2n^2} - \frac{Cn\sigma}{(k-1)}\sqrt{\frac{4k}{m}}. \tag{27}$$

When $m \geq \frac{64C^2 n^6}{\sigma^2 k(k-1)^4}$, we have

$$\frac{Cn\sigma}{(k-1)}\sqrt{\frac{4k}{m}} \leq \frac{k(k-1)\sigma^2}{4n^2},$$

which reduces Equation (27) to:

$$\frac{1}{m}\sigma_d\left(\sum_{h \in Q_i} \boldsymbol{\beta}^h \boldsymbol{\beta}^{h\top}\right) \geq \frac{k(k-1)\sigma^2}{2n^2} - \frac{k(k-1)\sigma^2}{4n^2} = \frac{k(k-1)\sigma^2}{4n^2}.$$

With a similar argument, we can combine Equation (25) and Equation (26) to get:

$$\frac{1}{m}\sigma_1\left(\sum_{h \in Q_i} \boldsymbol{\beta}^h \boldsymbol{\beta}^{h\top}\right) \leq |Q_i|\sigma^2 + \frac{Cn\sigma}{k-1}\sqrt{2n|Q_i|} \leq \frac{4k\sigma^2}{n}.$$

Finally, we take the union bound for all $i \in [n]$, leading to the overall probability of $1 - 2n\exp\{-n\} - 2n\exp\left\{-\frac{km}{8n}\right\}$. This completes the proof. □

# D. Detailed Experiments

## D.1. Experiments on Generative Model

All experiments reported in this section were performed in Python 3.9 on a MacBook Pro (14-inch, 2021) equipped with an Apple M1 Pro chip.

We generate samples according to the described generative model in Assumption 3.2, we construct the input data $\mathbf{x}$ or $\{\mathbf{x}^h\}_{h=1}^m$ by sampling each non-zero entry from a uniform distribution over the interval $[\gamma, \Gamma]$. The matrix $\boldsymbol{D}^*$ is chosen to be a randomly generated orthonormal matrix, and we construct $\boldsymbol{D}_{\text{init}}$ by setting $\boldsymbol{D}_{\text{init}} = \boldsymbol{D}^* + \boldsymbol{E}$, where each column of $\boldsymbol{E}$ is uniformly drawn from the set $\{\mathbf{e} \mid \|\mathbf{e}\|_2 \leq \rho\}$. For the results shown in Figure 4, we set $d = 10$, $k = 5$, $\rho = 0.2$, $\gamma = 0.5$, and $\Gamma = 1$. The first column of Figure 4 illustrates the scenario corresponding to Theorem 3.3, in which only a single input feature $\mathbf{x}$ is available. Here, we choose $n = 8$ and $\eta = 10^{-2}$. The plot showing $\mathcal{L}(\tilde{f}_{\boldsymbol{D}^{(t)}})$ confirms the linear convergence of the loss function to zero, in agreement with Theorem 3.3. However, as previously noted, this setting does not ensure exact recovery for each queried feature. Indeed, as shown in the right panel of the first row, $\|\boldsymbol{D}^{(t)} - \boldsymbol{D}^*\|_{1,2}$ does not converge to zero. Even if we only consider columns that are activated at each iteration, $\|\boldsymbol{D}_S^{(t)} - \boldsymbol{D}_S^*\|_{1,2}$ also remains nonzero, indicating that optimizing $\mathcal{L}$ with a single $\mathbf{x}$ is insufficient to recover the ground-truth dictionary $\boldsymbol{D}^*$.

In the second column of Figure 4, we apply projected gradient descent to minimize $\mathcal{L}_m$, as defined in Equation (6), under the assumption that $\boldsymbol{D}^*$ is rank-deficient. Specifically, we set $n = 8 < d$ and $\eta = 10^{-1}$. Evidently, $\|\boldsymbol{D}^{(t)} - \boldsymbol{D}^*\|_{1,2}$ does not converge to zero because the component of $\boldsymbol{D}^{(t)} - \boldsymbol{D}^*$ orthogonal to the column space of $\boldsymbol{D}^*$ remains unaffected throughout the iterations. In contrast, the third column of Figure 4 corresponds to the setting $n = d$, which guarantees the full-rankness of $\boldsymbol{D}^*$ is full rank. In this case, $\|\boldsymbol{D}_S^{(t)} - \boldsymbol{D}_S^*\|_{1,2}$ converges to zero, thus supporting the conclusion of Theorem 3.4.

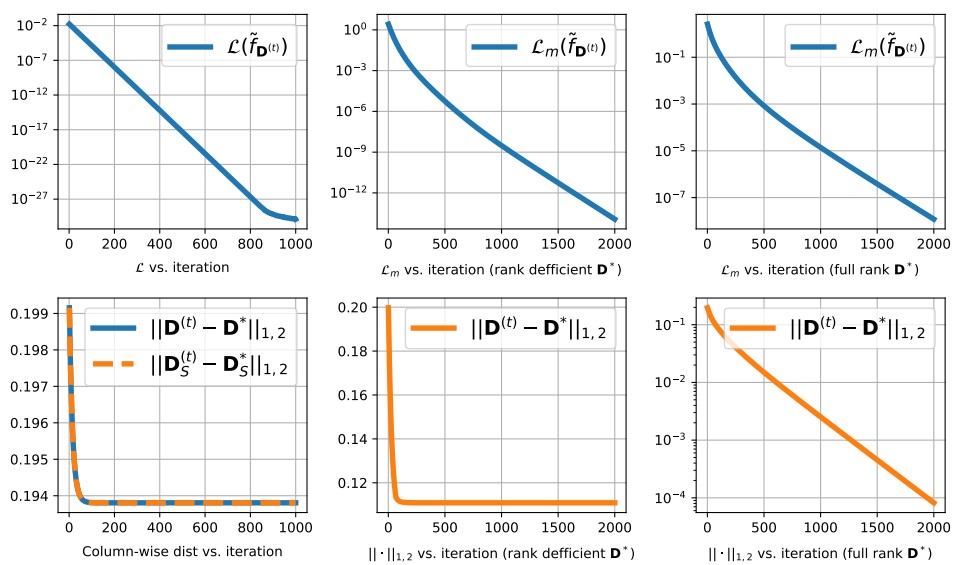

*Figure 4.* Results on synthetic dataset.

## D.2. More Details on Algorithm 2

Here we provide a detailed description for the **concept dispersion** step and the **embedding normalization and projection** step in Algorithm 2. In Algorithm 3, we present the pseudo-code for concept dispersion. The algorithm first calculates the mean concept embedding for the given $\{\mathbf{d}_i\}_{i=1}^n$ and calculates the angle between each $\mathbf{d}$ and the mean. Then, Algorithm 3 increases these angles by a constant factor $r$.

---

**Algorithm 3** Concept dispersion

---

1: **Input**: $\{\mathbf{d}_i\}_{i=1}^n$, dispersion factor $r$.
2: Calculate the mean concept embedding $\bar{\mathbf{d}} = \sum_{i=1}^n \mathbf{d}_i / \| \sum_{i=1}^n \mathbf{d}_i \|_2$.
3: **for** each $\mathbf{d}_i$ **do**
4:     Decompose $\mathbf{d}_i$ as $\mathbf{d}_i = \cos \alpha_i \, \bar{\mathbf{d}} + \sin \alpha_i \, \mathbf{e}_i$, where $\alpha_i \in [0, \pi/2]$, $\|\mathbf{e}_i\|_2 = 1$, and $\mathbf{e}_i \perp \bar{\mathbf{d}}$.
5:     Calculate the dispersed query feature $\mathbf{d}_i^{\mathrm{new}} = \cos(r\alpha_i) \, \bar{\mathbf{d}} + \sin(r\alpha_i) \, \mathbf{e}_i$.
6:     Add $\mathbf{d}_i^{\mathrm{new}}$ as the $i$th column of $\boldsymbol{D}^{\mathrm{init}}$.
7: **end for**
8: **Return** $\boldsymbol{D}^{\mathrm{init}}$.

---

The effectiveness of Algorithm 3 is shown in Figure 5, where we compare the correlation between concept embeddings before and after this process across various datasets.

Algorithm 4 presents the pseudo-code for the projection and normalization step in Algorithm 2.

---

**Algorithm 4** Embedding normalization and projection

---

1: **Input**: current query feature matrix $D$, initial query feature matrix $\boldsymbol{D}^{\mathrm{init}}$, and radius $\rho$.
2: **for** each query feature $\mathbf{d}_i$ in $\boldsymbol{D}$ **do**
3:     Normalize $\mathbf{d}_i$ as $\mathbf{d}_i = \mathbf{d}_i / \|\mathbf{d}_i\|_2$.
4:     **if** $\|\mathbf{d}_i - \mathbf{d}_i^{\mathrm{init}}\|_2 \geq \rho$ **then**
5:         $\mathbf{d}_i = \arg\min_{\mathbf{d}:\|\mathbf{d}-\mathbf{d}_i^{\mathrm{init}}\|_2 \leq \rho, \|\mathbf{d}\|_2=1} \|\mathbf{d} - \mathbf{d}_i\|_2$.
6:     **end if**
7: **end for**
8: **Return** $D$.

---

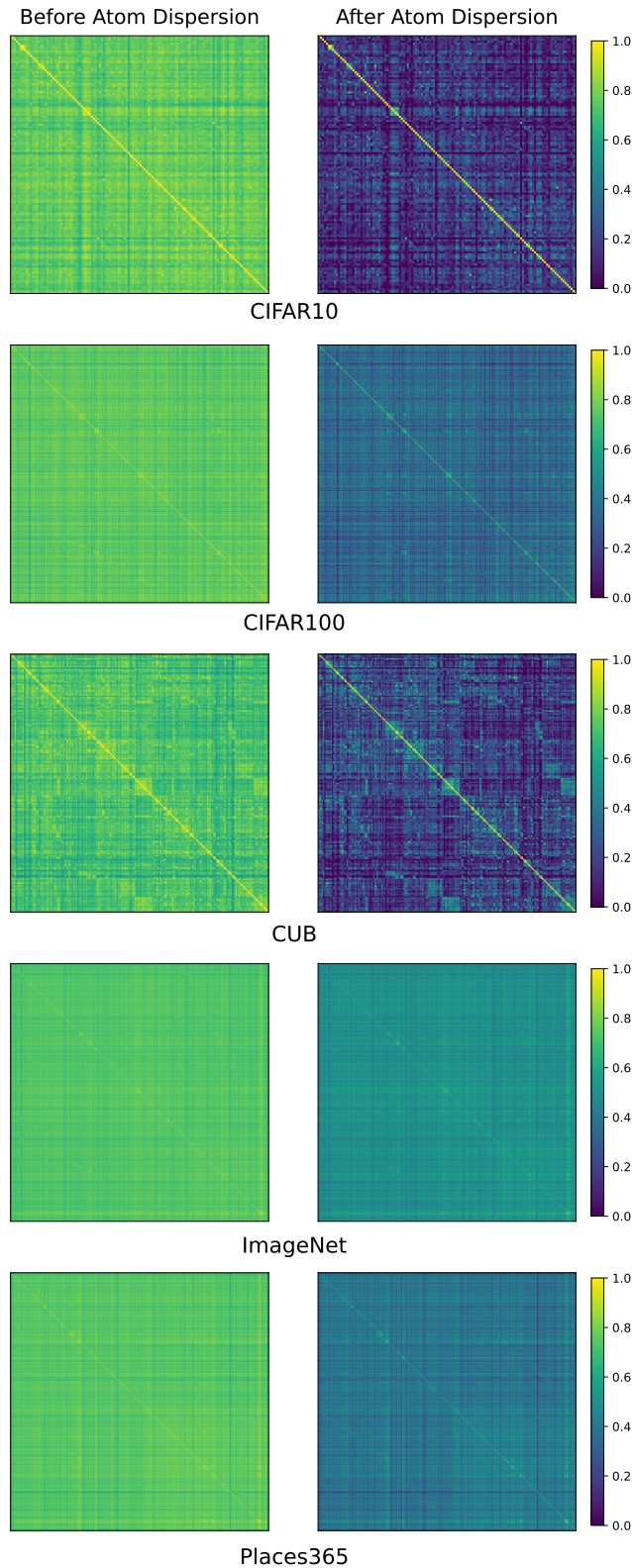

*Figure 5.* We calculate the correlation matrix ($\mathbf{D}^\top \mathbf{D}$) for dictionaries before and after Algorithm 3, and present them in the format of heatmaps. As can be seen, the proposed dispersion process effectively reduces the correlation between concept embeddings generated by CLIP.

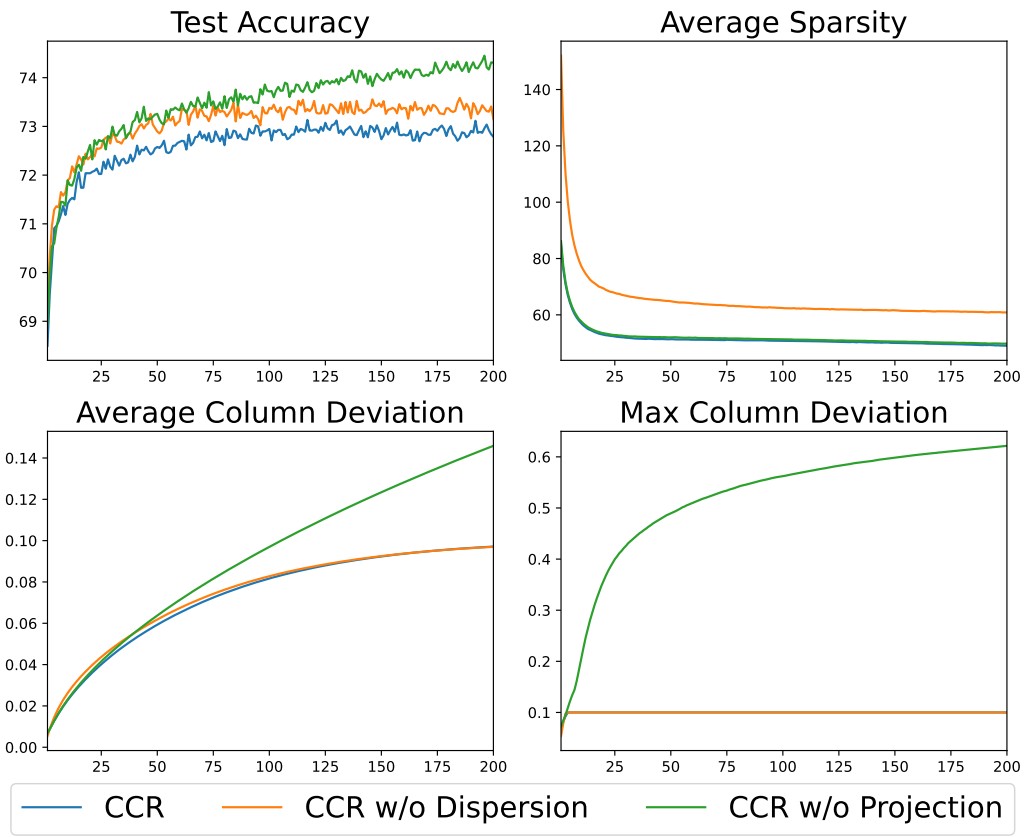

*Figure 6.* Ablation study for Algorithm 3 and Algorithm 4.

### D.3. Ablation Study

In this section, we present empirical findings from our ablation study. As a preliminary note, we emphasize that the comparison between CCR and its baseline shown in Figure 2 serves as a key ablation analysis for the CCR module as a whole. Here, we concentrate on evaluating the contributions of two critical components introduced in the preceding section: **Atom Dispersion** and **Atom Projection**. To this end, we individually exclude these two steps—Algorithm 3 and Algorithm 4—and compare the resulting models to the full CCR framework in terms of test accuracy, average sparsity, average column deviation, and maximum column deviation over the course of training.

As illustrated in Figure 6, the omission of either step leads to an improvement in performance. However, these gains incur different trade-offs. Specifically, removing Algorithm 3 results in a substantial increase in the average sparsity of the learned sparse codes. In contrast, eliminating Algorithm 4 causes the concept atoms to drift significantly from their initial CLIP embeddings, thereby compromising the interpretability of the model.

### D.4. Hyperparameter Tuning

This section presents an empirical evaluation of two critical hyperparameters—namely, the hard-threshold parameter $\lambda$ and the radius bound $\rho$—which significantly influence the performance of Algorithm 2. The corresponding results are illustrated in Figure 7 and Figure 8.

As depicted in Figure 7, increasing the value of $\lambda$ results in sparser explanations by reducing the number of activated concepts. However, this sparsity comes at the cost of reduced test accuracy. Therefore, selecting an appropriate threshold entails balancing the trade-off between prediction accuracy and the level of explanation sparsity—a consideration of particular relevance in explainable AI applications.

It is important to note that including more concepts does not inherently ensure improved accuracy. This point is exemplified

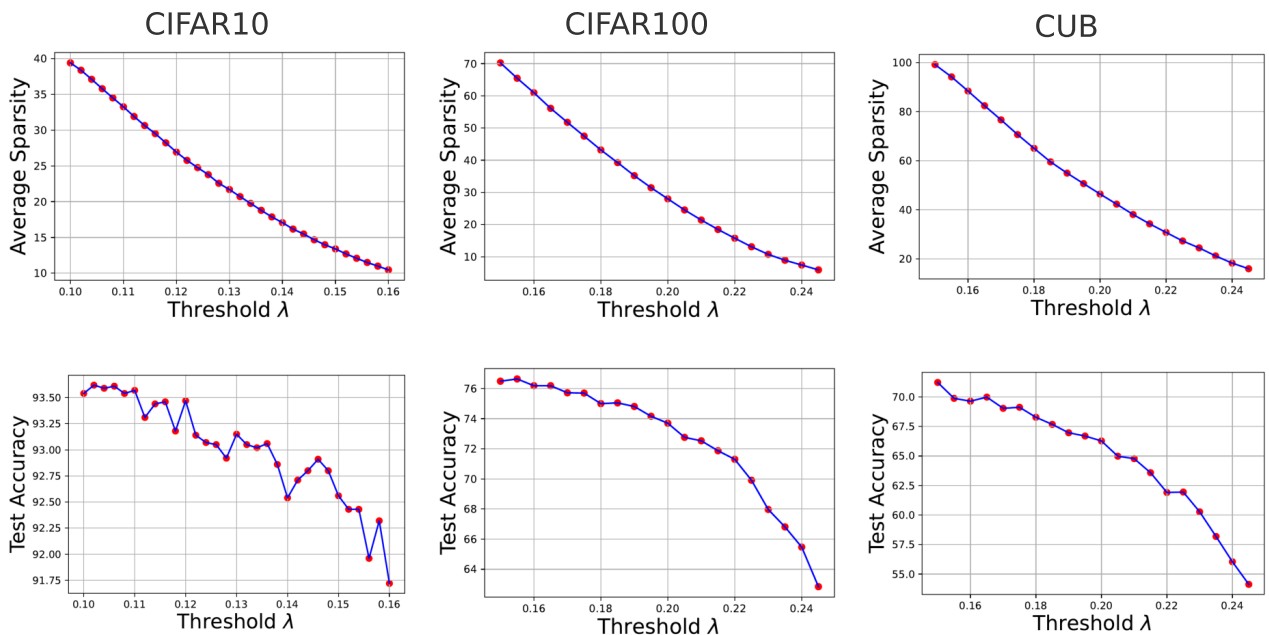

*Figure 7.* Average sparsity and test accuracy for varying thresholds $\lambda$.

in the left column of Figure 8, which shows that expanding the search radius enables the CCR algorithm to identify solutions that are simultaneously sparser and more accurate. Furthermore, the empirical results suggest that the performance gains of CCR saturate beyond $\rho > 0.1$, indicating that $\rho = 0.1$ serves as a suitable choice for this hyperparameter.

## D.5. More Experiments on Interpretability

In this section, we expand on the discussion of CCR's interpretability from Section 4.2 by conducting a comprehensive case study across all five datasets used to evaluate CCR. For each dataset, we select three representative image-label pairs and present the ten highest-ranking concept scores on the left, alongside their corresponding weights associated with the predicted label on the right. To facilitate a comparison of explainability with and without CCR, we also report the corresponding results for the baseline model, which is obtained by setting $\eta_{\mathbf{D}} = 0$ in Algorithm 2, presented at the end of this section. Based on our case study, we derive several key observations:

- **Semantic correlation.** In most cases, concepts with higher concept scores exhibit a strong semantic correlation with the input images. This finding substantiates the reliability of our algorithm as an interpretable AI model. A notable example is illustrated in Figure 16 and Figure 17, where the algorithm effectively identifies the primary distinguishing features between two visually similar bird species—their coloration—leading to an accurate classification.

- **Weight distribution.** In instances where misleading elements are present in an image, certain concepts unrelated to the ground truth label may receive high concept scores. Nevertheless, the linear layer appropriately assigns a small weight to such concepts. These weights encapsulate the algorithm's interpretation of the label, thereby rendering it comprehensible to human users. Examples of such cases include "a beak" in Figure 10, "a plow" in Figure 12, "a seagull"/"a bird" in Figure 14, "a highlighter" in Figure 18, "iridescent" in Figure 19, and "abandoned buildings"/"a sunset" in Figure 21.

- **Debugging capability for incorrect predictions.** Inevitably, our framework makes incorrect predictions for some samples. A key application of explainable AI is to help human experts understand the reasoning behind these errors. Our findings indicate that the proposed algorithm effectively captures interpretable misconceptions introduced by encoders. For instance, in Figure 22, concepts such as "a display case" and "exhibits" suggest that the primary misleading factor causing the model to predict "natural history museum" instead of the correct label "shopping mall" is the manner in which the store displays its goods. This insight is valuable for human intervention and model correction.

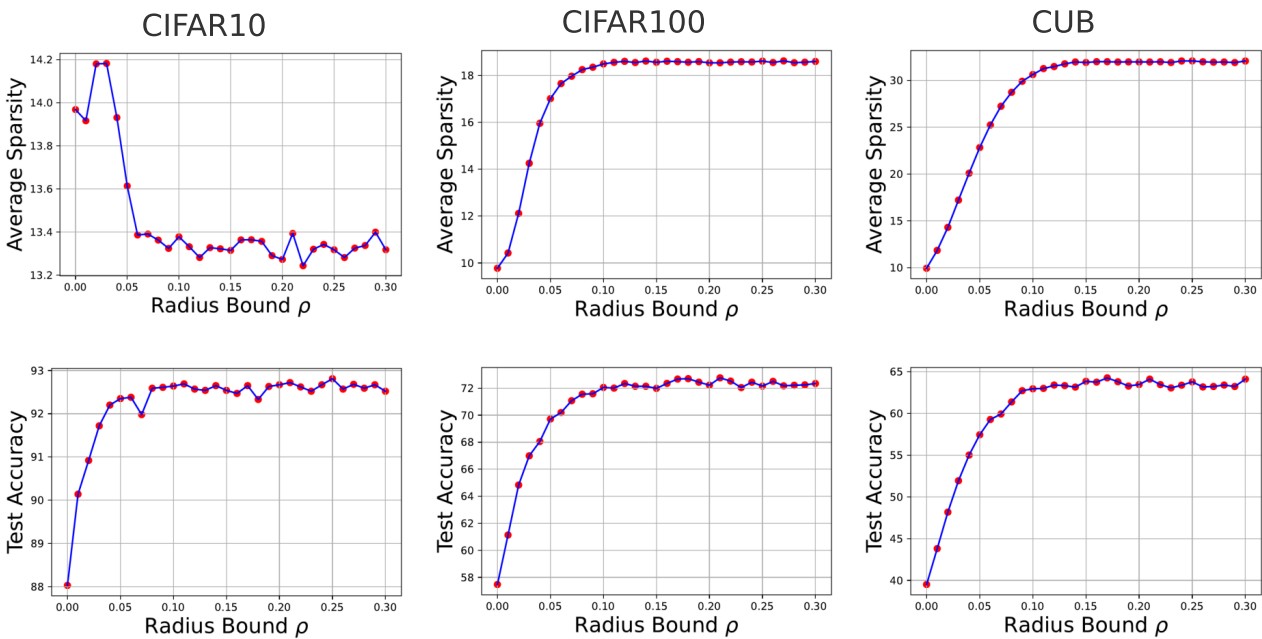

*Figure 8.* Average sparsity and test accuracy for varying radius bounds $\rho$.

- **Reliance on the richness of the concept set.** The effectiveness of the algorithm is inherently influenced by the quality of the concept set, a phenomenon that aligns with the fundamental motivation behind explainable AI. As demonstrated in Figure 23, the concept set lacks distinctive features that differentiate between "bus interior" and "train interior". Consequently, the model is unable to distinguish between these two labels.

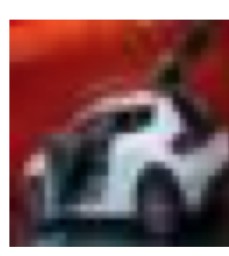
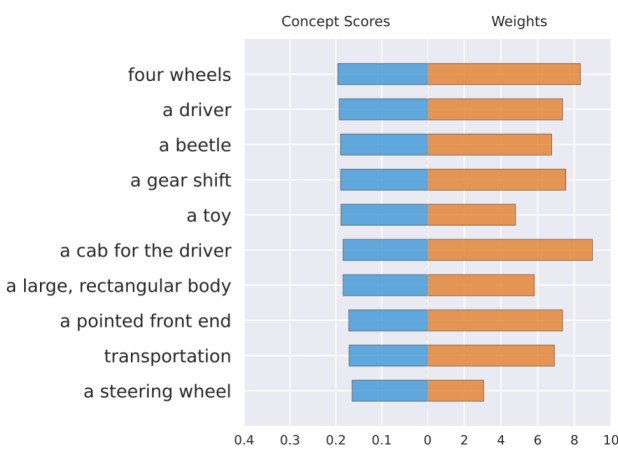

True label: automobile
Predicted label: automobile

*Figure 9.* CIFAR-10 (a)

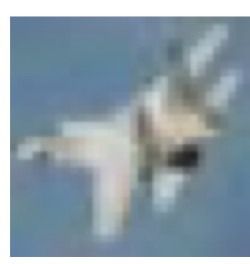
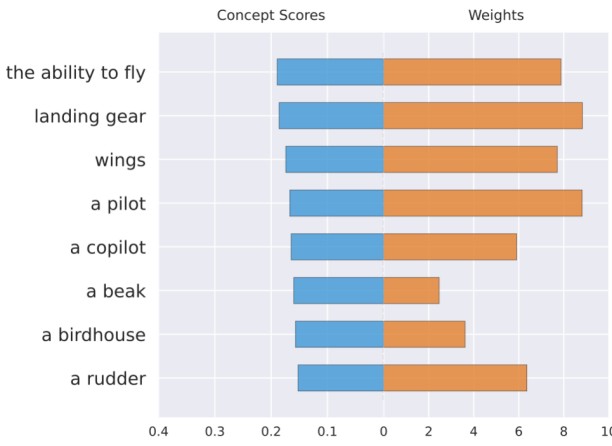

True label: airplane
Predicted label: airplane

*Figure 10.* CIFAR-10 (b)

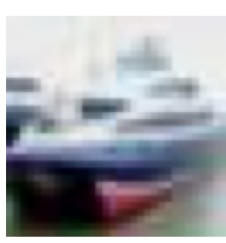
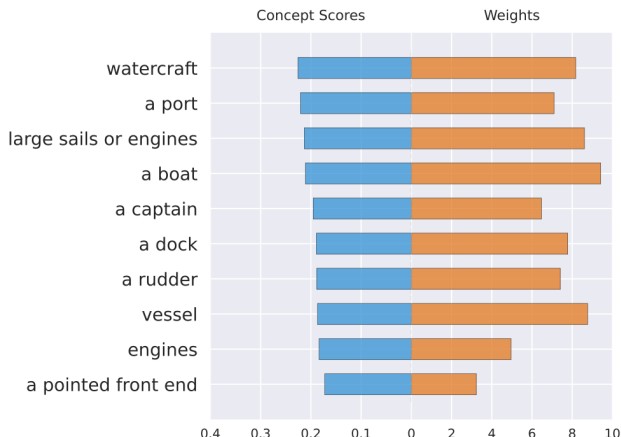

True label: ship
Predicted label: ship

*Figure 11.* CIFAR-10 (c)

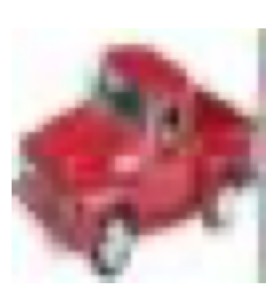

True label: pickup truck
Predicted label: pickup truck

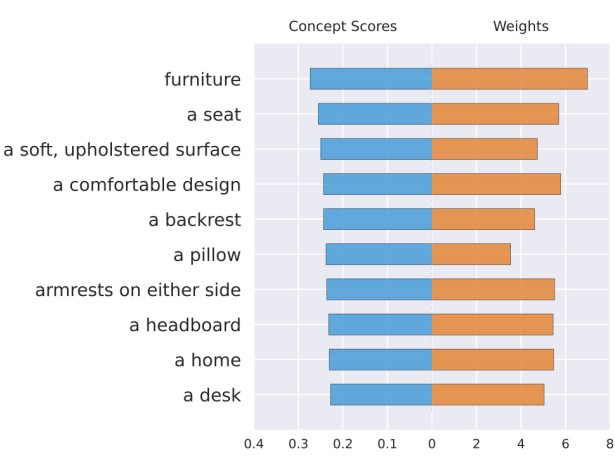

*Figure 12.* CIFAR-100 (a)

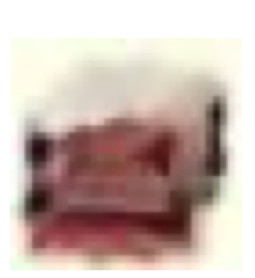

True label: couch
Predicted label: couch

*Figure 13.* CIFAR-100 (b)

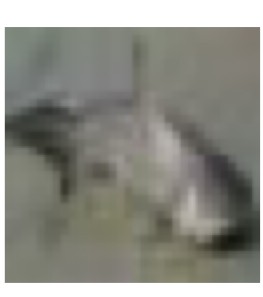

True label: shark
Predicted label: shark

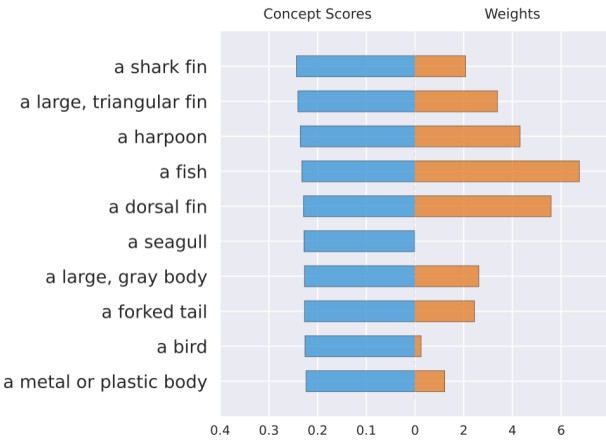

*Figure 14.* CIFAR-100 (c)

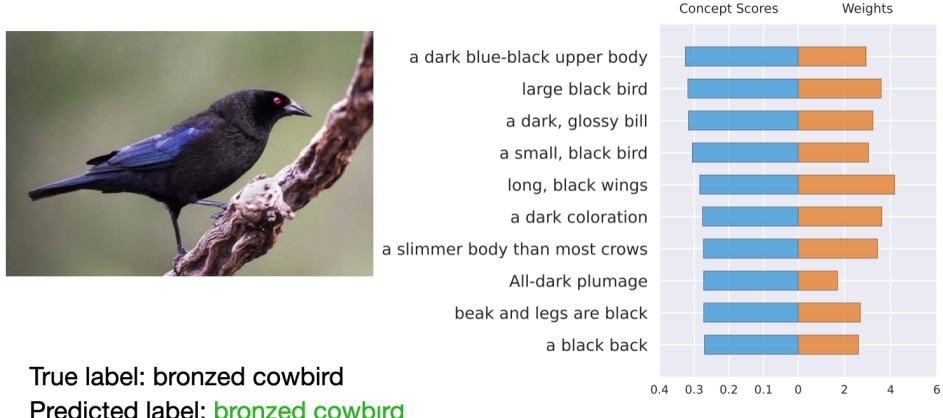

*Figure 15.* CUB-200 (a)

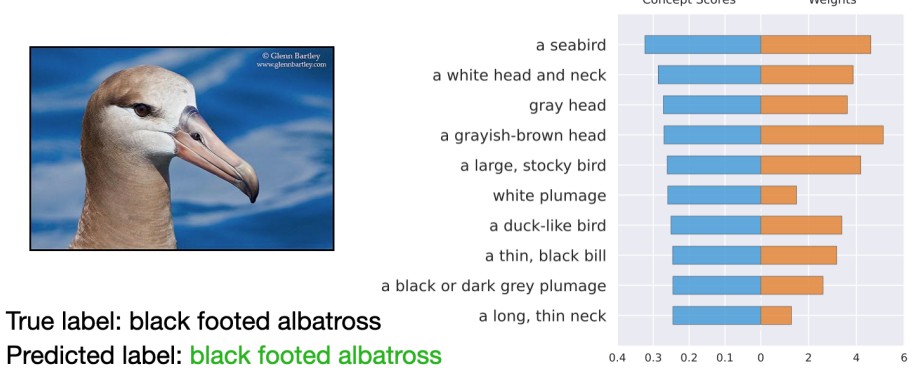

*Figure 16.* CUB-200 (b)

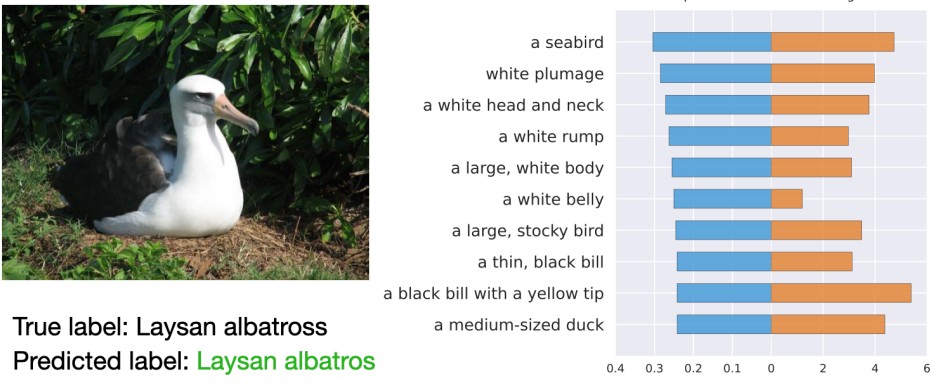

*Figure 17.* CUB-200 (c)

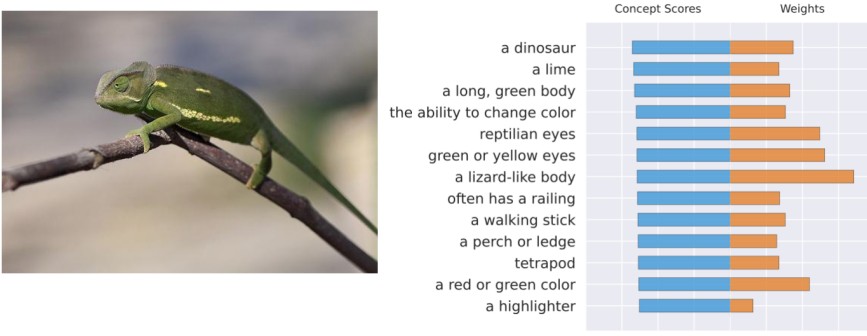

True label: African chameleon
Predicted label: African chameleon

*Figure 18.* ImageNet (a)

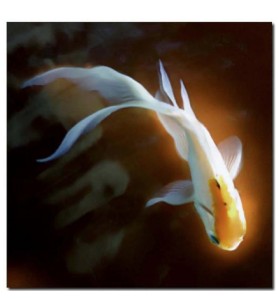

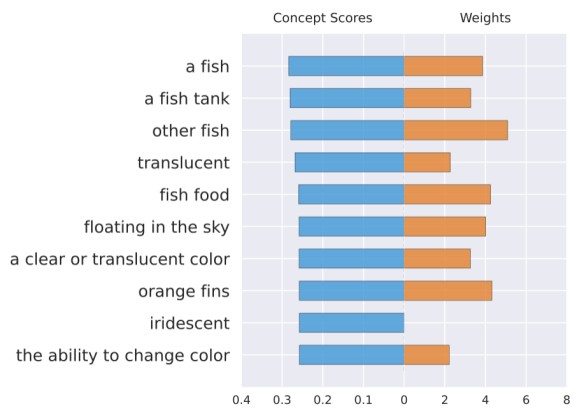

True label: goldfish
Predicted label: goldfish

*Figure 19.* ImageNet (b)

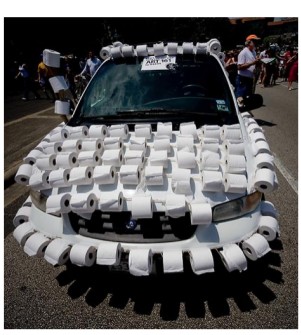

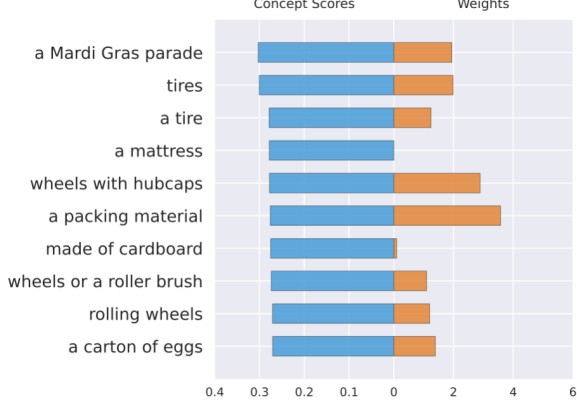

True label: toilet tissue
Predicted label: car wheel

*Figure 20.* ImageNet (c)

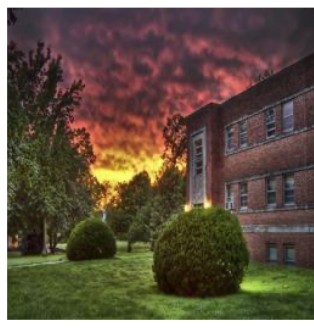

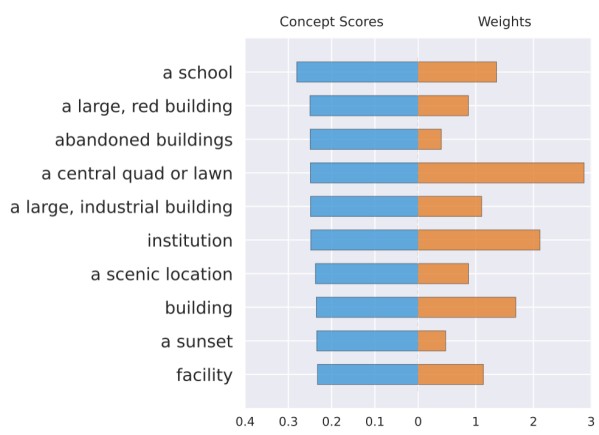

True label: campus
Predicted label: campus

*Figure 21.* Places365 (a)

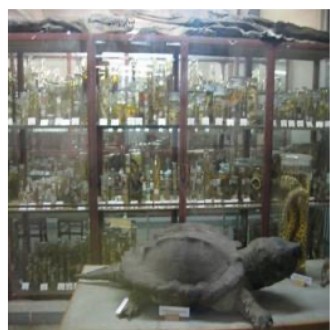

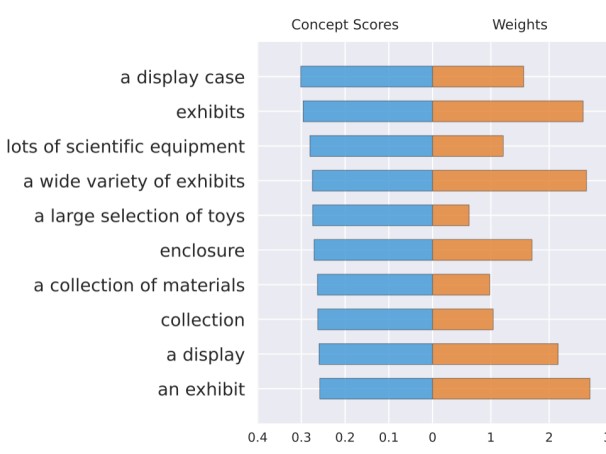

True label: shopping mall
Predicted label: natural
history museam

*Figure 22.* Places365 (b)

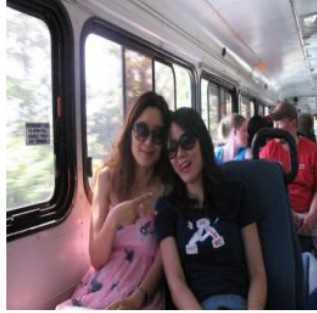

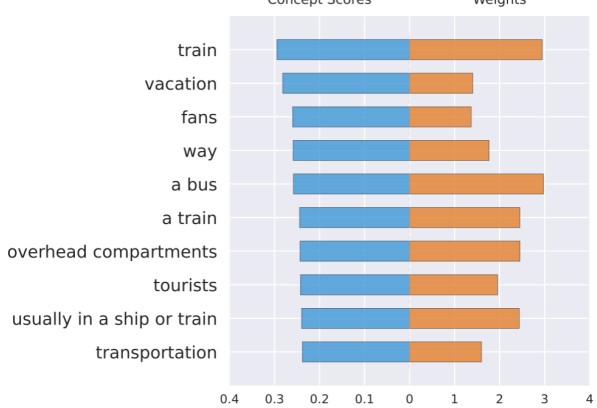

True label: bus interior
Predicted label: train interior

*Figure 23.* Places365 (c)

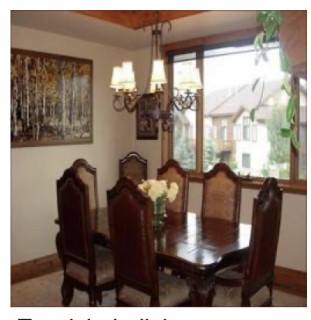

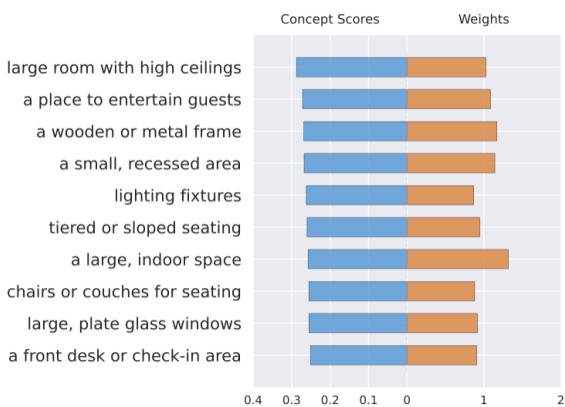

True label: dining room

Predicted label: dining room

*Figure 24.* Baseline comparison to the top figure in Figure 3

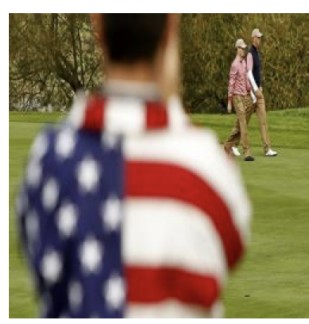

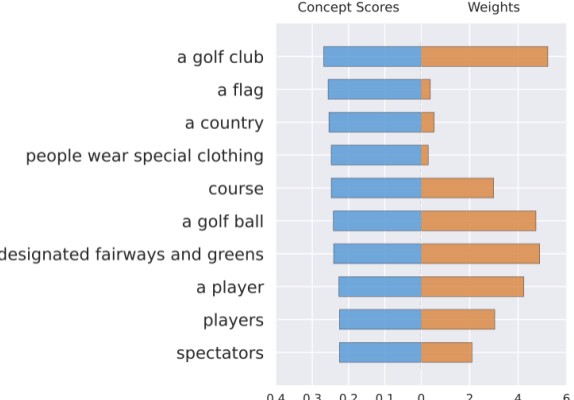

True label: golf course

Predicted label: golf course

*Figure 25.* Baseline comparison to the bottom figure in Figure 3

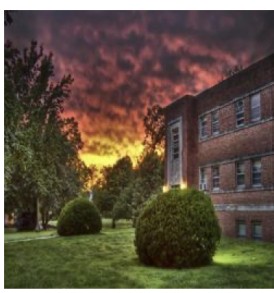

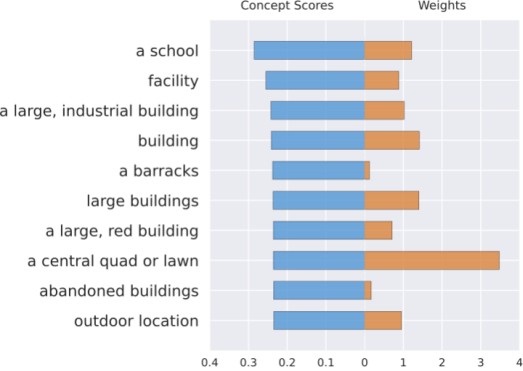

True label: campus
Predicted label: campus

*Figure 26.* Baseline comparison to Figure 21

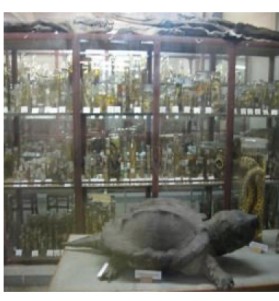

True label: shopping mall
Predicted label: natural
history museam

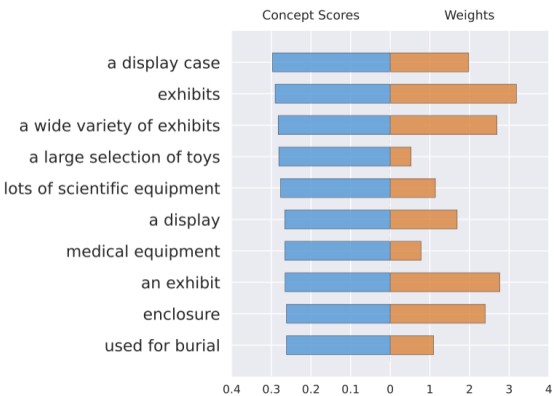

*Figure 27.* Baseline comparison to Figure 22

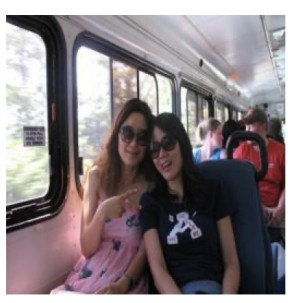

True label: bus interior
Predicted label: bus interior

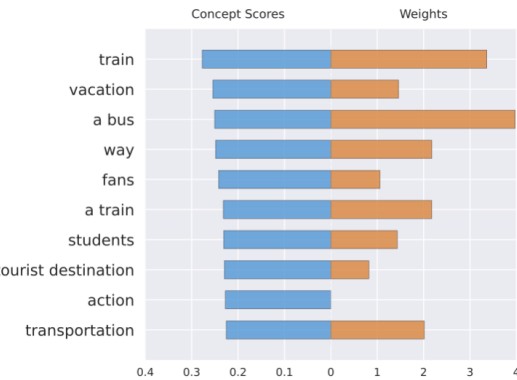

*Figure 28.* Baseline comparison to Figure 23

