# OpenReview forum: "Enhancing Performance of Explainable AI Models with Constrained Concept Refinement"
_ICML.cc/2025/Conference — ICML 2025 poster_

### Official Review · Reviewer_grrY · 2025-02-27

**Overall Recommendation:** 3

**Summary:**

This paper proposes to tackle the potential gap between performance and understandability of concept embedding models, but allowing the concept embeddings to be refined in a constrained manner, thereby allowing for more flexibility, but still understandability. The authors have proposed a theoretical framework embedded in the IP-OMP framework and ran several experiments.

**Claims And Evidence:**

Honestly I find it very difficult to even identify the claims of the paper. It starts with an introduction that is well written and easy to follow, though it is somewhat generic in that it does not become clear what the authors are specifically tackling and why this is relevant. Section 2 and section 3 is then pretty much buried in mathematical statements without a high-level explanation of what this is for. The experiments look promising, but their significance falls somewhat short in how they are tied to the authors claims (which I am still not 100% I have identified). I believe the claims are: "concept embedding models should allow for refining their embeddings, but these should only be refined in a small constrained manner."
In this context I think having a concrete example throughout the paper would really help understanding the problem and relevance of the problem in the first case, and what the method does.
Another question I have is what the authors mean by "concept embeddings" related to previous work. It would be good if the authors define this either in the introduction or section 2

**Essential References Not Discussed:**

See above.

**Experimental Designs Or Analyses:**

This seems fine for me, however I would suggest to move some of the experimental details into the appendix in order to make room for embedding and discussing the experiments more in the context of the authors claims.

**Methods And Evaluation Criteria:**

Please see my remarks in the previous box.
In addition, it is not clear why the authors frame the problem within the framework of section 2, i.e. "predicting a random variable by sequentially selecting a list of related random variables, termed queries, and checking their values." However, this seems quite central to the work and more clarification would be necessary here.

**Other Comments Or Suggestions:**

Please write a discussion and conclusion of your work. This makes it additionally difficult to understand what the paper is about. Overall the authors should rework the overall presentation of their work. What is the goal? Maybe add a running example. How does this relate to previous work? What is the conclusion/what have we learned from each section?

**Other Strengths And Weaknesses:**

The experimental results look valuable, but it is hard to understand them without understanding the claim and importance of the underlying problem.

**Questions For Authors:**

n/a

**Relation To Broader Scientific Literature:**

This is a section that is quite missing in the paper. I really think it would help to move this from the supplements to the main paper.
But in addition to this remark, I believe the section is very sparse, the majority of the referenced papers is repeatedly from Chattopadhyay et al.. I consider it important to add references from the field of XAI, concept learning and dictionary learning to increase the breadth here, but also make it more understandable how this work relates to them.

**Theoretical Claims:**

I have not been able to check the correctness of the theoretical claims.

---

> ### Author Rebuttal · Authors · 2025-04-01
>
> We sincerely thank the reviewer for the time and effort dedicated to reading our manuscript and for providing valuable and constructive feedback. In response to the concerns raised regarding the clarity and coherence of the paper’s central ideas, we propose to make major revisions to the manuscript. The details of these modifications are outlined in the subsequent response. We hope that these changes will enhance the readability of our work and allow the reviewer to better appreciate the contributions presented in this paper.
>
> **Claims And Evidence**
>
> In response to the reviewer’s comments and upon careful reconsideration, we acknowledge that there exists a logical gap between the introductory section and the theoretical exposition, which may hinder the reader’s ability to follow the overall narrative. At a high level, the reviewer's understanding is accurate: *the principal contribution of this paper is that we propose refining the representations of concepts in interpretable AI models within a constrained deviation, thereby enabling the resulting model to achieve a more favorable trade-off between performance and interpretability.*
>
> To address the reviewer’s suggestions and improve the clarity of the manuscript, we will include the following paragraph at the end of the first section:
>
> On a high level, our main contribution is the introduction of a framework in which concept embeddings in explainable AI models are refined within a restricted parameter space to attain a better balance between predictive performance and interpretability. The remainder of the paper substantiates this claim along two key dimensions:
>
> 1. **Theoretical validation.** We demonstrate that our method is both **necessary** (Section 2.1) and **effective** (Section 3) under the theoretical framework introduced by [1], whose background is reviewed in Section 2. This framework serves as a suitable testbed for our contributions for two main reasons: (1) Among interpretable-by-design models [2-6], only [1] presents a generative model wherein both performance and interpretability are rigorously defined. (2) The algorithm motivated by this generative framework achieves state-of-the-art results within the class of interpretable AI methods.
>
> 2. **Empirical evaluation.** We conduct experiments on multiple benchmark datasets for image classification tasks to assess the practical effectiveness of our approach (Section 4).
>
> **Relation To Broader Scientific Literature**
>
> We thank the reviewer for their constructive suggestions regarding the improvement of the literature review section. We would also like to respectfully clarify that, given the close relationship between the motivation of our work and prior literature in explainable AI (XAI), we dedicate the majority of Section 1 to introducing the relevant background. Appendix A serves as a complementary component, offering a more in-depth discussion of works that are particularly closely aligned with our approach. Upon careful review, we believe that we have sufficiently highlighted and discussed the related literature in the areas of XAI, concept learning, and dictionary learning. If the reviewer has specific papers in mind, we would be happy to consider including them in the revised manuscript.
>
> **Other Comments Or Suggestions**
>
> Based on the reviewer's suggestion, we will add an additional section by the end of the paper in the revised manuscript. Due to the character limitations for each rebuttal, please see the first section in our response to Reviewer LpAZ for the full text.
>
> Reference:
>
> [1] A Chattopadhyay, R Pilgrim,R Vidal. Information maximization perspective of orthogonal matching pursuit with applications to explainable AI.
>
> [2] B Zhou, Y Sun, D Bau, A Torralba. Interpretable basis decomposition for visual explanation.
>
> [3] PW Koh, T Nguyen, YS Tang, S Mussmann, E Pierson, B Kim, P Liang. Concept bottleneck models.
>
> [4] M Yuksekgonul, M Wang, J Zou. Post-hoc concept bottleneck models
>
> [5] T Oikarinen, S Das, L Nguyen, T Weng. Label-free concept bottleneck models
>
> [6] A Chattopadhyay, K Chan, B Haeffele, D Geman, R Vidal. Variational information pursuit for interpretable predictions.

---

### Official Review · Reviewer_b4dR · 2025-03-08

**Overall Recommendation:** 3

**Summary:**

In this paper, the author has proposed an improvement for the original structure of the concept bottleneck models and CLIP-IP-OMP models for improving the interpretability and trustworthiness of the model without sacrificing the model's accuracy. The key idea of this method is trying to optimize the concept embeddings while keeping them within a constrained range, which addresses the issue of the inaccurate query features that are generated from the dataset like CLIP. The key contributions of this paper include the theoretical justifications of the necessity of the problem, the accuracy and interpretability guarantees, and empirical experiment results that show its performance compared to the existing methods.

**Claims And Evidence:**

In this paper the authors provides several claims and are mostly providing theoretical proofs that shows the necessity of concept refinement, and the CCR overcomes the problem. However, I am a little bit concerned about the authors' claim of improving the interpretability statement using CCR.

I understand the author's claim about enhanced interpretability through the introduction of the function $\rho$ in the model. However, for a concept bottleneck model or the similar concept-based models, the original input space/model has already provided sparse explanations. In other words, the output that the models like CILP provide is already sparse, so with a modification of $\Delta D$, it cannot guarantee that the level of interpretability (I am not sure if that's means sparsity in this case) is changed. I would like to see if the authors can better define what interpretability means here, as I can see the introduction of the residual part more correlated to the definition within the robust regression [1]. I can see that the results/definition is aiming for a robust solution to me, rather than saying that it is interpretable, and the related works that the authors states in line 210-219 about the sparse encoding also doesn't provides me an explanations on why the closer to the $D^*$, the more interpretable it would be. If the authors can provide me more explanations on that, that would really strengthen the claim.
[1] Xu, Huan, Constantine Caramanis, and Shie Mannor. "Robust regression and lasso." _Advances in neural information processing systems_ 21 (2008).

**Essential References Not Discussed:**

I don't think that there are essential references that are not discussed, but I have mentioned in the other strengths and weaknesses that asking about if the references about the robust regressions and their strategies should be included as the related works for the paper, as the problem statements seem to be similar to them.

**Experimental Designs Or Analyses:**

I think the proposed method CCR, is well-motivated for the problem of improving the accuracy of interpretable AI model while preserving its ability to explain the decisions. However, I am several concerns about the design of the experiments:

[Sparsity and $\rho$ trade-off]  In CCR, the constraints $\rho$ determines how the concepts embeddings can be refined. Therefore, it is actually risky when the $\rho$ goes larger. Therefore, I am wondering if the authors have experiments to show the relationship between the sparsity and the selection of the $\rho$ is real-world applications since it seems a bit hard to tune comparing to setting a strict top $k$ in the sparse autoencoder settings. I am wondering if the authors can provides more guidelines on how the accuracy and sparsity ($ASR$) have changed through the change of $\rho$.

[Missing baselines Metrics] In the experiments in table 1, the authors have shown the average explanations length, sparsity, and the concept embedding deviations for the CCR, However, there missed a baseline here on how without CCR would influence those metrics. As the results without CCR is already sparse, so it is not the case that we can regard 100% as the baseline. I am wondering if the authors could provide the baseline/comparisons for other methods as a guidelines in the table so that we can see how the ASR/AEL/ACED have changed through the optimization. How much it has sacrificed/improved from the sparsity level perspective. This comparison missing in the appendix sesssion where it is hard to tell the improvment for interpretability in Appendix D.3.

**Methods And Evaluation Criteria:**

Yes, the proposed methods and evaluation criteria make sense for the problem and applications. I am a little bit concerned about the interpretability metrics that the authors used both in the paper and the criteria, as they are mostly the case study instead of quantitative metrics. I have more comments in the experiment design sections for the concerns about the methods comparison.

**Other Comments Or Suggestions:**

No extra comments.

**Other Strengths And Weaknesses:**

I am a little bit concerned about the strategies that the author has proposed that can be aligned with the robust regression fitting problem [1]. There are several current works for using the perturbation idea for solving similar issue but not just to the concept-based problems [2].
[1] Xu, Huan, Constantine Caramanis, and Shie Mannor. "Robust regression and lasso." _Advances in neural information processing systems_ 21 (2008).
[2] Su, Peng, et al. "CR-lasso: Robust cellwise regularized sparse regression." Computational Statistics & Data Analysis 197 (2024): 107971.

**Questions For Authors:**

If the authors can provide better explanations/definitions for the interpretability before and after the concept refinement, that would better show the usefulness of this approach.

**Relation To Broader Scientific Literature:**

The contributions of the paper related to the broader scientific literature include:

[Concept-based Model Construction] The paper builds on Concept Bottleneck Models (CBMs)[1], which introduced the idea of human-interpretable intermediate representations. In CBMs, models first predict a set of predefined datasets and then use them to make final decisions. However, a major issue with CBMs is their reliance on imperfect concept annotations, leading to performance degradation when concepts are noisy or biased. CCR directly addresses this issue by refining concept embeddings while ensuring they remain interpretable, whereas CBMs typically assume fixed, pre-defined concepts.

[Theoretical proofs for the misaligned/adversarial attack leading to the high performance drop] Prior work [2] showed that explainability methods are often fragile to small changes in feature representations. CCR mitigates this fragility by ensuring that concept refinements remain interpretable.

[1] Koh, Pang Wei, et al. "Concept bottleneck models." International conference on machine learning. PMLR, 2020.
[2] Ghorbani, Amirata, Abubakar Abid, and James Zou. "Interpretation of neural networks is fragile." Proceedings of the AAAI conference on artificial intelligence. Vol. 33. No. 01. 2019.

**Theoretical Claims:**

I did check the correctness of the proofs for the theoretical claims. I have no specific questions for them.

---

> ### Author Rebuttal · Authors · 2025-04-01
>
> We sincerely thank the reviewer for their detailed feedback and positive assessment. We hope that our clarifications and additional experiments could further elevate their recommendation of our work in the final review.
>
> **Claims And Evidence**
>
> We would like to offer two clarifications that may help resolve the confusion expressed by the reviewer:
>
> 1. In this work, we evaluate the performance of CCR in two distinct settings: the theoretical model proposed by [1], and the interpretable image classification task. Interpretability can be quantitatively assessed only in the former setting; for applications such as image classification, the community currently lacks universally accepted metrics for interpretability. As a result, most related works instead rely on post-hoc, case-specific analyses to illustrate interpretability [2,3,4,5]. These analyses typically involve examining the predicted output and the most salient concepts identified by the model, and assessing whether these concepts provide a human-interpretable explanation for the output. Accordingly, our claim of *improved interpretability* applies solely to the theoretical setting. In both settings, it is important to note that sparsity does not equate to interpretability. We have proposed a revision to the manuscript to make this distinction explicit; please refer to our response to Reviewer grrY for further details.
>
> 2. In the theoretical setting, the input $\mathbf{x}$ is generated as a sparse linear combination of the columns of ground truth (but unknown) $\mathbf{D}^*$. Therefore, perfect interpretability in this setting corresponds to the exact recovery of $\mathbf{D}^*$, and not sparsity. In this simplified setting, improvements in accuracy and interpretability are aligned, as demonstrated by Theorem 3.4.
>
> **Experimental Designs Or Analyses**
>
> [Sparsity and $\rho$ trade-off]: We have included plots that shows the relationship between the performance/sparsity and $\rho$ for CIFAR-10 dataset through the following [anonymous link](https://anonymous.4open.science/r/ICML_Review-C283) (Plot 4 and 6). We will add the same experiment for all other datasets in the revised manuscript.
>
> [Missing baselines Metrics]: As mentioned above, sparsity is not used as a metric for interpretability in this work. Accordingly, metrics such as ASR, AEL, and ACED are not intended to reflect interpretability, but rather to highlight the simplicity and efficiency of our method relative to existing approaches. These metrics demonstrate that our method significantly reduces computational cost compared to state-of-the-art methods, while simultaneously achieving improved accuracy. Nonetheless, we appreciate and agree with the reviewers' suggestion that the interpretability of the baseline models needs examination. Accordingly, we will include a corresponding analysis—similar in nature to Appendix D.3—for the baseline models in the revised manuscript.
>
> **Other Strengths And Weaknesses**
>
> We appreciate the reviewer's insightful observation regarding the connection between our problem and robust regression. Using Lasso as an illustrative example, it is indeed equivalent to solving a sparse coding problem, where the goal is to represent a signal $\mathbf{x}$ using a small number of columns from a dictionary matrix $\mathbf{D}$.
>
> Nevertheless, there are two key distinctions between our setting and that of Lasso:
>
> 1. Similar to sparse coding problem, in Lasso, it is assumed that $\mathbf{D}$ is fixed and known, and the primary goal is to promote sparsity in the reconstruction. In contrast, our setting does not assume that $\mathbf{D}$ is fixed; rather, we treat the given $\mathbf{D}$ (typically obtained from CLIP) as uncertain and potentially erroneous. Our objective is to co-optimize both $\mathbf{D}$ and the reconstruction. This formulation is more closely related to dictionary learning, a connection we discuss extensively in the paper.
>
> 2. The primary objective of Lasso is to identify a sparse code $\mathbf{s}$ such that $\|\mathbf{x} - \mathbf{D}\mathbf{s}\|_2$ is minimized. In our framework, however, the emphasis is not solely on reconstruction accuracy, but on using the resulting $\mathbf{s}$ to improve performance on downstream tasks.
>
> [1] A Chattopadhyay, R Pilgrim,R Vidal. Information maximization perspective of orthogonal matching pursuit with applications to explainable AI.
>
> [2] PW Koh, T Nguyen, YS Tang, S Mussmann, E Pierson, B Kim, P Liang. Concept bottleneck models.
>
> [3] M Yuksekgonul, M Wang, J Zou. Post-hoc concept bottleneck models
>
> [4] T Oikarinen, S Das, L Nguyen, T Weng. Label-free concept bottleneck models
>
> [5] A Chattopadhyay, K Chan, B Haeffele, D Geman, R Vidal. Variational information pursuit for interpretable predictions.

---

> > ### Comment · Reviewer_b4dR · 2025-04-05
> >
> > Thank you for the clarification — it addresses all my concerns. I will keep the score unchanged

---

### Official Review · Reviewer_1v3q · 2025-03-19

**Overall Recommendation:** 4

**Summary:**

This paper addresses the challenge of balancing accuracy and interpretability in machine learning models, particularly for interpretable-by-design methods that often sacrifice accuracy for transparency. The authors identify that deviations in concept representations, a crucial component of interpretable models, can negatively impact prediction performance. To counter this, they propose a novel framework called Constrained Concept Refinement (CCR) that optimizes concept embeddings under constraints to preserve interpretability. CCR significantly improves the prediction accuracy of interpretable models like IP-OMP on image classification tasks (CIFAR 10/100, ImageNet, CUB200, and Places365) while maintaining interpretability and achieving a reduction in runtime compared to other explainable methods.

## update after rebuttal
I appreciate the thoughtful and detailed responses from the authors, which successfully addressed all my concerns. Consequently, I have decided to increase my score for the paper, as I now believe it holds significant potential to contribute meaningfully to the field of explainable AI. I also hope that, if the paper is accepted, the final version will incorporate the additional results and address reviewer suggestions to further strengthen its impact.

**Claims And Evidence:**

The claims made in the submission are generally supported by clear and convincing evidence, but several key assertions have limitations or rely on restrictive assumptions.

1. Assumption of Column-Orthogonality and Full Rank : The theoretical guarantees rely heavily on assumptions such as column-orthogonality and full-rank latent feature matrices (e.g., Assumption 3.2). These assumptions may not hold in realistic scenarios where concept embeddings often exhibit correlations or redundancy. The authors acknowledge this limitation briefly but do not empirically investigate how violations of these assumptions affect performance in practice.

2. Dependency on Pre-trained Models (CLIP) : The method relies significantly on pre-trained CLIP embeddings, which introduce biases from large-scale internet data used for training CLIP. The authors acknowledge potential biases but do not empirically analyze how these biases might impact CCR's reliability across diverse datasets or domains.

3. Interpretability Preservation : While qualitative examples (Figures 3, 5–19) show interpretable concepts, there is no quantitative metric for interpretability (e.g., human evaluation scores or concept alignment metrics). Interpretability is subjective, but the paper relies on only ad-hoc visualizations without rigorous validation.

4. Hyperparameter Choice : The paper mentions the correction radius ρ as a crucial hyperparameter that controls the trade-off between accuracy and interpretability. The process for selecting the optimal value for this and other hyperparameters is not detailed extensively, and the sensitivity of the results to these choices could be further explored. Also, optimal ρ likely varies across datasets, but the paper provides no guidance for tuning it.

5. Concept Set Dependency : Performance and interpretability depend heavily on the predefined concepts, which are treated as a black box. Here the concept set is generated using GPT-3, which may introduce bias (e.g., cultural or linguistic limitations). The paper does not analyze how concept quality affects CCR.

**Essential References Not Discussed:**

The authors has added critical references or related works that are essential to understanding the context of its key contributions.

**Experimental Designs Or Analyses:**

The experimental designs and analyses in this paper are generally sound and appropriate for evaluating CCR in image classification tasks. However, there are some limitations that should be addressed to strengthen the validity and generalizability of the findings.

1. The paper does not report statistical significance tests for its results (e.g., accuracy improvements over baselines). This omission makes it difficult to assess whether the observed improvements are robust across different runs or datasets.

2. The paper could benefit from ablation studies to understand the contribution of different components of the CCR framework to the overall performance improvement.

**Methods And Evaluation Criteria:**

The proposed methods and evaluation criteria in the paper are generally well-suited to the problem of improving explainable AI (XAI) models, particularly in interpretable image classification. However, there are areas where the choices could be questioned or improved for broader applicability and robustness.

1. Interpretability Metrics: While AEL, ASR, and ACED are useful metrics, they do not fully capture human-centric notions of interpretability. For example, there is no user study or human evaluation to validate whether the explanations generated by CCR are genuinely understandable or actionable for end-users.

2. Limited Analysis of Misleading Concepts: Although qualitative examples highlight CCR's ability to handle misleading concepts (e.g., assigning low weights to irrelevant features), a systematic analysis of failure cases or edge cases is missing.

**Other Comments Or Suggestions:**

I don't have any specific comments or suggestions for the authors beyond what I already mentioned.

**Other Strengths And Weaknesses:**

The paper makes significant contributions to explainable AI by introducing Constrained Concept Refinement (CCR), which improves accuracy and interpretability while maintaining computational efficiency. Its originality lies in directly optimizing concept embeddings within constrained neighborhoods, a departure from prior methods that rely on additional black-box modules.

However, the paper's generalizability beyond image classification tasks, reliance on pre-trained models like CLIP, lack of human-centric interpretability validation, strong theoretical assumptions, and limited analysis of failure cases are notable weaknesses that should be addressed.

**Questions For Authors:**

Despite all these limitations mentioned before, the paper represents a substantial step forward in interpretable machine learning research, offering both theoretical insights and practical improvements that could inspire further advancements in the field. Hence I am leaning towards accepting this paper. But addressing these weaknesses would enhance its impact and generalizability, and improve the chance of acceptance from my side.

**Relation To Broader Scientific Literature:**

The paper contributes to the field by addressing a key challenge in interpretable AI, building upon existing interpretable-by-design methods, focusing on the crucial role of concept embeddings, proposing a novel constrained optimization framework with theoretical guarantees, and demonstrating its practical effectiveness in a relevant application domain.

**Theoretical Claims:**

The theoretical claims are largely correct within their stated assumptions and provide valuable guarantees for CCR's performance under idealized conditions. However, some assumptions (e.g., orthogonality, full rank) and practical considerations (e.g., sample complexity) limit their applicability to real-world scenarios. The proofs for Theorems 2.6, 3.3, and 3.4 are mathematically rigorous to me, but I did not check them thoroughly.

---

> ### Author Rebuttal · Authors · 2025-04-01
>
> We are sincerely grateful to the reviewer for their thorough evaluation and for the positive feedback on our work. In response to their suggestions, we have proposed additional experiments and provided further clarifications in the revised manuscript. We hope that our responses effectively address the concerns raised and that the reviewer would consider raising the score in the final assessment.
>
> **Assumption of Column-Orthogonality and Full Rank**
>
> We thank the reviewer for this insightful comment. We divide our response into two parts:
>
> Full-rank condition: This condition is fundamental and cannot be relaxed; it is not an artifact of our theoretical analysis. In particular, when the dictionary is not full-rank, our analysis shows that $\mathbf{D}$ may deviate from $\mathbf{D}^*$ in directions that are column-orthogonal to the column space of $\mathbf{D}^*$. Such deviations cannot be corrected by the algorithm. This behavior is further corroborated by numerical experiments. In Appendix D.1, we present synthetic data experiments that highlight the necessity of the full-rank condition. Specifically, the middle column of Figure~4 corresponds to a rank-deficient $\mathbf{D}^*$, and the results show that, in the absence of the full-rank assumption, the algorithm fails to achieve column-wise convergence. This outcome is fully consistent with the theoretical guarantees established in our main theorem.
>
> Orthogonality condition: In practical scenarios where the dictionaries are not orthogonal, we propose a novel procedure, termed *Concept Dispersion*, designed to mitigate the impact of highly correlated embeddings. The details of this sub-algorithm are provided in Appendix D.2. The effectiveness of it is shown in the Plot 3 from this [anonymous link](https://anonymous.4open.science/r/ICML_Review-C283), where we compare the correlation between concept embeddings before and after this process.
>
> **Dependency on Pre-trained Models**
>
> We would like to clarify that the introduction of CCR is specifically intended to address the errors, biases, and uncertainties that arise from CLIP embeddings. The reviewer is absolutely correct in noting that CLIP embeddings are often far from accurate. This observation is, in fact, the primary motivation behind the development of CCR in our work. In particular, we demonstrate—both empirically and theoretically—that while existing methods rely solely on the raw outputs of CLIP, without any mechanism for refining the embeddings, our proposed CCR framework not only leverages CLIP-derived embeddings but also further refines them to achieve strong performance within each specific domain.
>
> **Interpretability Preservation**
>
> We agree with the reviewer that large-scale human evaluation experiments would be valuable in demonstrating the interpretability of our method. This is indeed a limitation—not only of our work, but also of most existing efforts aimed at developing interpretable machine learning models. Addressing the reviewer’s comment and incorporating large-scale human evaluation would require substantial resources, which are currently beyond our reach. In the absence of such resources, our current interpretability evaluation is based on significantly smaller scale human evaluations on a set of randomly selected images, for which we report the model’s outputs and qualitatively demonstrate their interpretability. We are grateful for the reviewer’s constructive suggestion and will strive to incorporate more comprehensive evaluation practices in future work.
>
> **Hyperparameter Choice**
>
> For an empirical sensitivity analysis of $\rho$, we refer to the Plot 4 from this [anonymous link](https://anonymous.4open.science/r/ICML_Review-C283).
>
> **Concept Set Dependency**
>
> Our primary motivation for adopting the concept set generated by GPT-3 is to ensure fair comparisons between our method and existing approaches, as both lf-CBM and IP-OMP utilize the same concept set. The concern raised by the reviewer is indeed valid, as the quality of the concept set can significantly influence model performance—a consideration that applies to all interpretable-by-design algorithms. While the development of improved concept sets is undoubtedly important, we regard this as beyond the scope of the current work and leave it as a direction for future research.
>
> **Interpretability Metrics & Limited Analysis of Misleading Concepts**
>
> Please see our response above for Interpretability Preservation.
>
> **Statistical Significance Test**
>
> We thank the reviewer for their valuable suggestions. to better showcase the robustness of our method, we have enhanced our experimental protocol by performing 10 independent runs for each dataset and reporting the maximum, minimum, and average performance. See Plot 5 from this [anonymous link](https://anonymous.4open.science/r/ICML_Review-C283).
>
> **Ablation Studies**
>
> We thank the reviewer for this valuable suggestion and will include a ablation study in the revised manuscript.

---

> > ### Comment · Reviewer_1v3q · 2025-04-08
> >
> > I thank the authors for the detailed response, which has clarified my doubts. I appreciate the effort they put into addressing my concerns. However, there are still a couple of aspects that remain unclear:
> >
> > 1. The response does not mention any failure cases of the proposed method. Does this imply that no such cases were observed during your experiments?
> >
> > 2. Could you elaborate on your plans for including the ablation study? Specifically, what aspects of the method will be the focus of this analysis?
> >
> > I will maintain my current score.

---

> > > ### Author Response · Authors · 2025-04-08
> > >
> > > We are glad to hear that our response has addressed some of the reviewer’s concerns, and we thank the reviewer for the thoughtful follow-up questions. As these questions were raised on the final day of the author-reviewer discussion period, we will do our best to address them promptly and thoroughly below:
> > >
> > > **Failure cases:** Our method does indeed exhibit failure cases. In the context of our image classification experiments, failure can be characterized in two main ways: (i) misclassification and (ii) lack of interpretability.
> > >
> > > *Misclassification:*  We have already included several examples of misclassified images in the appendix (see Figures 16(c), 18(b), and 19(c)). We kindly refer the reviewer to these figures. While many similar examples exist, we believe the presented cases are representative and sufficient to demonstrate that, although our method outperforms existing approaches in terms of accuracy across most benchmarks, it is still susceptible to misclassification.
> > >
> > > *Lack of interpretability:* As discussed in our previous response under the section “Interpretability Preservation,” systematically identifying the causes of interpretability failures would require a more principled analysis—most likely involving large-scale human evaluation, as rightly suggested by the reviewer. While we currently lack the resources to conduct such a study, we will explicitly acknowledge this limitation in the impact statement of the revised manuscript.
> > >
> > > **Ablation study:** We appreciate the reviewer’s suggestion and apologize for the lack of clarity in our initial response. Conceptually, our approach comprises three distinct modules, each of which can be independently evaluated through ablation:
> > >
> > > *Concept refinement:* One of the key novelties of our method is the refinement of initial concepts via an optimization algorithm. An ablation study in this context would isolate the effect of concept refinement while keeping the other components fixed. This analysis has already been conducted in the main paper (although it was not explicitly labeled as an ablation study). Specifically, the blue curve (“Constrained Concept Refinement”) and the orange curve (“Baseline without CCR”) in Figure 2 illustrate the performance with and without concept refinement. We will explicitly mention this comparison as an ablation study in the revised version.
> > >
> > > *Concept dispersion:* In the anonymous link (plot_3), we have shown the effect of concept dispersion on the correlation between the concepts. We further plan to evaluate the performance of CCR with and without concept dispersion and include it in the ablation study section of the revised manuscript.
> > >
> > > *Sparse coding via hard-thresholding:* Another component of our method that can benefit from ablation study is the use of hard-thresholding to generate the sparse code $\mathbf{s}$. In the revised manuscript, we will replace this process with classical OMP algorithm. Note that the resulting model will still differ from IP-OMP due to the existence of the concept dispersion step. If time and resources allow, we also plan to implement task driven DL method [1] in this module. Based on the observation from some pilot experiments, our hard-thresholding approach demonstrates strong computational efficiency compared with other counterparts.
> > >
> > > [1] Mairal, J., Bach, F., and Ponce, J. Task-driven dictionary learning. IEEE transactions on pattern analysis and machine intelligence,

---

### Official Review · Reviewer_LpAZ · 2025-03-22

**Overall Recommendation:** 2

**Summary:**

The paper proposes CCR to improve both prediction accuracy and model interpretability.

**Claims And Evidence:**

It does not have the impact statements and ends abruptly. Seems like a work in progress?

**Essential References Not Discussed:**

Main papers that come to mind would probably be outside the scope of this paper

**Experimental Designs Or Analyses:**

See methods and eval criteria

**Methods And Evaluation Criteria:**

- metrics for interpretability like avg explanation length and avg concept embedding deviation are proxy measures. It remains unclear how well these reflect human understanding. There is a better figure on this in the Information Maximization Perspective of Orthogonal
Matching Pursuit with Applications to Explainable AI by Chattopadhyay et al. paper.
- Why is CIFAR-10 curve in Figure 2 for baseline flat? Puts into question if this was a fair comparison.
- I wonder if the theoretical analysis depends on overly strong assumptions that may not generalize

**Other Comments Or Suggestions:**

Missing Impact Statements.

**Other Strengths And Weaknesses:**

See above

**Questions For Authors:**

See other sections

**Relation To Broader Scientific Literature:**

Seem very tightly tied to Information Maximization Perspective of Orthogonal Matching Pursuit with Applications to Explainable AI by Chattopadhyay et al. including paper structure.

**Theoretical Claims:**

Not well enough.

---

> ### Author Rebuttal · Authors · 2025-04-01
>
> We sincerely thank the reviewer for the time and effort invested in evaluating our manuscript. Should the reviewer find our clarifications and revisions satisfactory, we would be grateful if they would consider raising their score in the final assessment.
>
> **Claims And Evidence**
>
> We apologize for not including the impact statement of the paper. We will include the following section in the final manuscript:
>
> Conclusion and Impact Statements
>
> This paper introduces Constrained Concept Refinement (CCR), a principled framework that helps bridge the long-standing gap between interpretability and accuracy in machine learning. By constraining the refinement of concept embeddings to lie within a small neighborhood around their initial values, CCR enables interpretable-by-design models to improve prediction performance without compromising their explainability. Our theoretical and empirical results demonstrate the effectiveness of CCR across a variety of tasks and datasets, both in terms of predictive performance and computational efficiency.
>
> The broader impact of this work lies in its potential to advance the practical adoption of interpretable machine learning methods in real-world settings. In particular, the computational efficiency afforded by CCR may facilitate the deployment of explainable artificial intelligence (XAI) techniques in resource-constrained environments.
>
> From an ethical perspective, the capacity to generate explanations that are faithful, stable, and aligned with human-interpretable concepts contributes to addressing critical concerns related to algorithmic accountability and bias. For the proposed method, hyperparameter tuning remains a crucial yet underexplored component for ensuring effective performance in practical applications. This aspect warrants careful consideration to ensure that the resulting outputs are both reliable and justifiable.
>
> **Methods And Evaluation Criteria**
>
> > metrics for interpretability like...
>
> We thank the reviewer for the constructive feedback. To address the comment, we conducted additional experiments to better illustrate the relationship between performance and sparsity. See the Plot 1 from this [anonymous link](https://anonymous.4open.science/r/ICML_Review-C283).
>
> > Why is CIFAR-10 curve ...
>
> The main reason for the lack of obvious improvement on the CIFAR-10 dataset is that we report test accuracy **after** each epoch. Since CIFAR-10 is relatively easy to train, its performance potential is largely realized after a single pass through the training data (i.e., after the first epoch). If we instead report test accuracy **before** each epoch, a clear improvement becomes apparent after the first epoch. This improvement is shown in the Plot 2 of this [anonymous link](https://anonymous.4open.science/r/ICML_Review-C283).
>
> > I wonder if the theoretical ...
>
> We thank the reviewer for raising this important point. Indeed, the concept of interpretability remains elusive from a theoretical perspective, and there is currently no consensus in the community on how to rigorously quantify it. As a result, existing works typically take one of two approaches: (i) relying on human annotations and evaluations to assess interpretability, while omitting theoretical guarantees [1-4]; or (ii) studying a well-defined generative model, where samples are drawn from an underlying ground-truth concept embedding, allowing interpretability to be quantified as the distance between the learned and true concept embeddings. Our work incorporates both of these perspectives.
>
> Admittedly, the generative model we consider is not fully general, and its assumptions may not hold in real-world settings. However, as we clearly state in the paper, our motivation for studying this model lies elsewhere: it provides a controlled test-bed in which the limitations of existing methods can be rigorously examined, and our proposed approach can be shown to overcome them effectively. Moreover, this generative model has been extensively explored in prior work. For instance, [1] uses a variant of it as a pragmatic compromise between theoretical tractability and empirical relevance. It is also closely related to a well-established line of research in dictionary learning, where similar generative models have been successfully applied.
>
> Reference:
>
> [1] A Chattopadhyay, R Pilgrim,R Vidal. Information maximization perspective of orthogonal matching pursuit with applications to explainable AI.
>
> [2] B Zhou, Y Sun, D Bau, A Torralba. Interpretable basis decomposition for visual explanation.
>
> [3] PW Koh, T Nguyen, YS Tang, S Mussmann, E Pierson, B Kim, P Liang. Concept bottleneck models.
>
> [4] M Yuksekgonul, M Wang, J Zou. Post-hoc concept bottleneck models
>
> [5] T Oikarinen, S Das, L Nguyen, T Weng. Label-free concept bottleneck models

---

### Decision · Program_Chairs · 2025-05-01

**Decision:**

Accept (poster)

**Comment:**

This paper introduces Constrained Concept Refinement (CCR), a method aimed to improve the predictive performance of interpretable-by-design models without sacrificing interpretability. The core idea is to refine the concept embeddings (typically derived from CLIP or similar pretrained sources) within a constrained neighborhood, thus preserving interpretability while correcting inaccuracies. The authors provide both theoretical analysis under a generative model setting and empirical validation on image classification tasks (CIFAR-10/100, CUB, ImageNet, Places365), demonstrating improved accuracy and runtime efficiency over baselines (IP-OMP and CBMs).

**Strengths**:
* CCR provides a mechanism to refine concept embeddings that are often noisy/
* The theoretical section is grounded in recent work and adds formal justification for the proposed approach, particularly in terms of recovery guarantees under suitable
* Empirical results are strong, showing consistent accuracy improvements while reducing explanation sparsity and computational cost.
* Rebuttal engagement was excellent—authors provided extensive clarifications, new experiments (e.g., ablations, spectral truncation, concept dispersion), and acknowledged limitations in interpretability evaluation and real-world applicability.

**Reviewer Sentiment**:
* Reviewer 1v3q (Accept) was initially critical but raised their score post-rebuttal after the authors clarified theoretical assumptions and experimental robustness. They still encourage further polish in a revision.
* Reviewers b4dR and grrY (Weak Accept) both acknowledge the value of the contribution but express reservations around presentation clarity (grrY), interpretability definitions (b4dR), and missing baselines or failure analyses. These were substantially addressed during the discussion.
* Reviewer LpAZ (Weak Reject) had a shorter and more skeptical review, flagging concerns around missing impact statements, unclear theory, and potential similarity to prior work. However, these were specifically and constructively handled in the rebuttal, including a newly drafted impact section, extended empirical plots, and a stronger contextualization of the contribution. And the reviewer did not engage with the authors further.

**Limitations and Remaining Issues**:
* The definition and evaluation of “interpretability” remain somewhat underspecified, and no large-scale human evaluation is presented—though this is common in the literature.
* The theoretical guarantees depend on assumptions (e.g., full-rank, orthogonality) that are naturally limited and may not always hold in practice. While the authors discuss this, some readers may still find the gap between theory and practice notable.
* The presentation could benefit from tighter exposition and a clearer narrative arc, which the authors have committed to improving in the final version.


**Summary**

Despite some critiques, this paper offers a nice and timely contribution to interpretable machine learning. It builds on a strong line of recent work, introduces a novel and theoretically motivated refinement strategy, and demonstrates consistent gains in accuracy and efficiency.